# Functional brain reconfiguration during sustained pain

**Jae-Joong Lee[1,2], Sungwoo Lee[1,2,3], Dong Hee Lee[1,2,3], Choong-Wan Woo[1,2,3]\***

[1]Center for Neuroscience Imaging Research, Institute for Basic Science, Suwon, Republic of Korea; [2]Department of Biomedical Engineering, Sungkyunkwan University, Suwon, Republic of Korea; [3]Department of Intelligent Precision Healthcare Convergence, Sungkyunkwan University, Suwon, Republic of Korea

**Abstract** Pain is constructed through complex interactions among multiple brain systems, but it remains unclear how functional brain networks are reconfigured over time while experiencing pain. Here, we investigated the time-varying changes in the functional brain networks during 20 min capsaicin-induced sustained orofacial pain. In the early stage, the orofacial areas of the primary somatomotor cortex were separated from other areas of the somatosensory cortex and integrated with subcortical and frontoparietal regions, constituting an extended brain network of sustained pain. As pain decreased over time, the subcortical and frontoparietal regions were separated from this brain network and connected to multiple cerebellar regions. Machine-learning models based on these network features showed significant predictions of changes in pain experience across two independent datasets ($n$ = 48 and 74). This study provides new insights into how multiple brain systems dynamically interact to construct and modulate pain experience, advancing our mechanistic understanding of sustained pain.

## Editor's evaluation

This article will be of great interest to researchers interested in the brain mechanisms of pain. It shows how the connectivity of brain networks associated with sustained pain changes over time. These findings are supported by compelling fMRI analyses of a tonic pain paradigm in two cohorts of healthy human participants. These important insights advance the understanding of the brain mechanisms of sustained pain, which is the hallmark of chronic pain as a major healthcare problem.

**\*For correspondence:**
waniwoo@g.skku.edu

**Competing interest:** The authors declare that no competing interests exist.

## Introduction

The experience of pain unfolds over time and dynamically changes (*Kucyi and Davis, 2015*), and the sustained and spontaneously fluctuating nature is a key characteristic of clinical chronic pain (*Apkarian et al., 2001*). Pain is known to consist of multiple component processes ranging from sensory and affective to cognitive and motivational processes (*Melzack and Casey, 1968*), and thus the construction and modulation of pain are subserved by distributed brain systems (*Coghill, 2020*; *Mano and Seymour, 2015*). Importantly, the degree to which each component process contributes to pain experience changes over time (*Hashmi et al., 2013*). For example, fear-avoidance (*Vlaeyen and Linton, 2012*), maladaptive coping (*Jensen et al., 1991*), and learning and memory (*Apkarian et al., 2009*; *Phelps et al., 2021*) become more important in explaining pain experience as the pain becomes chronic. Therefore, identifying the whole-brain network features that can explain and predict the natural fluctuations and changes of sustained pain (*Farmer et al., 2012*; *Kucyi and Davis, 2015*) is crucial to understanding why pain naturally decreases in some cases and individuals but not in others (*Ploner et al., 2011*). In this study, we examined the reconfiguration of whole-brain functional

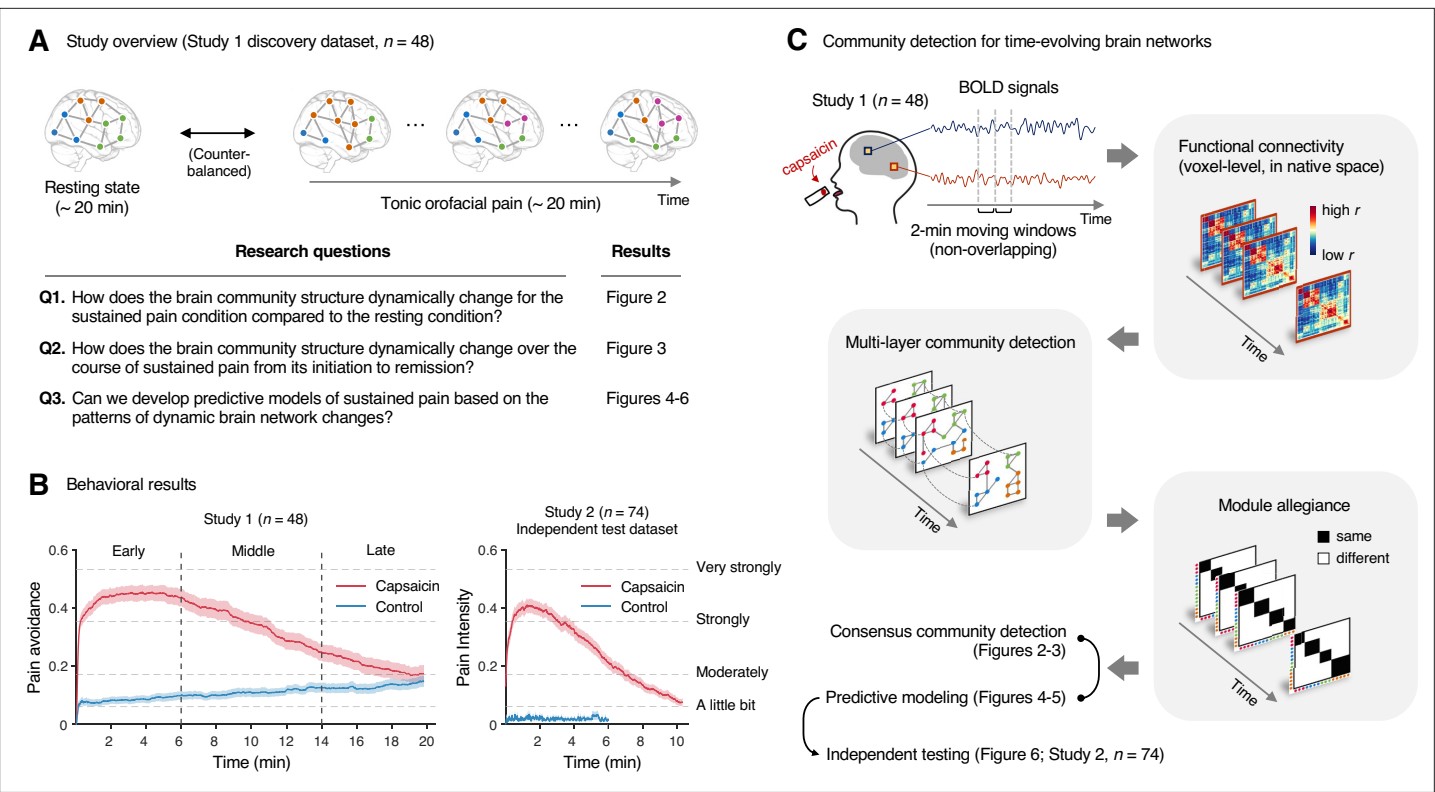

**Figure 1.** Study overview and behavioral results. (**A**) Three main research questions of the current study. We aimed to answer the research questions by examining the time-varying patterns of functional brain network reconfiguration during 20 min of tonic orofacial pain experience and comparing them to the pain-free resting state. (**B**) Behavioral results. We asked participants to continuously report their pain in the scanner using either a pain avoidance rating scale ('how much do you want to avoid this experience in the future?') for Study 1 ('discovery dataset,' n = 48) or a pain intensity rating scale ('how intense is your pain?') for Study 2 ('independent test dataset,' n = 74). The anchors of the scale (the horizontal dashed lines) were based on a modified version of the general labeled magnitude scale (gLMS). The vertical dashed lines show how we define the early, middle, and late periods of sustained pain. The solid lines represent group mean ratings (red for capsaicin, and blue for control), and the shading represents standard errors of the mean (SEM). (**C**) The overview of main analyses. Voxel-level functional connectivity was estimated in the native space using 2 min moving windows and thresholded at 0.05 network density (see 'Materials and methods' for details). We identified the time-evolving network community structures from these suprathreshold connectivity matrices using the multilayer community detection method (*Mucha et al., 2010*), and calculated the module allegiance (*Bassett et al., 2011*). Using the module allegiance values as input features, we either identified group-level consensus community structures or conducted predictive modeling and tested the models on Study 2 independent test dataset (n = 74). Colored circles in the multilayer community detection and module allegiance analysis represent different community labels. Black- and white-colored boxes in the module allegiance matrices indicate whether or not the two different network nodes have the same community affiliation, respectively.

networks underlying the natural fluctuation in sustained pain to provide a mechanistic understanding of the brain responses to sustained pain. To this end, our key research questions include (*Figure 1A*): How does the brain community structure change (i) for the sustained pain condition compared to the resting condition and (ii) over the course of sustained pain from its initiation to remission? (iii) Can we develop predictive models of sustained pain based on the patterns of brain network changes?

To answer these questions, we conducted two functional magnetic resonance imaging (fMRI) experiments with 48 and 74 healthy participants while they were experiencing 20 and 10 min of capsaicin-induced tonic orofacial pain, respectively. Tonic pain has long been used as an experimental model of clinical pain (*Dubuisson and Dennis, 1977*), and our previous study demonstrated that capsaicin-induced tonic orofacial pain shows network-level brain representations similar to clinical pain, suggesting its clinical relevance (*Lee et al., 2021*). The application of a capsaicin-rich hot sauce on a participant's tongue can effectively elicit pain sensation for approximately 10 minutes, followed by a slow remission of pain toward the end of the scan. This experimental paradigm is particularly suitable for identifying reliable features of the brain network for different periods of pain (i.e., initiation to remission), which has been challenging in clinical fMRI studies because of the small variance and heterogeneity of clinical pain trajectories within a few sessions of fMRI scans.

We first identified the time-evolving functional brain network structures using a community detection method for multi-layer networks (*Mucha et al., 2010*) using data from Study 1 ('discovery dataset,' *n* = 48). We conducted this network analysis at the voxel level on an individual's native brain space to fully utilize personalized and fine-grained pattern information. We then compared the group-level representative brain community structures between the sustained pain condition versus the no-pain resting condition (the first research question) and between different phases of pain, including its early, middle, and late periods (the second research question). Finally, we developed machine-learning models based on the brain community patterns using the data from Study 1, either to classify pain versus no-pain conditions or to predict the dynamic changes in pain ratings, and tested the performances of these models on an unseen dataset (Study 2, 'independent test dataset,' *n* = 74) (the third research question).

The results showed that the orofacial areas (i.e., ventral part) of the primary somatomotor regions were separated from the other (i.e., dorsal) primary somatomotor regions and instead integrated with subcortical (e.g., thalamus, basal ganglia) and frontoparietal regions (e.g., dorsolateral prefrontal cortex) during sustained pain. Interestingly, this pain-induced somatomotor-dominant brain community structure changed over time. The subcortical and frontoparietal regions affiliated with the somatomotor-dominant network in the early period of pain were gradually separated from the somatomotor network and strongly connected to multiple cerebellar regions as pain decreased. Machine-learning models based on these brain network organization patterns could discriminate the sustained pain from the pain-free control condition and predict dynamic changes in pain experience including pain avoidance and pain intensity over time. These models were further generalized to the independent tonic pain dataset (Study 2).

Overall, this study contributes to the understanding of dynamic interactions among multiple functional brain networks in response to sustained pain, offering new insights into the mechanistic understanding of chronic pain.

## Results
### Experimental design and behavioral results

In Study 1 ('discovery dataset'), we scanned 48 participants while we delivered the capsaicin-rich hot sauce onto the participants' tongues ('capsaicin' condition) or had the participants rest without stimuli ('control' condition). The fMRI scan duration was 20 min for both conditions, which was sufficient to cover the entire period of sustained pain from its initiation to the complete remission. During the scan, we asked participants to report the continuous changes in pain avoidance by asking the following question, "How much do you want to avoid this experience in the future?" (For details of the rating procedure, please see *Appendix 1—figure 1*.) With this question, we aimed to measure the continuous changes in the avoidance motivation induced by sustained pain, which is known as a core component of clinical chronic pain (*Vlaeyen and Linton, 2012*). We employed a modified version of the general labeled magnitude scale (gLMS) to better represent pain experience at the super-high pain range (*Bartoshuk et al., 2004*). The order of the experimental conditions was counterbalanced across participants.

The pain avoidance ratings for the capsaicin condition were higher than those for the control condition throughout the 20 min of the experiment (*Figure 1B*), $\beta$ = 0.11 ± 0.01 (mean ± standard error of the mean [SEM]), $z$ = 4.25, p=$2.10 \times 10^{-5}$ (multilevel general linear model with bootstrap tests, 10,000 iterations, two-tailed, gender and the order of experimental conditions were modeled as covariates), indicating that the capsaicin stimulation effectively induced sustained pain experience and avoidance motivation. The pain avoidance ratings during the capsaicin condition exhibit an evident rise and fall. The differences in the pain avoidance between the capsaicin versus control conditions became nonsignificant toward the end of the scan (from 17.3 min to the end, two-tailed ps>0.05, paired *t*-test, $BF_{01}$ = 1.01–4.71), suggesting that pain subsided. A similar rise-and-fall pattern of pain ratings was observed in Study 2 with a pain intensity question, "How intense is your pain?", except the short duration of pain because of the smaller amount of pain stimulus (i.e., hot sauce) compared to Study 1. Note that the overall trend of pain ratings over time was similar across participants because of the characteristics of our experimental design, which has also been observed in the previous studies that used

oral capsaicin (*Berry and Simons, 2020*; *Lu et al., 2013*; *Ngom et al., 2001*). However, also note that each individual's time course of pain ratings was not entirely the same (*Appendix 1—figures 2 and 3*).

## Community detection of time-evolving brain networks

To examine the reconfiguration of the whole-brain functional networks over time, we used a multilayer community detection approach (*Mucha et al., 2010*) based on the Louvain community detection algorithm (*Blondel et al., 2008*). This approach is designed to find community structures of time-evolving networks by connecting the same nodes across different time points (*Figure 1C*) and has been successfully used in previous studies on brain network reconfigurations for learning (*Bassett et al., 2011*; *Bassett et al., 2015*), working memory (*Braun et al., 2015*; *Finc et al., 2020*), planning and reasoning (*Pedersen et al., 2018*), emotion (*Betzel et al., 2017*), and pharmacological intervention (*Braun et al., 2016*). In this study, we used this approach to examine the temporal changes of brain network structures during sustained pain, which cannot be done with conventional functional connectivity-based analyses (*Lee et al., 2021*).

We first divided the fMRI data into 2 min nonoverlapping time windows (i.e., 10 time-bins) and estimated the functional connectivity for each time window using Pearson's correlation for each participant and each condition. Although the long duration of the time window without overlaps may obscure the fine-grained temporal dynamics in functional connectivity patterns, we chose to use this long time window to obtain more reliable estimates of network structures and their transitions as previous literature (*Bassett et al., 2011*; *Robinson et al., 2015*). Importantly, we computed the functional connectivity at the voxel level on each participant's native brain space to avoid an arbitrary choice of brain parcellation schemes and minimize the potential loss of information due to anatomical normalization. Also, since many of the graph analytics were developed for sparse networks (*Newman, 2010*) and voxel-level connectivity data were likely to contain many spurious correlations, we applied proportional thresholding to functional connectivity matrices. Because an arbitrary choice of the threshold can have a substantial impact on the results (*van den Heuvel et al., 2017*), we selected the optimal threshold level (0.05; top 5% of connections) that maximized the differences in commonly used network attributes between the capsaicin versus control conditions (*Appendix 1—figure 4*; for details of how we determined the optimal threshold level, see 'Materials and methods'). We used this threshold level for the remaining analyses.

## Differences in consensus community structure between sustained pain versus no pain (Q1)

To address the first research question, "How does the brain community structure change for the sustained pain condition compared to the no-pain resting condition?", we compared the consensus brain community structures for the capsaicin versus control conditions. Here, the consensus community means the group-level representative structures of the distinct community partitions of individuals. To determine the consensus community across different individuals and times, we first obtained the module allegiance (*Bassett et al., 2011*) from the community assignment of each individual. Module allegiance assesses how much a pair of nodes is likely to be affiliated with the same community label and is defined as a matrix $T$ whose element $T_{ij}$ is 1 when nodes $i$ and $j$ are assigned to the same community and 0 when assigned to different communities. This conversion of the categorical community assignments to the continuous module allegiance values allows group-level summarization of different community structures of individuals. We projected the module allegiance values onto the common MNI space and then averaged them across participants and time, which yielded a group-level module allegiance matrix that ranges from 0 to 1 (*Figure 2A*). Finally, we obtained the consensus community by applying a community detection algorithm to the group-level module allegiance matrix (*Bassett et al., 2013*). For more details on the consensus community detection, see 'Materials and methods'.

The consensus community structures of capsaicin and control conditions are illustrated in *Figure 2B*. We identified four main brain communities across both conditions and found the most prominent differences between the capsaicin versus control conditions in Community 2, of which the size became larger in the capsaicin condition than in the control condition (*Figure 2C*). *Figure 2D* shows the network decomposition of the communities using Yeo's large-scale network scheme (*Yeo et al., 2011*)—the major player for each community was the visual network for Community 1, the somatomotor network

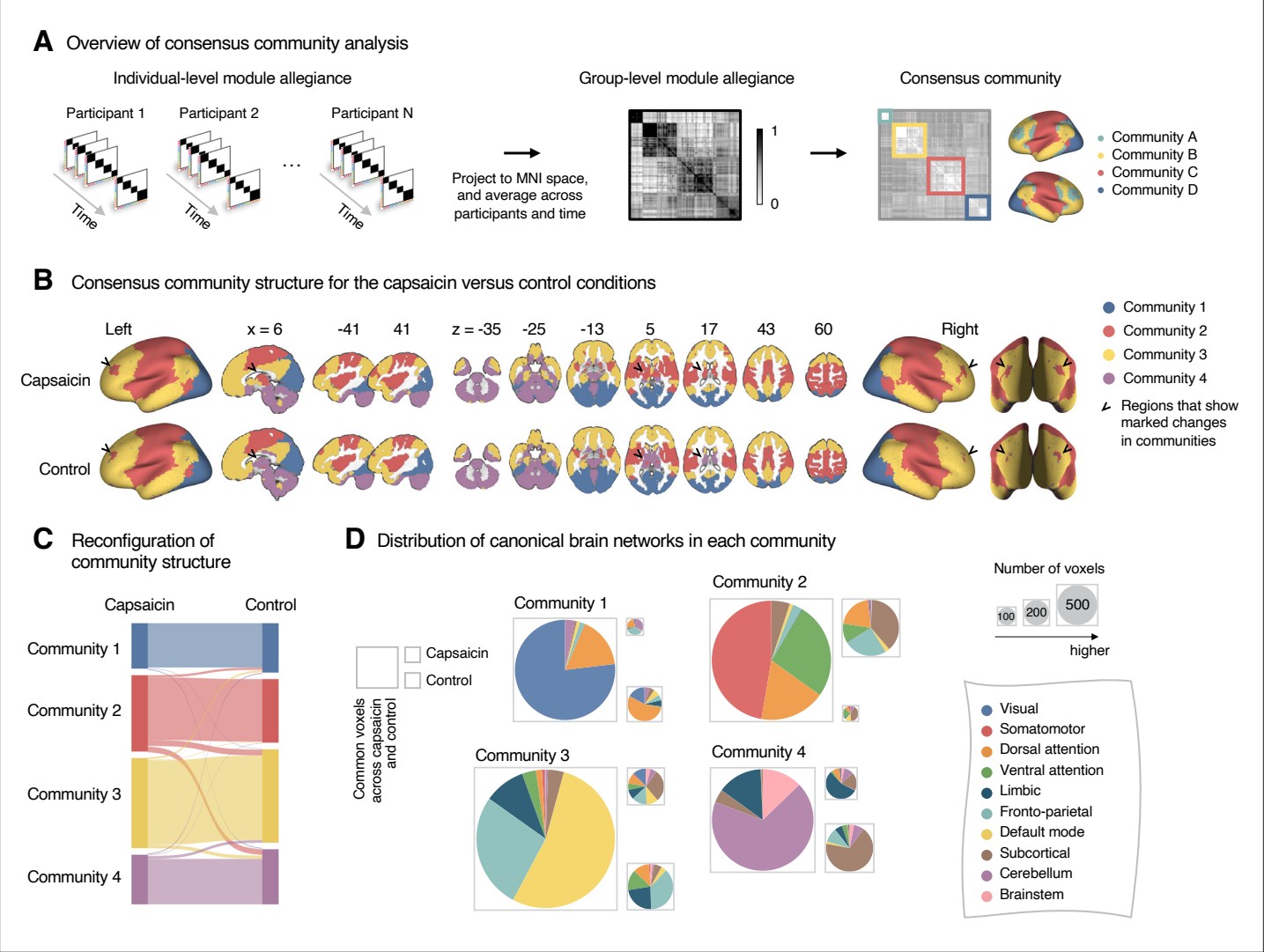

**Figure 2.** Reconfiguration of community structure for the sustained pain versus control conditions. (**A**) Analysis overview. Individual module allegiance values computed in the native spaces were projected onto MNI space and averaged across all participants and all time-bins (i.e., ten bins) to yield a group-level module allegiance matrix. Then, we identified the group-level consensus community structures by decomposing the group-level module allegiance matrix into distinct modules using the Louvain community detection algorithm (*Blondel et al., 2008*). (**B**) Consensus community structure for the capsaicin versus control conditions. Community colors were determined based on the canonical network membership of the largest proportion of voxels across the two conditions. V-shaped arrows indicate the regions that showed marked changes in communities. (**C**) Voxel-wise changes in community assignments between the capsaicin and control conditions. (**D**) Proportions of 10 canonical brain networks—7 resting-state large-scale networks (*Yeo et al., 2011*), subcortical, cerebellum, and brainstem regions—for different communities. The large square on the left shows the network composition of the voxels common across the capsaicin and control conditions, and the squares on the upper and lower right represent the voxels uniquely assigned to the community for the capsaicin or control conditions, respectively. The sizes of the squares are proportional to the number of voxels.

for Community 2, the default mode network for Community 3, and the cerebellum for Community 4. The expansion of Community 2 during the capsaicin condition was mainly driven by the fronto-parietal network regions from Community 3 (including the dorsolateral prefrontal and inferior parietal regions) and subcortical regions from Community 4 (including the thalamus and basal ganglia regions; *Figure 2D* and *Appendix 1—figure 5*). Permutation tests confirmed that the community assignment in the frontoparietal and subcortical regions showed significant changes between the capsaicin versus control conditions (*Appendix 1—figure 6A*). These results suggested that sustained pain induced a global functional reconfiguration of multiple brain networks that included segregation from the default mode network and cerebellar regions and integration with the somatomotor network

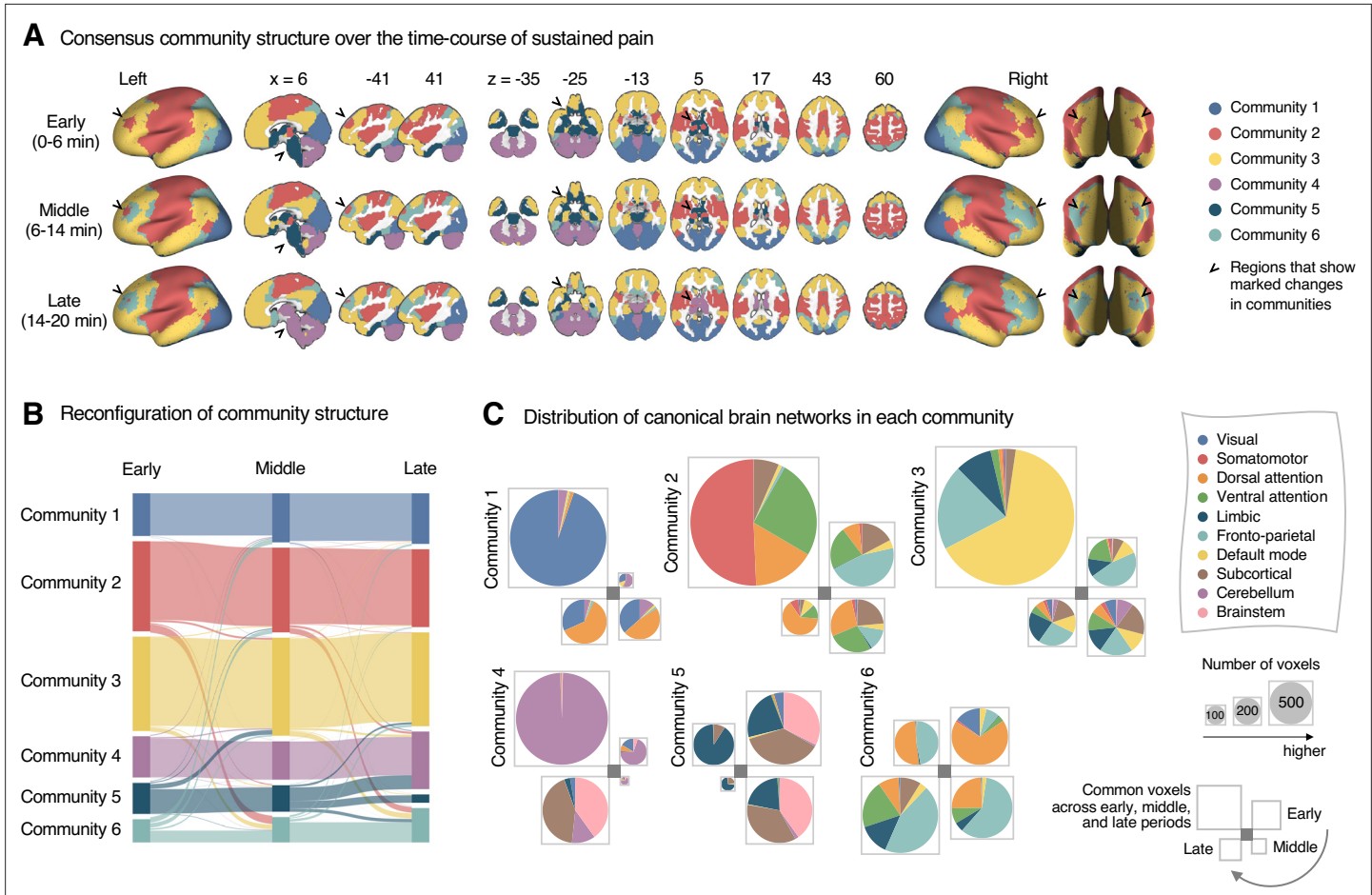

**Figure 3.** Reconfiguration of community structure over the time course of sustained pain experience. (**A**) Consensus community structures of the early (0–6 min), middle (6–14 min), and late (14–20 min) periods of sustained pain. Colors indicate distinct community assignments and were determined based on the canonical network membership with the largest proportion of voxels across the periods. (**B**) Voxel-wise changes in community assignments from the early to late periods of pain. (**C**) Proportions of 10 canonical brain networks for different communities. The large square on the upper left shows the network composition of the voxels common across all periods, and the squares on the upper right, lower right, lower left represent the voxels of the early, middle, and late periods of pain after removing the common voxels, respectively. The sizes of the squares are proportional to the number of voxels.

to form an extended brain system for pain, which is a replication of the previous study that termed this enlarged somatomotor-dominant network as a 'pain supersystem' (*Zheng et al., 2019*).

## Changes in consensus community structure over the course of sustained pain (Q2)

Next, to address the second research question, "How does the brain community structure change throughout sustained pain from its initiation to the remission?", we first divided the capsaicin run into three time periods, 'early' (1–3 bins, 0–6 min), 'middle' (4–7 bins, 6–14 min), and 'late' (8–10 bins, 14–20 min). These three time periods corresponded to (i) initiation and maintenance of pain, (ii) gradual pain decrease, and (iii) full remission of pain, respectively (*Figure 1B*). Similar to the analysis shown in *Figure 2A*, we then averaged module allegiance matrices across participants for each period and calculated the consensus community structures from the averaged allegiance matrices.

As shown in *Figure 3*, the algorithm detected six consensus communities across time periods. The first four communities (communities 1–4) were consistent with the communities identified in the previous analysis based on the averaged data for the capsaicin and control conditions (*Figure 2*). Patterns of changes in communities 1–4 were similar to those from previous results. For example, Community 2 (somatomotor-dominant) was larger in the early period than in the middle and late

periods (i.e., the emergence of the extended somatomotor-dominant community during capsaicin condition), and the frontoparietal network and subcortical regions were the main drivers of this change (*Figure 3A and C* and *Appendix 1—figure 7A*).

Community 6, which was one of the newly detected communities, displayed an interesting pattern of changes over the course of sustained pain. The community mainly consisted of the frontoparietal and dorsal attention network regions and gradually increased in size over time (*Figure 3A and B*). The increase in community size in the later periods was mainly driven by the frontoparietal network regions that were among Community 2 (somatomotor-dominant) during the early period. In the later periods, these frontoparietal regions, including the dorsolateral prefrontal and inferior parietal regions, changed their affiliation to Community 6, forming a separate frontoparietal-dominant community (*Appendix 1—figure 7C*).

Community 5 also showed an interesting pattern. Community 5 mainly consisted of the limbic cortices and subcortical and brainstem regions during the early and middle periods. However, the subcortical and brainstem regions changed their affiliation to Community 4 (cerebellum-dominant community), leaving only a small number of voxels within the limbic regions in Community 5 in the late period (*Figure 3* and *Appendix 1—figure 8A*). The composition of Community 5 during the early period suggested that the brainstem regions were closely linked with the limbic brain regions, including the medial temporal lobe structures (e.g., hippocampus, amygdala, and parahippocampal gyrus) and the temporal pole, consistent with recent findings that showed the involvement of the spino-parabrachio-amygdaloid circuit in sustained pain (*Chiang et al., 2019*; *Huang et al., 2019*; *Rodriguez et al., 2017*). In addition, the reconfiguration of Community 4 showed that the cerebellum-dominant community extended its connections to the brainstem and thalamus during the late period of sustained pain (*Figure 3C* and *Appendix 1—figure 8B*), suggesting an important, though less investigated, pain-modulatory role of the cerebellum in pain (*Claassen et al., 2020*; *Moulton et al., 2010*).

These results suggested that decrease in sustained pain induced the dissociation of frontoparietal-somatomotor and limbic-subcortical-brainstem coupling, and the emergence of a separate frontoparietal-dominant community and an enlarged cerebellum-dominant community. Permutation tests further confirmed that the community assignment in the frontoparietal, subcortical, brainstem, and cerebellar regions showed significant changes between the early versus late period of pain (*Appendix 1—figure 6B*).

## Module allegiance-based classifier for sustained pain versus no pain (Q3-1)

To further characterize the functional network changes induced by sustained pain, we conducted predictive modeling using the module allegiance patterns (third research question, "Can we develop predictive models of sustained pain based on the patterns of brain network changes?"). To obtain module allegiance matrices on a common feature space and also to reduce the computational burden for model fitting, we projected a whole-brain parcellation comprising 263 regions defined on the MNI space (Schaefer atlas [*Schaefer et al., 2018*] with the additional brainstem, cerebellum, and subcortical regions; see 'Materials and methods' for details) onto an individual's native space (*Figure 4A*). Then, we averaged module allegiance for each region, resulting in a 263 × 263 module allegiance matrix for each participant and for each time window. Here, high module allegiance indicates the voxels of two regions are likely to be in the same community affiliation, and vice versa. With the module allegiance matrices, we conducted two different types of predictive modeling: classification (developing a support vector machine [SVM] classifier to discriminate the capsaicin condition from the control condition) and regression (developing a principal component regression [PCR] model to predict the fluctuations in sustained pain ratings). We chose to use the SVM and PCR because they are representative linear algorithms for finding the low-dimensional latent components of highly correlated data such as brain networks.

To develop a classifier for the capsaicin versus control conditions, we averaged the module allegiance data across time points, creating one allegiance matrix per person and experimental condition. *Figure 4B* displays the group averages of the module allegiance matrices for the capsaicin condition (lower triangle) and the control condition (upper triangle). As shown in *Figure 4C and D*, the module allegiance-based classifier showed a high classification accuracy for the capsaicin versus

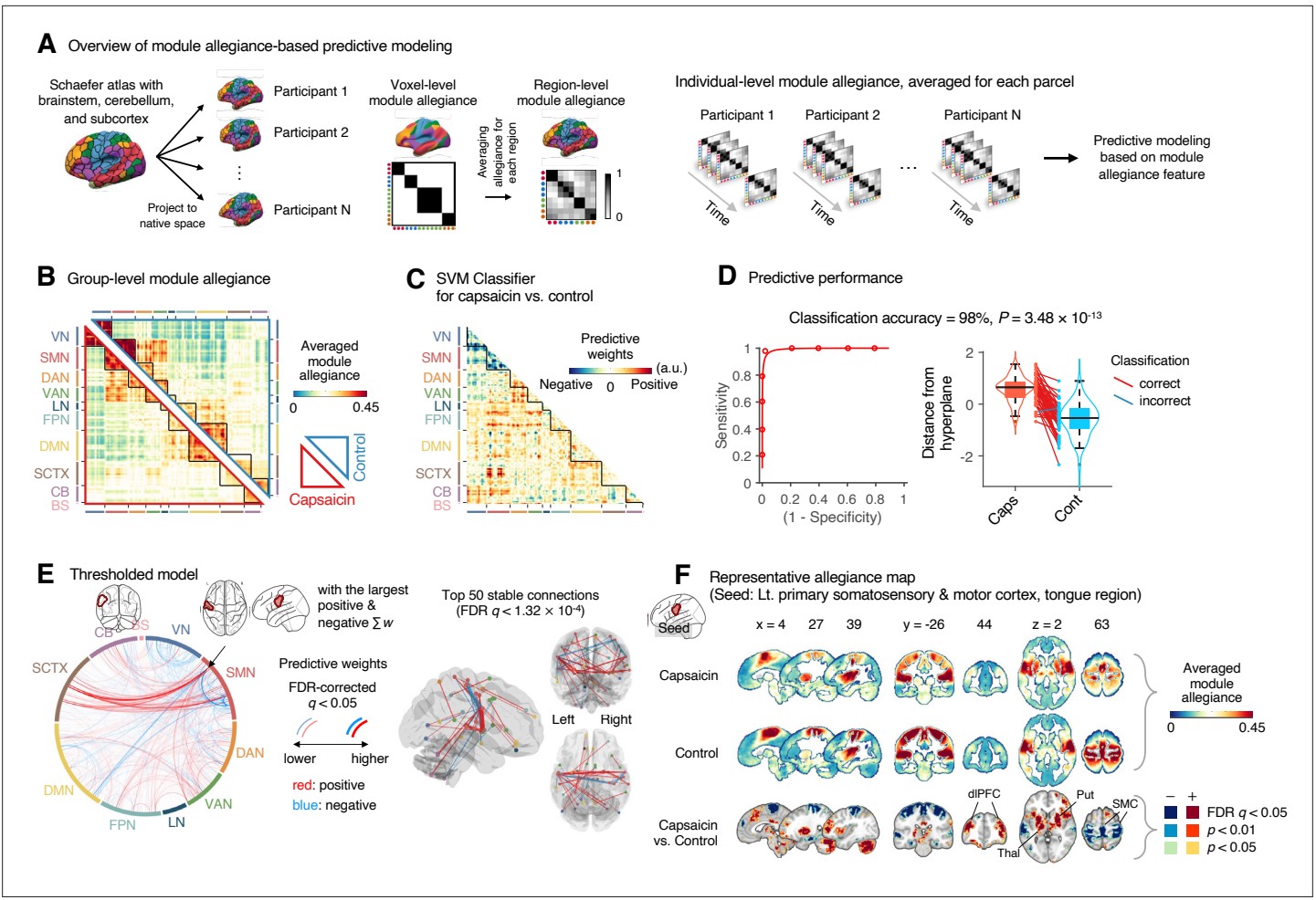

**Figure 4.** Module allegiance-based classifier for sustained pain versus resting. (**A**) The analysis overview of the module allegiance-based predictive modeling. We first projected the whole-brain atlas onto an individual's native brain space. We then averaged the voxel-level module allegiance values for each region to create region-level module allegiance matrices. These region-level module allegiance values were used to predict whether an individual was in pain or not. (**B**) Group-level module allegiance matrix. We averaged the region-level module allegiance matrices across all participants and all time-bins, and sorted the brain regions according to their canonical functional network membership. Lower and upper triangles represent the allegiance matrices of the capsaicin and control conditions, respectively. (**C**) Raw predictive weights of the support vector machine (SVM) classifier. (**D**) To obtain an unbiased estimate of the classifier's classification performance, we conducted the forced-choice test with leave-one-participant-out cross-validation. Left: receiver-operating characteristics (ROC) curve. Right: cross-validated model responses for different conditions. Each line connecting dots represents an individual participant's paired data (red: correct classification, blue: incorrect classification). p-Value was based on a binomial test, two-tailed. (**E**) Thresholded weights based on bootstrap tests with 10,000 iterations. Left: predictive weights thresholded at false discovery rate (FDR)-corrected $q < 0.05$ (which corresponds to uncorrected p<0.003), two-tailed. We indicated the brain region with the largest weighted degree centrality for positive and negative weights based on the thresholded model with the black arrow on the plot. Right: top 50 stable predictive weights (FDR $q < 1.32 \times 10^{-4}$, uncorrected p<1.91 × 10$^{-7}$, two-tailed). (**F**) Seed-based allegiance map for the hub region that we identified in (**E**), left primary somatosensory and motor cortex (tongue region). The bottommost row shows the contrast map for the capsaicin versus control conditions, thresholded at $t_{47} = 2.82$, FDR $q < 0.05$ (uncorrected p<0.007), two-tailed, paired t-test. Put, putamen; dlPFC, dorsolateral prefrontal cortex; Thal, thalamus; SMC, somatomotor cortex; VN, visual network; SMN, somatomotor network; DAN, dorsal attention network; VAN, ventral attention network; LN, limbic network; FPN, frontoparietal network; DMN, default mode network; SCTX, subcortical regions; CB, cerebellum; BS, brainstem.

control conditions in a forced-choice test (accuracy = 98%, p=3.48 × 10$^{-13}$, binomial test, two-tailed; *Figure 4D*), suggesting that the individuals' brain community structures had enough information to detect the existence of sustained pain. For the classification accuracy across all the participants instead of the forced-choice test, please see Appendix 1. When we thresholded the classifier weights to examine the important features of the model based on bootstrap tests (false discovery rate [FDR]-corrected $q < 0.05$, which corresponds to uncorrected p<0.003, two-tailed; the left panel of *Figure 4E*), the results suggested that the important network features of sustained pain included (i) segregation

within the somatomotor network (negative weights among the brain regions within the somatomotor network in *Figure 4E*) and (ii) integration between the subcortical regions and the somatomotor network regions (positive weights between the somatomotor network regions and the subcortical regions in *Figure 4E*). We could not observe the segregation within the somatomotor network in the group-level analysis presented in the previous section (*Figure 2*), suggesting that the individual-level analysis based on module allegiance provides additional insights into the brain network changes during sustained pain.

To further examine the important connections for the dynamic features, we visualized the top 50 stable connections (FDR $q < 1.32 \times 10^{-4}$, uncorrected $p < 1.91 \times 10^{-7}$, two-tailed) using a glass brain (the right panel of *Figure 4E* and *Appendix 1—table 1*). We also displayed these important connections focusing on the somatosensory and insular cortical regions. We then placed them on the sensory homunculus (*Appendix 1—figure 9*) to highlight that the negative weight connections were between the ventral (tongue area) and dorsal parts (other body areas) of the primary somatomotor cortex, and the positive weight connections between the tongue primary somatomotor cortex and some subcortical regions including the thalamus and basal ganglia.

We selected hub regions with the largest weighted degree centrality to provide a more detailed picture of the network reconfiguration, separately for the positive and negative weights, based on the thresholded model at FDR $q < 0.05$. The left ventral primary somatomotor region (tongue area) was selected as the sole hub region for both positive and negative weights. We then obtained a seed-based module allegiance map using the hub region as the seed. As shown in *Figure 4F*, the results reconfirmed that the hub region showed substantially decreased allegiance with the dorsal primary somatomotor regions and increased allegiance with the basal ganglia and thalamus in the capsaicin condition (thresholded at $t_{47} = 2.82$, FDR $q < 0.05$, uncorrected $p < 0.007$, paired $t$-test between capsaicin and control conditions, two-tailed). Moreover, the hub region showed increased allegiance with dorsolateral prefrontal cortex regions, consistent with previous findings that sustained pain induced integration between the frontoparietal and somatomotor networks (*Figure 2*).

## Module allegiance-based prediction model of pain ratings (Q3-2)

Next, we developed a PCR model to predict pain ratings. We used region-level module allegiance matrices across 10 time-bins of all participants as features. Because the number of features ($_{263}C_2$ = 34,453) was higher than the number of observations (10 ratings × 48 participants = 480), we first reduced the dimensionality of the features using principal component analysis (PCA). We then regressed the pain ratings on the principal components (PCs) of module allegiance. We selected the number of PCs that yielded the best predictive performance in leave-one-participant-out cross-validation (*Appendix 1—figure 10*). The newly developed PCR model (*Figure 5A*) showed significantly high prediction performance (mean prediction–outcome correlation $r = 0.29$, $p = 7.27 \times 10^{-6}$, bootstrap test, two-tailed; mean squared error = 0.043 ± 0.006 [mean ± SEM]; *Figure 5B*). For the between-individual prediction–outcome correlation of mean pain ratings, please see Appendix 1. Note that this model performance is biased because we conducted hyperparameter tuning (i.e., the number of PCs) using the same dataset. To obtain a less biased estimate of performance in the training data, we used nested leave-one-participant-out cross-validation that separates the hyperparameter tuning and testing (see 'Materials and methods' for details). The results showed that prediction performance was significant (mean prediction–outcome correlation $r = 0.28$, $p = 1.00 \times 10^{-5}$, bootstrap test, two-tailed, mean squared error = 0.044 ± 0.006 [mean ± SEM], number of PCs = 13.94 ± 0.14 [mean ± SEM]), suggesting that the individuals' brain community structures are predictive of the temporal change of sustained pain.

When we thresholded the predictive model based on bootstrap tests (*Figure 5C*, FDR-corrected $q < 0.05$, uncorrected $p < 9.24 \times 10^{-5}$, two-tailed), the results showed a dissociation between the cerebellum and the subcortical and frontoparietal regions for the high levels of sustained pain (negative weights in the circos plot in *Figure 5C*). Because few connections with positive weights were survived at the FDR correction, we examined the weight patterns at a more liberal threshold (uncorrected $p < 0.05$, two-tailed; the sky blue and reddish purple connections in *Figure 5D*). We observed many positive connections between the somatomotor and frontoparietal network regions, suggesting integration between the somatomotor and frontoparietal networks at high levels of sustained pain. The top 50 stable features (FDR $q < 0.043$, uncorrected $p < 6.09 \times 10^{-5}$, two-tailed; the left panel of

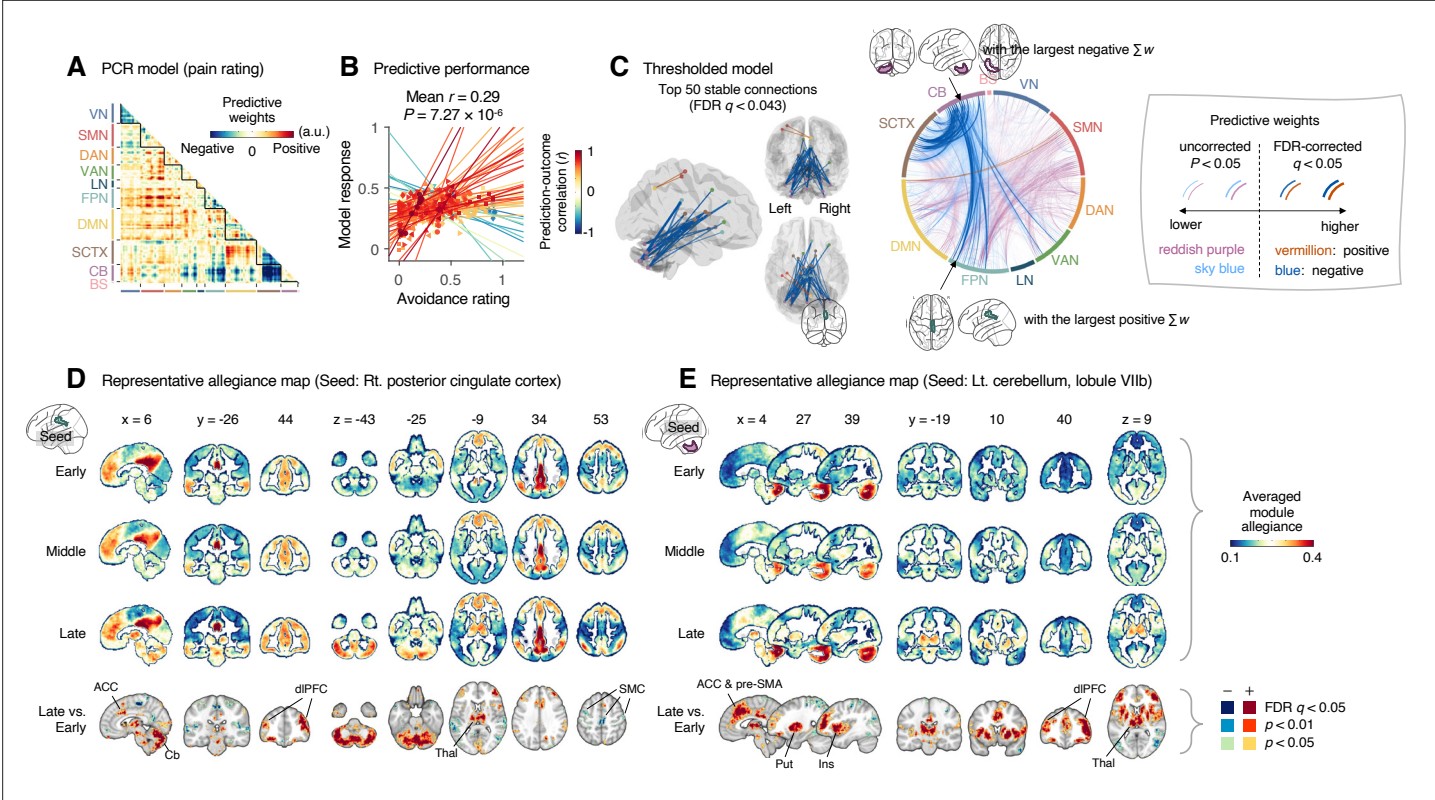

**Figure 5.** Module allegiance-based prediction model of pain rating. (**A**) The raw predictive weights of the principal component regression (PCR) model based on region-level time-bin module allegiance matrices. (**B**) Actual versus predicted ratings. Each colored line (and symbol) represents individual participant's ratings (10 time-bin average ratings per participant) during the capsaicin run (red: higher $r$; yellow: lower $r$; blue: $r < 0$). p-Value is based on bootstrap tests, two-tailed. (**C**) Thresholded weights based on bootstrap tests with 10,000 iterations. Left: top 50 stable predictive weights (false discovery rate [FDR]-corrected $q < 0.043$, which corresponds to uncorrected p<6.09 × 10⁻⁵, two-tailed). Right: FDR $q < 0.05$ (which corresponds to uncorrected p<9.24 × 10⁻⁵, vermillion and blue colors) or uncorrected p<0.05 (reddish purple and sky blue), two-tailed. The brain region with the largest weighted degree centrality separately for positive and negative weights is indicated with the black arrows. (**D, E**) Seed-based allegiance maps for the hub regions across different periods of sustained pain. The right posterior cingulate cortex (PCC) and left cerebellum lobule VIIb were identified as hub regions for the positive and negative weights, respectively. The bottommost row shows the contrast map for the late versus early periods, thresholded at $t_{47} = 3.13$ (**D**) and 3.22 (**E**), FDR $q < 0.05$ (which corresponds to uncorrected p<0.003 [**D**] and 0.002 [**E**]), two-tailed, paired $t$-test. ACC, anterior cingulate cortex; Cb, cerebellum; dlPFC, dorsolateral prefrontal cortex; Thal, thalamus; SMC, somatomotor cortex; pre-SMA, pre-supplementary motor area; Put, putamen; Ins, insula; VN, visual network; SMN, somatomotor network; DAN, dorsal attention network; VAN, ventral attention network; LN, limbic network; FPN, frontoparietal network; DMN, default mode network; SCTX, subcortical regions; CB, cerebellum; BS, brainstem.

*Figure 5C* and *Appendix 1—table 2*) were mostly negative weight connections that were connected to multiple cerebellar regions.

When we selected the hub regions with the largest weighted degree centrality based on the thresholded model at uncorrected p<0.05, the right posterior cingulate cortex (PCC) within the frontoparietal network (MNI center: 6, –26, 30) and the lobule VIIb of the left cerebellum (MNI center: –26, –66, –50) were selected for the positive and negative weights, respectively. *Figure 5D and E* show seed-based module allegiance maps. The right PCC region showed decreased connections with the somatomotor regions and increased connections with the dorsolateral prefrontal cortex, anterior cingulate cortex, thalamus, and cerebellar regions during the late period of pain compared to the early period (thresholded at $t_{47} = 3.13$, FDR $q < 0.05$, uncorrected p<0.003, paired $t$-test, two-tailed). The left cerebellar lobule VIIb showed increased connections with the dorsolateral prefrontal cortex, anterior cingulate cortex and pre-supplementary motor area, insula, thalamus, and basal ganglia during the late period of pain compared to the early period (thresholded at $t_{47} = 3.22$, FDR $q < 0.05$, uncorrected p<0.002, two-tailed). This implicates that both the right PCC and the left cerebellum may play essential roles in integrating frontoparietal and subcortical regions to construct functional communities separate from the somatomotor regions as pain decreased.

These results suggested that the frontoparietal network interacted with the somatomotor network during the early period of sustained pain. However, these connections were weakened, and the frontoparietal and subcortical regions were connected to the cerebellar regions as pain decreased.

## Specificity of the module allegiance-based predictive models

To examine whether the predictive models were specific to pain and the prediction performances were not influenced by confounding variables such as head motion and physiological changes, we conducted additional analyses as shown in *Appendix 1—figures 11–13*. The SVM and PCR models showed significant prediction performances even after controlling for head motion (i.e., framewise displacement) and physiological responses (i.e., heart rate and respiratory rate) (SVM: accuracy = 89%, p=1.41 × 10$^{-7}$, binomial test, two-tailed; PCR: $r$ = 0.20, p=0.003, bootstrap test, two-tailed, mean squared error = 0.159 ± 0.022; *Appendix 1—figures 11 and 12*) and did not respond to the nonpainful but aversive conditions including the bitter taste (SVM: accuracy = 79%, p=6.17 × 10$^{-5}$, binomial test, two-tailed; PCR: $r$ = 0.05, p=0.41, bootstrap test, two-tailed, mean squared error = 0.036 ± 0.006) and aversive odor conditions (SVM: accuracy = 83%, p=3.31 × 10$^{-6}$, binomial test, two-tailed; PCR: $r$ = 0.12, p=0.06, bootstrap test, two-tailed, mean squared error = 0.044 ± 0.004; *Appendix 1—figure 13*), supporting the specificity of our predictive models to pain. For details, please see Appendix 1.

## Testing the module allegiance-based predictive models on an independent dataset

Although our module allegiance-based predictive models demonstrated significant cross-validated prediction performances in our discovery dataset (n = 48), these results could be biased toward the training data. Thus, to provide unbiased estimates of model performance, we tested our models on an independent test dataset (Study 2, n = 74). Study 2 dataset had the same experimental design, but with a shorter scan duration—the 'capsaicin' run was 10 min, and the 'control' run was 6 min. Also, the pain rating was collected using pain intensity scale, not the pain avoidance scale, to ensure generalizability of the allegiance-based PCR model. The test results showed significant classification and prediction performances. The accuracy of the SVM model in classifying the capsaicin versus control conditions was 81%, p=6.22 × 10$^{-8}$, binomial test, two-tailed (*Figure 6A*), and the average prediction–outcome correlation of the PCR model was $r$ = 0.32, p=1.20 × 10$^{-7}$, bootstrap test, two-tailed, mean squared error = 0.041 ± 0.004 (*Figure 6B*).

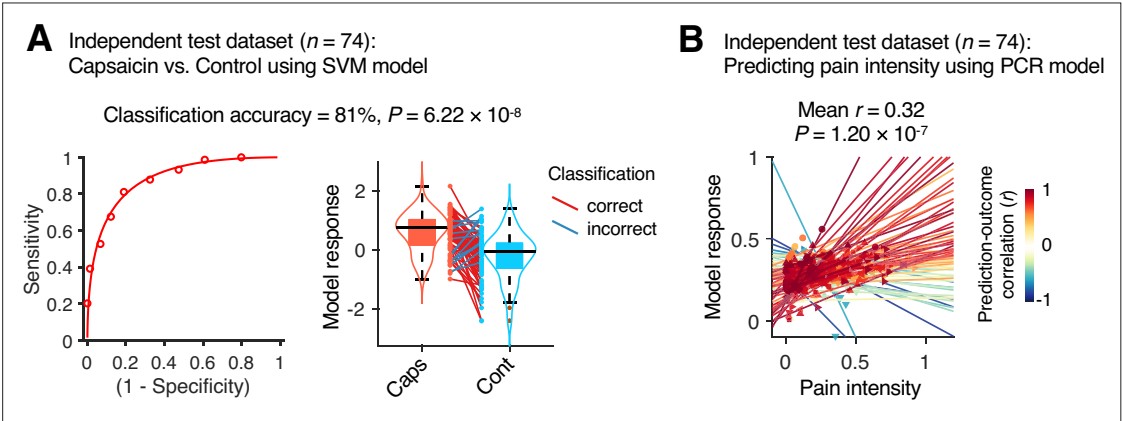

**Figure 6.** Model performance on an independent test dataset. To provide unbiased estimates of prediction performance and test the generalizability of the module allegiance-based predictive models, we tested the support vector machine (SVM) and principal component regression (PCR) models on an independent test dataset (Study 2, n = 74). (**A**) We conducted a forced-choice test to compare the model responses for the capsaicin versus control conditions. Left: receiver-operating characteristics (ROC) curve. Right: model responses for different conditions. Each line connecting dots represents an individual participant's paired data (red: correct classification; blue: incorrect classification). p-Value was based on a binomial test, two-tailed. (**B**) Actual versus predicted ratings. Each colored line (and symbol) represents individual participant's ratings during the capsaicin run (red: higher $r$; yellow: lower $r$; blue: $r$ < 0). p-Value was based on bootstrap tests, two-tailed.

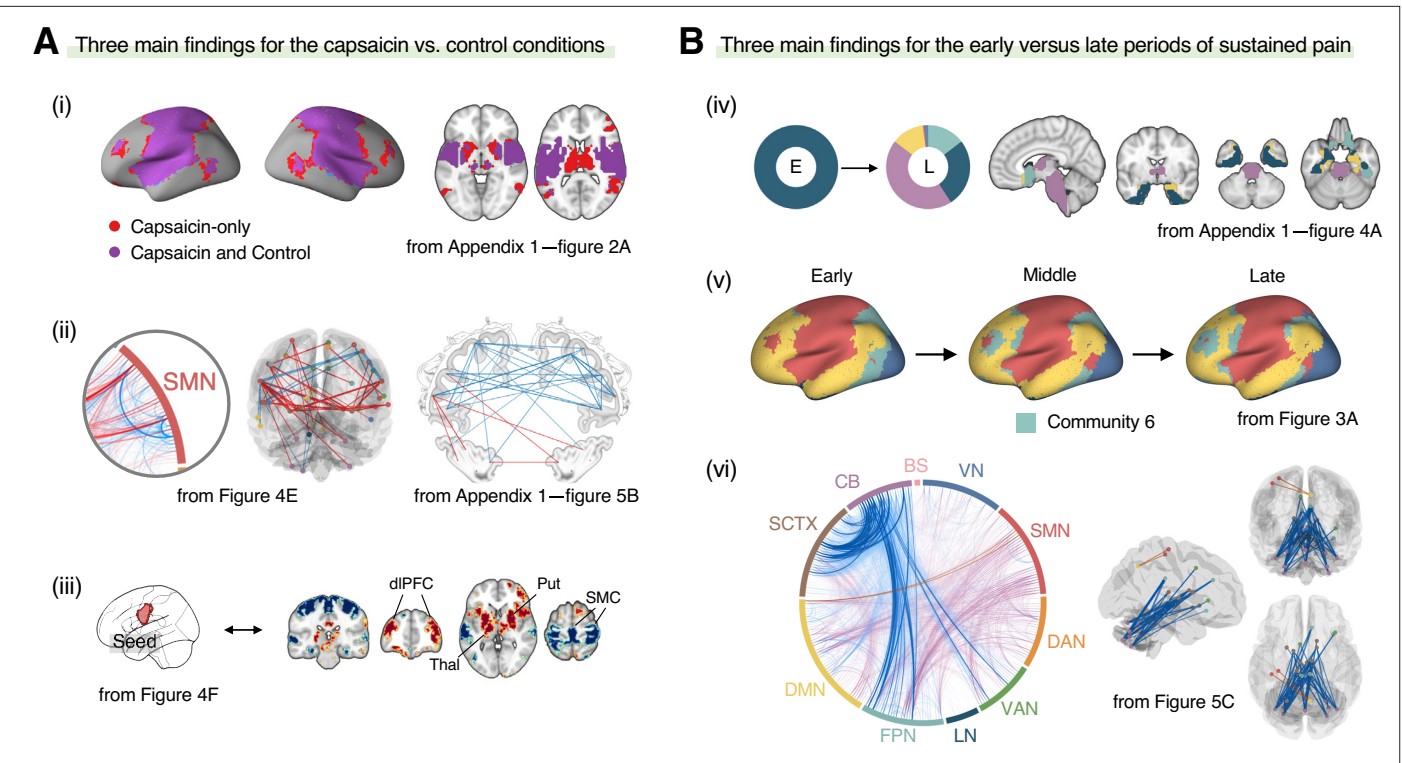

**Figure 7.** Graphical summary of the main findings. We summarized the main findings to determine the main conclusion of this study. (**A**) Three main findings from the comparisons between the functional brain architectures of sustained pain (i.e., capsaicin) versus pain-free resting state (i.e., control) conditions (related to Q1 and Q3 in **Figure 1A**). During sustained pain, (i) an extended somatomotor-dominant community emerged. More specifically, the ventral part of somatomotor cortex regions (i.e., tongue area) was (ii) segregated from the original somatomotor-dominant community and (iii) integrated with the subcortical (e.g., basal ganglia, thalamus) and frontoparietal (e.g., dorsolateral prefrontal cortex) regions during sustained pain. (**B**) Three main findings from the comparisons between the early versus late periods of sustained pain (related to Q2 and Q3 in **Figure 1A**). (iv) In the early period, the brainstem was connected to the limbic brain regions including hippocampus, amygdala, parahippocampal gyrus, and temporal pole (c.f., the spino-brachio-amygdaloid circuit). (v) In the late period, the frontoparietal regions were dissociated from the extended somatomotor-dominant community and formed their own community. (vi) The cerebellar connections to the subcortical and frontoparietal regions were predictive of pain decrease.

## Discussion

In this study, we investigated the reconfiguration of functional brain networks during sustained pain. The main findings of this study are summarized in **Figure 7**. We compared the brain network structures for the capsaicin versus control conditions (**Figure 7A**) and found that (i) the somatomotor dominant community was enlarged during sustained pain by incorporating multiple subcortical and frontoparietal regions, resulting in the emergence of the 'pain supersystem' (**Zheng et al., 2019**). (ii) The ventral primary somatomotor region (tongue area) was segregated from the dorsal primary somatomotor regions and (iii) integrated with the subcortical and frontoparietal regions. When we examined the brain network changes over time within the capsaicin condition (**Figure 7B**), we observed that (iv) the brainstem regions were connected to the limbic brain regions (e.g., hippocampus, amygdala, parahippocampal gyrus, and temporal pole) during the early period of pain, but soon these connections were lost as pain decreased. (v) During the late period, the frontoparietal regions were dissociated from the somatomotor-dominant community and formed their own community. Lastly, (vi) a module allegiance-based predictive model of pain ratings showed that cerebellar connections with the frontoparietal and subcortical regions were important for pain decrease.

Although researchers have highlighted the importance of understanding the 'dynamic pain connectome' (**Kucyi and Davis, 2015**), previous studies were limited mainly to examining summary metrics of dynamic connectivity, such as mean and standard deviation (**Bosma et al., 2018**; **Cheng et al., 2018**; **Cottam et al., 2018**). Therefore, a detailed characterization of how multiple brain networks dynamically interact throughout pain has yet to be conducted. A few studies examined multiple brain

states over a period of pain (*Cheng et al., 2022*; *Lee et al., 2019b*; *Robinson et al., 2015*), but their experimental designs and analyses were not suitable to associate the changes in brain states with different phases of pain, that is, from initiation, peak, and to remission. To better characterize the dynamic changes in functional brain community structures for different phases of pain experience, we used oral delivery of capsaicin to induce sustained and gradually decreasing pain for approximately 20 min. Although future studies need to examine whether our findings can be generalized to other stimulus modalities of sustained pain, this study provides a reliable experimental paradigm to study time-varying network changes during sustained pain and spontaneous pain decrease.

One of the most critical observations in this study is the emergence of an extended somatomotor-dominant brain community in response to sustained pain. This finding is consistent with a previous study in which subcortical and frontoparietal regions were integrated with the somatomotor network to constitute the 'pain supersystem' for a brief phasic heat pain stimulus (*Zheng et al., 2019*). The integration of the somatomotor and frontoparietal regions also correspond to the global workspace theory (*Baars, 2002*), in which the experience of pain requires frontoparietal involvement to enable recurrent processing of nociceptive information across multiple brain areas (*Bastuji et al., 2016*). Also, considering that subcortical structures, such as the thalamus, are important for the early processing of nociceptive signals from the peripheral nervous system, our finding suggests that the recruitment of both bottom-up (subcortical) and top-down (frontoparietal) systems and their integration are crucial to the experience of sustained pain. Previous studies that reported modular reorganization of the somatomotor network (*Mano et al., 2018*) and frontoparietal network (*Barroso et al., 2021*; *Mano et al., 2018*) in chronic pain are also in line with our results. Importantly, this systems-level integration occurred in a somatotopy-specific way—the SVM classifier (*Figure 4*) showed that the ventral primary somatomotor area, which subserves tongue sensation, was segregated from the other somatomotor regions and integrated with the subcortical regions during the early period of pain. These results are consistent with previous studies of sustained pain, which showed that the functional connectivity of the somatotopic primary somatomotor area was decreased for the other somatomotor areas, while being increased for other networks, such as the salience network (*Cheng et al., 2022*; *Kim et al., 2015*; *Kim et al., 2013*; *Kim et al., 2019*; *Lee et al., 2019a*).

In addition, we characterized how the functional brain networks were dynamically reconfigured over the period of sustained pain from the initiation to its resolution and found that the cerebellar regions played an important role in pain decrease. During the early period of pain, the subcortical and frontoparietal regions were integrated with the somatomotor network. However, as pain is resolved, these regions shifted their connections to multiple cerebellar regions. Although pain neuroimaging studies have consistently reported cerebellar activations (*Moulton et al., 2010*), its functional role in pain remains unclear. Interestingly, the cerebellum not only projects efferent nociceptive neurons to the thalamus (*Liu et al., 1993*), but it also has afferent and efferent interconnections via the thalamus to the dorsolateral prefrontal (*Middleton and Strick, 2001*) and inferior parietal (*Clower et al., 2001*) cortices, which are the main components of the frontoparietal network (*Allen et al., 2005*; *Dosenbach et al., 2008*). These neuronal connections implicate the cerebellum's functional roles in top-down pain regulation. Furthermore, recent studies suggest that cerebellum plays an important role in pain-related prediction and its error (*Ernst et al., 2019*), which are keys to the fear-avoidance (*Vlaeyen and Linton, 2012*) and motivation-decision (*Fields, 2018*) model of sustained pain. Therefore, our findings provide additional evidence that the cerebellar regions may be important drivers of endogenous modulation of pain avoidance and spontaneous coping responses to sustained pain. A previous lesion study in which patients with cerebellar damage exhibited increased pain sensitivity and decreased placebo analgesia (*Ruscheweyh et al., 2014*) supports our interpretation.

The predictive modeling based on individual-level module allegiance values (*Figures 4 and 5*) provided novel findings complementary to the group-level consensus community analysis (*Figures 2 and 3*). For example, the dissociation between the ventral versus dorsal primary somatomotor regions during sustained pain was not evident in the group-level analysis of the consensus community structure. This finding suggests that group-level averaging could obscure fine-grained information of functional anatomy (*Gordon et al., 2017*) with distinct regions merging together as the same network community (*Braga and Buckner, 2017*). In this regard, our multivariate pattern-based predictive modeling approach can provide an alternative way to study dynamic changes in functional network structures while preserving fine-grained information about individuals (*Finn et al., 2015*; *Rosenberg*

*et al., 2017*). Moreover, developing models to directly predict the pain ratings is helpful to complement the group-level analysis because the changes in consensus community structure over the early, middle, and late periods only indirectly reflect the different levels of pain. Lastly, the predictive modeling approach can provide information about the robustness and usefulness of the multilayer community detection method by allowing us to test the prediction performance for the discovery dataset and generalizability for the independent test dataset.

An interesting future direction would be to examine whether the current results can be generalized to clinical pain. Experimental tonic pain has been known to share similar characteristics with clinical pain (*Rainville et al., 1992*; *Stohler and Kowalski, 1999*). In addition, in a recent study, we showed that an fMRI connectivity-based signature for capsaicin-induced orofacial tonic pain can be generalized to chronic back pain (*Lee et al., 2021*). Therefore, a detailed characterization of the brain responses to tonic pain has the potential to provide useful information about clinical pain. However, there are also differences between the characteristics of capsaicin-induced tonic pain versus clinical pain. For example, clinical pain continuously fluctuates over time in an idiosyncratic pattern (*Apkarian et al., 2001*), whereas capsaicin-induced tonic pain showed a similar time-course pattern across the participants—that is, increasing rapidly and then decreasing gradually (*Figure 1B*). This typical time course of pain ratings has been reported in previous studies that used oral capsaicin (*Berry and Simons, 2020*; *Lu et al., 2013*; *Ngom et al., 2001*). Although we would expect our results to reflect the general pattern of brain network changes during the rise and fall of sustained pain, it remains an empirical question how much they will be generalizable across different clinical pain conditions. Interestingly, a recent fMRI study on the temporal summation of pain in fibromyalgia patients reported results similar to ours (*Cheng et al., 2022*), including the intra-network dissociation within the somatomotor network and the inter-network integration between the somatomotor and other networks during pain. Although we cannot directly examine whether the temporal summation of pain gave rise to these network-level changes due to the limitation of our experimental paradigm, these consistent findings between the two studies may suggest that our findings could be generalized to clinical conditions and that delineating the role of temporal summation of pain would be a promising future direction.

Note that Study 1 used a pain avoidance measure that is not yet fully validated in healthy participants. However, we chose to use the pain avoidance measure, which can provide integrative information on the multidimensional aspects of pain (*Melzack, 1999*; *Waddell et al., 1993*). It also has a clinical implication considering that the maladaptive associations of pain avoidance to innocuous environments have been suggested as a putative mechanism of transition to chronic pain (*Vlaeyen and Linton, 2012*). Lastly, the avoidance measure can provide a common scale across different modalities of aversive experience, allowing us to compare their distinct brain representations (*Ceko et al., 2022*) or test the specificity of their predictive models (*Lee et al., 2021*; *Appendix 1—figure 13*). Although the psychometric properties of the pain avoidance measure should be a topic of future investigation, we expect that the pain avoidance measure would have a high level of convergent validity with pain intensity given the observed similarity between pain avoidance (Study 1) and pain intensity (Study 2) in their temporal profiles. The generalizability of our PCR model across studies 1 and 2 also supports this speculation. However, there would also be situations in which pain avoidance is dissociated from pain intensity. For example, capsaicin can be experienced to be intense but less aversive or even appetitive in some contexts, such as cravings for spicy food (*Stevenson and Yeomans, 1993*). In addition, the gradual rise in avoidance ratings during the late period of the control condition in Study 1 would not be observed if the intensity measure was used. Future studies need to examine the relationship between pain avoidance and the other pain assessments and the advantage of using the pain avoidance measure.

This study also had some limitations. First, with the current experimental paradigm, it is difficult to dissociate the pain duration from the level of pain because the delivery of oral capsaicin commonly induces initial bursting and then a gradual decrease of pain over time. Though we aimed to model the effects of pain duration and pain avoidance ratings with our two primary analyses, that is, consensus community detection and predictive modeling, we cannot fully dissociate the impact of time duration versus pain ratings. Second, although the prediction performances of our predictive models were significant, their effect sizes were moderate (SVM model: 81% accuracy; PCR model: mean $r$ = 0.32). These moderate levels of model performance may be due to information loss during a few of

the multilayer community detection analysis steps, for example, thresholding and binarization of the connectivity matrices. For example, a previous predictive model of sustained pain based on unthresholded functional connectivity values showed better performances (*Lee et al., 2021*). However, the previous model provides a limited level of mechanistic understanding because of the high dimensionality of the model and its features. In addition, functional connectivity itself provides only limited insight into how functional brain networks are structured and reconfigured over time. Considering that the mechanistic understanding of brain responses to sustained pain is the main focus of this study, our moderate levels of model performance may be acceptable, but future studies can still investigate how to improve the predictive performance of the predictive models based on time-varying network features. Third, the choice of two parameters for the multilayer community detection algorithm (i.e., intra- and inter-layer coupling parameters, $\gamma$ and $\omega$) can affect the overall results (*Mucha et al., 2010*). Here, we followed the conventional choice ($\gamma = 1$ and $\omega = 1$) from a previous study (*Bassett et al., 2015*), but the effects of these parameters on our results remain unclear. We used the predictive modeling approach and tested the model generalizability on an independent dataset to avoid excessive reverse-inference from these parameter-sensitive results alone. However, future studies will have to assess the effects of parameter choices on the overall results in a systematic way. Lastly, the optimization of the network density threshold to maximize the differences in a priori network attributes between the capsaicin versus control conditions may influence the results comparing the community structures between the two conditions. However, our main results were based on temporal dynamics across all time-bins, whereas the global-level network attributes that we used to determine the level of threshold were based on the averaged data across all time-bins. In addition, our SVM and PCR models were generalized to the independent test dataset, which was not used to determine the threshold level. Therefore, the confounding effect of thresholding on our main results should be minimal.

Overall, this study contributes to a deeper understanding of how multiple brain systems dynamically interact in response to pain, paving the way for developing novel brain-based interventions for sustained pain. Although further studies are needed to show that our findings can be generalized to clinically relevant sustained pain conditions, we believe that this study provides new insights into the reconfiguration of functional brain networks for pain and a novel framework to investigate the neural mechanisms of sustained pain from a dynamic network perspective.

# Materials and methods

## Participants

Study 1 (discovery) dataset included 48 healthy, right-handed participants (age = 22.8 ± 2.4 [mean ± SD], 21 females) after we excluded 4 participants who provided avoidance rating scores higher in the control run than the capsaicin run and 1 participant whose brain coverage of MRI was insufficient. This dataset was included in a previous publication as an independent test dataset (as Study 3 dataset) (*Lee et al., 2021*). Study 2 (independent test) dataset included 74 healthy, right-handed participants (age = 22.1 ± 2.4 [mean ± SD], 34 females). All participants were recruited from the Suwon area in South Korea. The institutional review board of Sungkyunkwan University approved the study (IRB 2017-05-001). All participants provided written informed consent. The preliminary eligibility of the participants was determined through an online questionnaire. Participants with psychiatric, neurological, or systemic disorders and MRI contraindications were excluded.

## Capsaicin stimulation

In Study 1, we applied capsaicin-containing hot sauce (a food ingredient, Capsaicin Hot Sauce from Jinmifood, Inc) to the participants' tongues to induce tonic pain with minimal risk. We did not include pre-calibration to match the subjective level of pain. To deliver the hot sauce inside the scanner, we dropped a small amount of hot sauce (0.1 ml) onto filter paper (2 cm × 6.5 cm). We spread the hot sauce in a circular format (diameter = 1 cm) on the upper 1/3 of the filter paper. While the participants were lying in the scanner, we handed the filter paper to the participants. The participants carefully placed the capsaicin side of the paper on their tongue (the participants had an opportunity to practice this procedure with a paper without capsaicin inside of the scanner). We then asked them to close their mouths. After 30 s, we asked them to open their mouths and put the paper on the towel on their chest. We then asked participants to keep opening their mouths and breathing only through the

mouth for 1 min to prevent the capsaicin liquid from flowing into the throat. In this way, the liquid dried up, and the capsaicin settled down on a specific area of the tongue. After 1 min, we asked the participants to close their mouths (and keep closing their mouths) while starting the fMRI scan. The participants provided their ratings using an MR-compatible trackball device when a rating scale appeared on the screen. We used this particular procedure for the following reasons: (i) it reduces the risk of coughing in the scanner, (ii) can keep the pain within a tolerable range while maximizing the pain intensity, and (iii) simplifies the delivery method without additional equipment.

In Study 2, we used an almost similar capsaicin delivery procedure, but there were also some differences. First, we used a smaller amount of hot sauce (0.05 ml) because the scan duration of the independent test dataset was shorter (approximately half) than the duration of the discovery dataset. Second, capsaicin was delivered in the middle of the scan. More specifically, participants held the filter paper with their hands for the first 43 s (13 s for additional removal of volumes for image stabilization) since the start of the scan, and then placed the filter paper on their tongue and closed their mouths for 20 s. The participants then removed the filter paper, opened their mouths for 20 s, and then closed their mouths. The total duration of sustained pain was approximately 10 min and 20 s.

## Aversive tastant stimulation

To induce aversive, but not painful, oral sensation for the specificity testing, we used quinine that has a bitter taste in Study 1. A small amount of quinine sulfate (50 mg) was dissolved in distilled water (0.1 ml), which was sufficient to induce aversive and bitter taste. The quinine solution was subsequently dropped onto a filter paper, and the rest of this procedure was same as the 'Capsaicin stimulation' above.

## Aversive odor stimulation

We used fermented skate as an additional aversive stimulus for specificity testing in Study 1. The fermented skate (Hongeo) is a food in South Korea famous for its bad smell. We chose to use the fermented skate among many other options (including multiple strong-smell cheese, Maroilles Fauquet, Bons Mayennais Lingot, Gorgonzolla Piccante) based on aversiveness ratings in a pilot study (n = 15). We attached a slice of fermented skate, covered with a filter paper, to the interior nasal part of a mask. The mask was designed to cover the nose and the mouth. For the delivery, we first moved the bed out of the scanner and unlocked and lifted the head coil. While the participants were breathing through their mouth, we placed the mask to cover participants' nose and mouth. After the head coil was installed again, the participants were re-entered into the scanner. We asked the participants to start breathing through the nose after we started the scanning. We instructed the participants to breathe only through the nose until the end of the scan.

## Experimental design

In Study 1, there were four experimental conditions: (i) capsaicin, (ii) bitter taste, (iii) aversive odor, and (iv) control. After a structural scan, we administered the four condition runs, and the order of the conditions was counterbalanced across participants. Each run lasted for 20 min, and participants provided the avoidance ratings continuously throughout the run. We designed the experiment to have long scans to capture the full rise and fall of each sensation. For the rating scale, we used a modified version of the gLMS (*Bartoshuk et al., 2004*): the anchors began with 'Not at all' [0] to the far left of the scale and continued to the right in a graded fashion with anchors of 'A little bit' [0.061], 'Moderately' [0.172], 'Strongly' [0.354], and 'Very strongly' [0.533], until 'Most (I never want to experience this again in my life)' [1] on the far right. To prevent participants from falling asleep and to help maintain a certain level of alertness during the scan, we used an intermittent simple response task, in which the color of the rating bar on the screen was changed from orange to red for 1 s every minute with a jitter, and the participants had to respond to the color change by clicking the button on the trackball device. During the preprocessing of the data, we included additional regressors of the color changes and button clicks to remove confounding effects related to the task. After the scan, we asked the participants multiple post-scan questions regarding their thought contents during the scan, which were not included in this study.

Study 2 had three experimental conditions: (i) capsaicin, (ii) control, and (iii) phasic heat stimulation. We used only the capsaicin and control runs. The control and capsaicin runs were conducted at the

start and the end of the session, respectively. The control run lasted for 6 min and 13 s, and the capsaicin run lasted for 11 min and 43 s. After the first 13 s (13 s data were discarded before further analyses), participants provided intensity ratings continuously throughout the run using a trackball. After the scan, we asked the participants multiple post-scan questions regarding their thought contents during the scan.

## fMRI data acquisition

Whole-brain fMRI data were acquired using a 3T Siemens Prisma scanner at Sungkyunkwan University. High-resolution T1-weighted structural images were also acquired. Functional EPI images were acquired with TR = 460 ms, TE = 29.0 ms, multiband acceleration factor = 8, field of view = 248 mm, 82 × 82 matrix, 3 × 3 × 3 mm$^3$ voxels, 56 interleaved slices. Stimulus presentation and behavioral data acquisition were controlled using MATLAB (MathWorks) and Psychtoolbox (http://psychtoolbox.org/).

## fMRI data preprocessing

Structural and functional MRI data were preprocessed using our in-house preprocessing pipeline (https://github.com/cocoanlab/surface_preprocessing, *Lee and Woo, 2020*) based on AFNI, FSL, and Freesurfer. This is similar to the Human Connectome Project (HCP) preprocessing pipeline (*Glasser et al., 2013*). For structural T1-weighted images, magnetic field bias was corrected, and non-brain tissues were removed using Freesurfer. Nonlinear transformation parameters projecting native T1 space to MNI 2 × 2 × 2 mm$^3$ template space were also calculated using FSL. For functional EPI images, the initial volumes (22 images [10 s] for Study 1, 29 images [13 s] for Study 2) of fMRI images were removed to allow for image intensity stabilization. Then, the images were motion-corrected using AFNI and distortion-corrected using FSL. These EPI images were co-registered to T1-weighted images using the boundary-based registration (BBR) technique (*Greve and Fischl, 2009*) that used FSL for the first registration and Freesurfer for refinement, similar to the HCP pipeline. We then removed motion-related signals from co-registered EPI images using ICA-AROMA (*Pruim et al., 2015*). Additional preprocessing modules, including (i) removal of nuisance covariates, (ii) linear de-trending, and (iii) low-pass filtering at 0.1 Hz, were combined and conducted in one step using the 3dTproject function in AFNI to avoid introducing unwanted artifacts (*Lindquist et al., 2019*). We included mean BOLD signals from white matter (WM) and cerebrospinal fluid (CSF) (*Pruim et al., 2015*), and timepoints where intermittent arousal maintenance tasks appeared (total of 20 times) as nuisance covariates. For computational efficiency in further analyses, including community detection, these de-noised EPI images were resampled to 4 × 4 × 4 mm$^3$ spatial resolution and masked with an individually defined gray matter boundary image. This gray matter mask obtained using Freesurfer was dilated and eroded five times to create smooth edges, and then resampled to 4 × 4 × 4 mm$^3$ spatial resolution to match the spatial dimension of EPI images.

## Functional connectivity calculation and proportional thresholding

Whole-brain voxel-wise functional connectivity was computed using Pearson's correlation. To determine the optimal threshold level, we tested multiple network density thresholding options (0.01, 0.05, 0.10, 0.20, 0.30, and 0.40) and compared the global-level network characteristics between the capsaicin versus control conditions. Five network attributes, including assortativity, transitivity, characteristic path, global efficiency, and modularity, were used for comparison. We calculated the network attributes using the Brain Connectivity Toolbox (https://sites.google.com/site/bctnet/; *Rubinov and Sporns, 2010*).

   A brief description of these measures is as follows: (i) *Assortativity:* This attribute is related to how often each node (here, a voxel) is connected to the other nodes that have a similar number of links (here, functional connectivity) (*Newman, 2002*). It can be measured using Pearson's correlation between the degrees of every pair of connected nodes. High assortativity means that there are mutual connections between high-degree hub nodes, reflecting the overall resilience of a network. The 'assortativity_bin' function of the toolbox was used. (ii) *Transitivity:* This attribute is related to how often the two nodes that are connected to the same node are also connected to each other (*Newman, 2003*) and measured as the ratio of the number of interconnected triplets of nodes to the number of all triplets of nodes. High transitivity indicates that nodes of a network are more likely to be clustered together. The 'transitivity_bu' function is used. (iii) *Characteristic path:* This attribute is the average

of the shortest path between all pairs of connected nodes (estimated using 'distance_bin' function), reflecting the functional dissociation of a network. A high characteristic path length suggests that the network is in a disintegrated state. (iv) *Global efficiency:* This attribute is the average of the inverse of the shortest path between all pairs of connected nodes (estimated from 'distance_bin' function), reflecting the functional integration of a network. A high global efficiency suggests that a network is in an integrated state. (v) *Modularity:* This attribute is related to how much a network can be clearly divided into a set of modular structures. A network with fewer between-module connections and more within-module connections has a higher level of modularity. The Louvain community detection algorithm (*Blondel et al., 2008*) was used to determine the inherent community structure of a network ('community_louvain' function).

Using these global-level network attributes, we calculated the group-level z-statistics of the differences between conditions for different network density levels. Then, we selected the density level that maximized the sum of absolute z-scores as the optimal threshold level. As shown in *Appendix 1—figure 4*, we selected a network density of 0.05 (i.e., only the top 5% connections were survived) as the optimal threshold level.

## Dynamic functional connectivity (moving time window)

Whole-brain voxel-wise connectivity was computed for every 2 min nonoverlapping time window within individuals using Pearson's correlation. Proportional thresholding at a predetermined network density of 0.05 was applied to each of these dynamic connectivity matrices (Study 1 dataset: 10 matrices per run and participant; Study 2 dataset: five matrices per participant for the capsaicin run and three matrices per participant for the control run), without binarization of connectivity values.

## Multilayer community detection

To identify the time-evolving network community structure, we used the multilayer community detection approach (*Mucha et al., 2010*), which is a generalized version of the Louvain algorithm (*Blondel et al., 2008*). This method finds the optimal community structure that maximizes the 'multilayer modularity' function $Q$, which is defined as

$$Q = \frac{1}{2\mu} \sum_{ijlr} \left\{ \left( A_{ijl} - \gamma_l P_{ijl} \right) \delta_{lr} + \delta_{ij}\omega_{jlr} \right\} \delta \left( g_{il}, g_{jr} \right) \ ,$$

where $A_{ijl}$ is the adjacency matrix of layer $l$ (i.e., dynamic connectivity matrices), $P_{ijl}$ is the optimization null model of layer $l$ (Newman–Girvan null model $P_{ij} = \frac{k_i k_j}{2m}$ , where $k_i = \sum_j A_{ij}$ and $m = \frac{1}{2}\sum_{ij} A_{ij}$), $\gamma_l$ is the intra-layer resolution parameter of layer $l$, $\omega_{jlr}$ is the inter-layer coupling parameter between node $j$ in layer $l$ and node $j$ in layer $r$, $\mu$ is the total sum of edge weights $\mu = \frac{1}{2}\sum_{jr} \left( k_{jr} + \sum_l \omega_{jlr} \right)$, the Kronecker delta $\delta \left( g_{il}, g_{jr} \right)$ is equal to 1 when the community assignment of node $i$ in layer $l$ ($g_{il}$) is the same as the community assignment of node $j$ in layer $r$ ($g_{jr}$), and equal to 0 otherwise. Here, we set all the $\gamma_l$ and $\omega_{jlr}$ parameters to 1 as in previous studies (*Bassett et al., 2015*).

This modularity-maximizing community detection method is inherently nondeterministic (NP-hard problem). Therefore, we iterated this procedure 100 times to determine the consensus community structure, as described in previous studies (*Bassett et al., 2013*; *Bassett et al., 2015*). Details of the within-individual consensus community detection procedure are as follows:

i.    Module allegiance, which is defined as a matrix $T$ whose element $T_{ij}$ is equal to 1 if nodes $i$ and node $j$ are assigned to the same community and 0 otherwise, was calculated for each iteration of multilayer community detection.
ii.   The module allegiance matrices from (i) were averaged across all iterations.
iii.  Null models of module allegiance matrices were obtained by repeating (i) and (ii) with a random permutation of the original community assignments. Permutation was performed once per each of the original community assignments.
iv.   The averaged allegiance matrix from (ii) was thresholded at the maximum value of the null-model allegiance matrix from (iii).
v.    The Louvain community detection algorithm was applied to the thresholded allegiance matrix with 100 iterations. If all the 100 community assignment vectors were identical, the community assignment was determined to be the consensus community. Otherwise, steps (i)–(v) were repeated, using the 100 community assignment vectors as the input of (i).

Using this within-individual consensus community detection procedure, we obtained a consensus community for each layer (i.e., 2 min time window), run, and participant.

## Projecting individuals' community assignments into MNI space

For analyses requiring group-level inferences, we computed one-to-one mapping between the voxels in an individual's native space and the voxels in the MNI space as follows: First, every voxel within the resampled gray matter mask ($4 \times 4 \times 4$ mm$^3$ native space; obtained from the preprocessing steps) was labeled with unique indices. Next, the labeled gray matter mask was resampled to $2 \times 2 \times 2$ mm$^3$ native space, projected to the MNI space using the pre-computed parameters for nonlinear transformation, and then resampled to $4 \times 4 \times 4$ mm$^3$ space, with the nearest neighbor interpolation method. The output contains voxel labels indicating the original location in the native space, which then can be used as a mapping rule for projecting community assignments defined in the native space onto the MNI space.

## Group-level consensus community detection (Figures 2 and 3)

To examine the group-level consensus community structure, we performed the following consensus community detection procedure:

i.   The individual-level consensus community assignments in the native space were projected onto the MNI space using the pre-computed one-to-one mapping rule.
ii.  The projected module community assignments were converted into module allegiance matrices.
iii. The module allegiance matrices were averaged across all individuals and across time depending on the type of consensus community. For the analyses on the capsaicin versus control conditions (as in *Figure 2*), allegiance values were averaged across all time-bins for each run. For the analyses on the early/middle/late periods (as in *Figure 3*), the allegiance values of the capsaicin run were averaged within three separate time-bins; that is, averaging 1–3, 4–7, and 8–10 layers into early, middle, and late-period module allegiance matrices, respectively.
iv.  Null models of module allegiance matrices were obtained by repeating (i)–(iii) with a random permutation of the original community assignments. Permutation was performed once per each of the original community assignments.
v.   The averaged allegiance matrices from (iii) were thresholded at the maximum value of the null-model allegiance matrix from (iv).
vi.  The Louvain community detection algorithm was applied to the thresholded allegiance matrix with 100 iterations. If all the 100 community assignment vectors were identical, the community assignment was determined to be the consensus community. Otherwise, steps (i)–(vi) were repeated using the 100 community assignment vectors as the input of (i).

We excluded voxels that were disconnected after the iterative consensus community detection procedure or that were assigned to a community with a small number of voxels, that is, <20 voxels.

## Permutation tests for regional differences in community structures

To test the statistical significance of the voxel-level difference of consensus community structures (*Figures 2 and 3*), we performed the following Phi-test (*Alexander-Bloch et al., 2012*; *Lerman-Sinkoff and Barch, 2016*). First, for each given voxel, we compared the community label of the voxel to the community label of all the voxels, generating a list of voxel-seed module allegiance values that allow quantitative comparison of voxel-level community profile (e.g., [1, 0, 1, 1, 0, 0,…], whose element is equal to 1 if the seed and target voxels were assigned to the same community and 0 otherwise). Next, a correlation coefficient was calculated between the module allegiance values of the two different brain community structures (i.e., capsaicin versus control, and early versus late). This correlation coefficient is an estimate of the regional similarity of community profiles (here, the correlation coefficient is Phi coefficient because module allegiance is a binary variable). To estimate the statistical significance of the Phi coefficient, we performed permutation tests, in which we randomly shuffled the labels and then obtained the group-level consensus community structures from the shuffled data. Then, the Phi coefficient between the module allegiance values of the two shuffled consensus community structures was calculated. We repeated this procedure 1000 times to generate the null distribution of the Phi coefficient for each voxel. Lastly, we examined the probability to observe a smaller Phi coefficient (i.e., a more dissimilar community profile) than the one from the non-shuffled original data, which

corresponds to the p-value of the permutation test. All the p-values were one-tailed as the hypothesis of this permutation test is unidirectional.

## Module allegiance-based predictive modeling (Figures 4 and 5)

To quantify the relative contribution of network communities to sustained pain experience, we conducted predictive modeling using module allegiance values as input features. We first projected the whole-brain atlas onto each individual's native brain space. The whole-brain atlas originally consisted of 265 regions, but we had to exclude two cerebellar regions (vermis VIIb and X) because these regions had a small number of voxels (23 and 42 in $2 \times 2 \times 2$ mm$^3$ space, respectively) and thus were not successfully transformed to native space in a few participants, resulting in a total of 263 regions. This atlas included 200 cortical regions from the Schaefer atlas (*Schaefer et al., 2018*), 61 subcortical and cerebellar regions from the Brainnetome atlas (*Fan et al., 2016*), and the periaqueductal gray and brainstem regions used in previous studies (*Beissner et al., 2014*; *Roy et al., 2014*). Then, voxel-wise module allegiance matrices were obtained from the individual's consensus community structure and grouped and averaged into $263 \times 263$ region-wise allegiance matrices.

For the classification problem, as shown in *Figure 4*, we averaged 10 time-varying region-wise module allegiance matrices for each run (capsaicin and control runs). Then we trained an SVM classifier to determine whether a participant was in pain or not (i.e., capsaicin versus control conditions) using region-level allegiance matrices across participants (i.e., 34,453 allegiances × 2 runs × 48 participants). A regularization hyperparameter C was set to 1, which is a conventional choice for SVM. To obtain unbiased estimates of classification performance, we used leave-one-participant-out cross-validation on the training dataset (Study 1) and tested the model on an independent test dataset (Study 2). To quantify the model performance, we conducted a forced two-choice classification test, which directly compared the predicted values (here, distances from the hyperplane) of two conditions for each individual. This test did not require the assumption that all participants' brain responses to stimuli are on the same scale (*Wager et al., 2013*).

For regression-based modeling, as shown in *Figure 5*, we trained a PCR model (*Hastie et al., 2009*) to predict pain avoidance ratings (10 time-bins × 48 participants) based on concatenated module allegiance matrices of capsaicin run across 10 time-bins and participants (i.e., 34,453 allegiances × 10 time-bins × 48 participants). We selected 14 PCs for the regression modeling because the PC number yielded the best leave-one-participant-out cross-validated predictive performance on the training dataset (Study 1) (*Appendix 1—figure 10*). To overcome the potential bias in the performance estimation due to the optimal selection of PC number, we additionally conducted nested cross-validation that has double loops of leave-one-participant-out cross-validation; the inner loop where the hyperparameter (i.e., the PC number) was selected, and the outer loop where the actual prediction was done using the hyperparameters chosen from the inner loop. Since the hyperparameter tuning and testing were separated into the inner and outer loops, this procedure provides a less biased estimate of prediction performance even in the training dataset (Study 1). We further tested the prediction model on an independent test dataset (Study 2).

To identify important features for the classification and prediction models, we used bootstrap tests. We randomly sampled 48 participants 10,000 times with replacement and conducted model training with the resampled data. We calculated the statistical significance of predictive weights using z-statistics based on 10,000 samples of 34,453 predictive weights.

## Seed-based allegiance analysis (Figures 4F and 5D and E)

Using the voxel-wise module allegiance in the native space, we calculated the individual's seed-based module allegiance map, which consisted of averaged module allegiance values between voxels within a seed region and the rest of the brain. This seed-based allegiance map was then transformed into MNI space using the pre-computed one-to-one mapping between the native and MNI spaces and averaged across participants. We then conducted a paired *t*-test between the capsaicin versus control conditions for the classification analysis (*Figure 4F*) and between the late versus early period of pain for the regression analysis (*Figure 5D and E*).

## Model response calculation (Figure 6)

To test the allegiance-based classification and prediction models, we calculated a model response score (the intensity of pattern expression) using a dot product of vectorized functional connectivity with model weights:

$$\text{Model response} = \vec{w} \cdot \vec{x} = \sum_{i=1}^{n} w_i x_i$$

where $n$ is the number of connections within the connectivity-based predictive models, $w$ is the connection-level predictive weights, and $x$ is the test data.

## Statistical analysis

In *Figures 4D and 6A*, we used binomial tests for significance testing of whether forced two-choice classification accuracies were significantly higher than the probability of chance (here, 50%). The sample sizes for *Figures 4D and 6A* were n = 48 and 74, respectively. In *Figures 5B and 6B*, we conducted bootstrap tests with 10,000 iterations to test whether the sampling distribution of the within-individual prediction–outcome correlations was significantly higher than zero. Note that we performed the *r*-to-*z* transformation before the bootstrap tests. The sample sizes for *Figures 5B and 6B* were n = 48 and 74, respectively. In *Figures 4E and 5C*, we used bootstrap tests (with 10,000 iterations) to threshold each model's predictive weights. In *Figures 4F and 5D and E*, we used paired *t*-tests either between the seed-based allegiance maps of the capsaicin versus control conditions (*Figure 4F*) or between the seed-based allegiance maps of the late versus the early period of pain (*Figure 5D and E*). Further details of the statistical analyses are provided in each relevant description in the article.

## Data availability

All the data that were used to generate the main figures are available at https://github.com/cocoanlab/brain_reconfig_pain, (copy archived at swh:1:rev:077a65b3d3905182a207349919697e550226fbe5, *Lee, 2022b*). The data that were not used in the main figures will be shared upon request.

## Code availability

The code for generating the main figures is available at https://github.com/cocoanlab/brain_reconfig_pain. In-house MATLAB codes for fMRI data analyses are available at https://github.com/canlab/CanlabCore, (copy archived at swh:1:rev:8d22b1b51ce3696ecd81c3f614e972791ea23df5, *Sun, 2022*) and https://github.com/cocoanlab/cocoanCORE, (copy archived at swh:1:rev:cdcb8a60a65e6d-fb8edbd536862d760b54dd39a4, *Lee, 2022a*).

## Acknowledgements

We thank Hongji Kim and Soo Ahn Lee for their help with conducting experiments. This work was supported by IBS-R015-D1 (Institute for Basic Science; to C-WW), 2019R1C1C1004512, 2021M3E5D2A01022515, and 2021M3A9E4080780 (National Research Foundation of Korea; to C-WW), 2E31511-22-090 (KIST Institutional Program; to C-WW), and by 2018H1A2A1059844 (National Research Foundation of Korea; to J-JL).

## Additional information

### Funding

| Funder | Grant reference number | Author |
| --- | --- | --- |
| Institute for Basic Science | IBS-R015-D1 | Choong-Wan Woo |
| National Research Foundation of Korea | 2019R1C1C1004512 | Choong-Wan Woo |
| National Research Foundation of Korea | 2021M3E5D2A01022515 | Choong-Wan Woo |

| Funder | Grant reference number | Author |
|---|---|---|
| National Research Foundation of Korea | 2021M3A9E4080780 | Choong-Wan Woo |
| Korea Institute of Science and Technology | 2E31511-22-090 | Choong-Wan Woo |
| National Research Foundation of Korea | 2018H1A2A1059844 | Jae-Joong Lee |

The funders had no role in study design, data collection and interpretation, or the decision to submit the work for publication.

#### Author contributions
Jae-Joong Lee, Conceptualization, Resources, Data curation, Software, Formal analysis, Funding acquisition, Validation, Investigation, Visualization, Methodology, Writing - original draft, Project administration, Writing - review and editing; Sungwoo Lee, Dong Hee Lee, Resources, Data curation, Writing - review and editing; Choong-Wan Woo, Conceptualization, Resources, Data curation, Software, Formal analysis, Supervision, Funding acquisition, Validation, Investigation, Visualization, Methodology, Project administration, Writing - review and editing

#### Author ORCIDs
Jae-Joong Lee (iD) http://orcid.org/0000-0002-7353-8683
Choong-Wan Woo (iD) http://orcid.org/0000-0002-7423-5422

#### Ethics
All participants were recruited from the Suwon area in South Korea. The institutional review board of Sungkyunkwan University approved the study (IRB 2017-05-001). All participants provided written informed consent.

#### Decision letter and Author response
Decision letter https://doi.org/10.7554/eLife.74463.sa1
Author response https://doi.org/10.7554/eLife.74463.sa2

## Additional files

#### Supplementary files
• Transparent reporting form

#### Data availability
All the data that were used to generate the main figures are available at https://github.com/cocoanlab/brain_reconfig_pain (copy archived at swh:1:rev:077a65b3d3905182a207349919697e550226fbe5).

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

# Appendix 1

## Results

### Specificity analysis (*Appendix 1—figures 11–13*)

To examine whether the predictive models (i.e., SVM and PCR models) were specific to pain and not influenced by confounding noises, we conducted additional specificity analysis assessing the independence of the models from head movement and physiology variables and specificity of our models to pain versus nonpainful aversive conditions (i.e., bitter taste and aversive odor) in Study 1.

First, we examined the overall changes of framewise displacement (FD) (*Power et al., 2012*), heart rate (HR), and respiratory rate (RR) in sustained pain (*Appendix 1—figure 11*). For the univariate comparison between capsaicin versus control conditions (*Appendix 1—figure 11A*), the results showed that, as expected, capsaicin condition caused significant changes in motion and autonomic responses. The mean FD and HR were significantly higher, and the RR was lower in the capsaicin condition compared to the control condition (FD: $t_{47}$ = 5.30, p=$2.98 \times 10^{-6}$; HR: $t_{43}$ = 4.98, p=$1.10 \times 10^{-5}$; RR: $t_{43}$ = −1.91, p=0.063, paired *t*-test). For the temporal changes of movement and physiology variables (*Appendix 1—figure 11B*), the results showed that the increased motion and autonomic responses are more prominent in the early period of pain. The 10-binned (2 min per time-chunk) FD and HR showed decreasing trend while the RR showed increasing trend over time in capsaicin condition. Additional univariate comparisons between early (1–3 bins, 0–6 min) versus late (8–10 bins, 14–20 min) period of capsaicin condition showed that differences were significant for FD and HR (FD: $t_{47}$ = 6.45, p=$8.12 \times 10^{-8}$; HR: $t_{43}$ = 6.52, p=$6.41 \times 10^{-8}$; RR: $t_{43}$ = −1.61, p=0.11, paired *t*-test). This suggests that while participants were experiencing tonic pain, particularly in the early period, motion and heart rate was increased but breathing was slowed. Note that we needed to exclude four participants' data due to technical issues with physiological data acquisition.

Next, we examined whether the head movement and physiological responses are the main driver of our predictive models (*Appendix 1—figure 12*). For all the original model responses from SVM model (2 conditions × 44 participants = 88), we regressed out the mean FD, HR, and RR (concatenated across conditions and participants as the SVM model was trained) and calculated the classification accuracy (*Appendix 1—figure 12A*). Although the model responses were controlled for movement and physiology variables, the SVM model still showed a high classification accuracy for the capsaicin versus control conditions in a forced-choice test (n = 44, accuracy = 89%, p=$1.41 \times 10^{-7}$, binomial test, two-tailed). Similarly, for all the original model responses from PCR model (10 time-bins × 44 participants = 440), we regressed out the 10-binned FD, HR, and RR (concatenated across time-bins and participants as the PCR model was trained) and calculated the within-individual prediction–outcome correlation (*Appendix 1—figure 12B*). Again, the PCR model showed a significantly high predictive performance (n = 44, mean prediction–outcome correlation $r$ = 0.20, p=0.003, bootstrap test, two-tailed, mean squared error = 0.159 ± 0.022 [mean ± SEM]) while controlling for movement and physiology variables. These results suggest that our SVM and PCR models capture unique variance in tonic pain above and beyond the head movement and physiological changes.

Lastly, we examined the specificity of our predictive models to pain by testing the models onto the nonpainful but tonic aversive conditions including bitter taste (induced by quinine) and aversive odor (induced by fermented skate) (*Appendix 1—figure 13*). All the model responses were obtained using leave-one-participant-out cross-validation. The results showed that the overall model responses of SVM model for bitter taste and aversive odor conditions were higher than those for control conditions, but lower than capsaicin condition (*Appendix 1—figure 13A*). Classification accuracy between capsaicin versus bitter taste and versus aversive odor was all significantly high (capsaicin versus bitter taste: accuracy = 79%, p=$6.17 \times 10^{-5}$, binomial test, two-tailed, *Appendix 1—figure 13C*; capsaicin versus aversive odor: accuracy = 83%, p=$3.31 \times 10^{-6}$, binomial test, two-tailed, *Appendix 1—figure 13E*), suggesting the specificity of SVM model to pain. Similarly, the temporal trajectories of the model responses of PCR model for bitter taste and aversive odor conditions were not overlapping with that of the capsaicin condition (*Appendix 1—figure 13B*). Furthermore, the model responses of bitter taste and aversive odor conditions do not have significant relationship with the actual avoidance ratings (bitter taste: mean prediction–outcome correlation $r$ = 0.05, p=0.41, bootstrap test, two-tailed, mean squared error = 0.036 ± 0.006 [mean ± SEM], *Appendix 1—figure 13D*; aversive odor: mean prediction–outcome correlation $r$ = 0.12, p=0.06, bootstrap test, two-

tailed, mean squared error = 0.044 ± 0.004 [mean ± SEM], *Appendix 1—figure 13F*), suggesting the specificity of PCR model to pain.

Overall, we have provided evidence that the module allegiance-based models can predict pain ratings above and beyond the movement and physiological changes, and are more responsive to pain compared to nonpainful aversive conditions, which suggest the specificity of our results to pain.

## Between-individual predictive performances

For SVM model, we evaluated the classification accuracy for the capsaicin versus control conditions across all the participants, instead of the forced-choice test that compared the two conditions within individuals. The results for Study 1 were as follows: accuracy with an optimal threshold = 88% (p=1.82 × 10$^{-14}$, binomial test, two-tailed), 85% sensitivity, 90% specificity, area under the curve (AUC) = 0.94. The results for Study 2 were as follows: accuracy with an optimal threshold = 76% (p=2.84 × 10$^{-10}$, binomial test, two-tailed), 72% sensitivity, 80% specificity, AUC = 0.80.

For PCR model, we calculated the correlation between mean pain ratings and mean model responses (i.e., between-individual prediction–outcome correlation) for capsaicin condition. The results for Study 1 were as follows: $r = 0.41$, p=0.004, one-sample $t$-test, two-tailed, mean squared error = 0.024. The results for Study 2 were as follows: $r = 0.27$, p=0.018, one-sample $t$-test, two-tailed, mean squared error = 0.022.

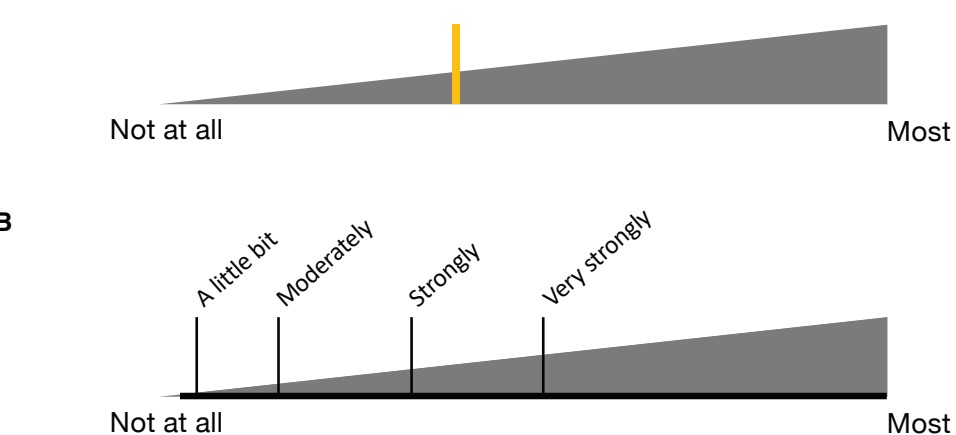

**A**   Please continuously report how much do you want to avoid this experience in the future.

Not at all                                                                 Most

**B**

A little bit   Moderately   Strongly   Very strongly

Not at all                                                                 Most

- **A little bit:** "I will try a little bit to avoid this experience."
- **Moderately:** "I want to avoid this experience in the future."
- **Strongly:** "I strongly want to avoid this experience in the future."
- **Very strongly:** "I very strongly want to avoid this experience in the future. (i.e., It's unbearable)"

**Appendix 1—figure 1.** Avoidance rating in Study 1. (**A**) Participants reported their subjective ratings of avoidance continuously throughout the scan by moving the yellow-colored rating bar on the rating scale (gray triangle). The rating question was "how much do you want to avoid this experience in the future?" During the scan, we showed only the two extreme descriptors (i.e., 'Not at all' and 'Most') on the screen. (**B**) To enhance the reliability, we provided a detailed explanation of the locations and meaning of the anchors before the scan. The anchors included 'Not at all,' 'A little bit,' 'Moderately,' 'Strongly,' 'Very strongly,' and 'Most.' These anchors were not displayed during the scan.

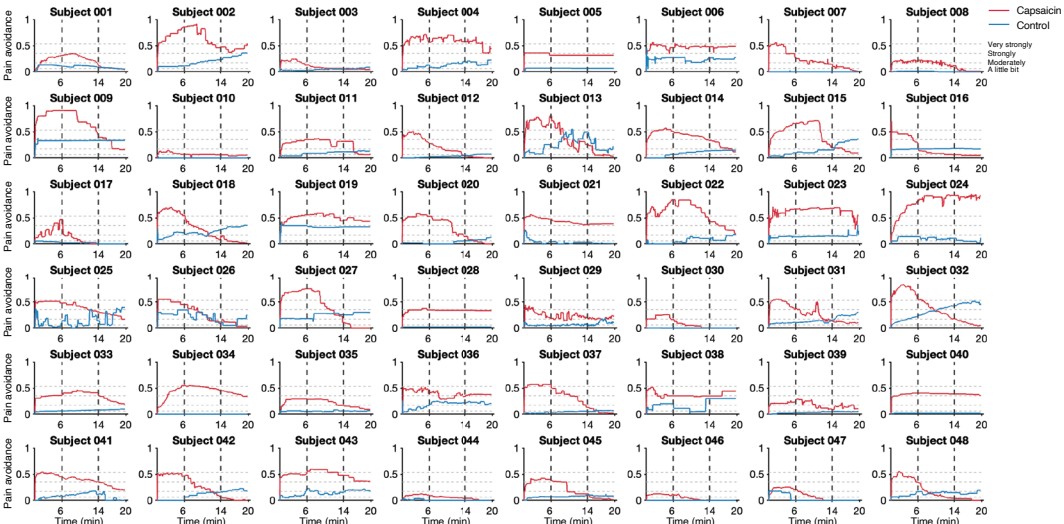

**Appendix 1—figure 2.** Pain avoidance ratings of individual participants. These plots show the pain avoidance ratings of 48 participants from Study 1. The horizontal dashed lines indicate the anchors of the general labeled magnitude scale (gLMS). The vertical dashed lines show the time boundaries for the early, middle, and late periods of sustained pain. The red and blue solid lines show the pain avoidance ratings (red for the capsaicin run and blue for the control run).

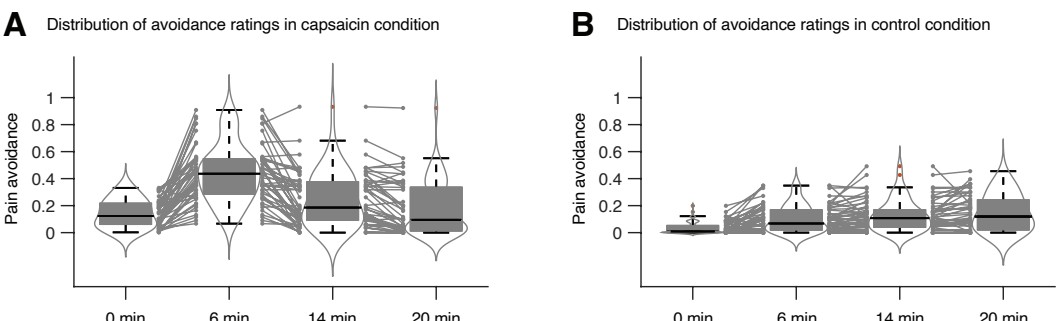

**Appendix 1—figure 3.** Distribution of avoidance ratings in Study 1. We used the violin and box plots to show the distributions of the avoidance ratings, for the timepoints of 0 min, 6 min, 14 min, and 20 min. For each timepoint, 10 s time window was applied for averaging the ratings. The box was bounded by the first and third quartiles, and the whiskers stretched to the greatest and lowest values within median ± 1.5 interquartile range. The red dots outside of the whiskers were marked as outliers. Each gray line between gray dots represents each individual participant's paired data.

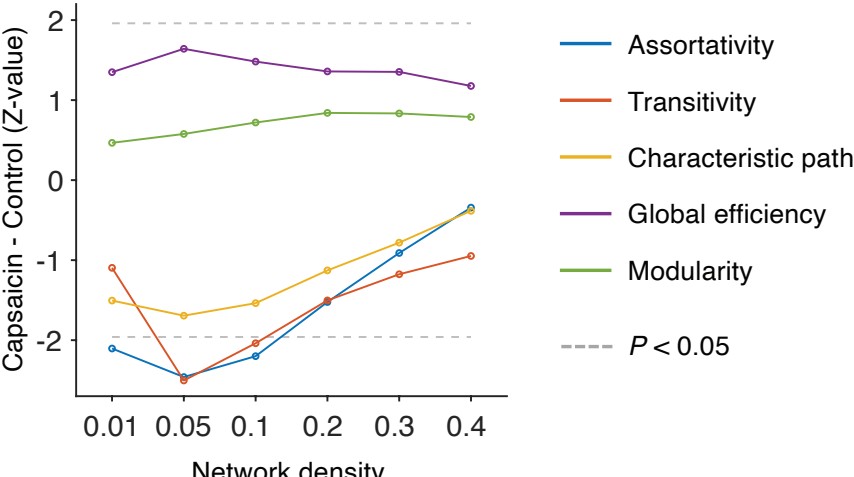

**Appendix 1—figure 4.** Global-level network attributes. We compared the five global-level network attributes (i.e., assortativity, transitivity, characteristics path, global efficiency, and modularity) of the capsaicin and control conditions (z-values from paired z-test) across different levels of network density. Note that the overall differences of network attributes between the capsaicin versus control conditions were maximal at the network density of 0.05. All network attributes were measured from the binarized static connectivity matrices (for details, see 'Materials and methods').

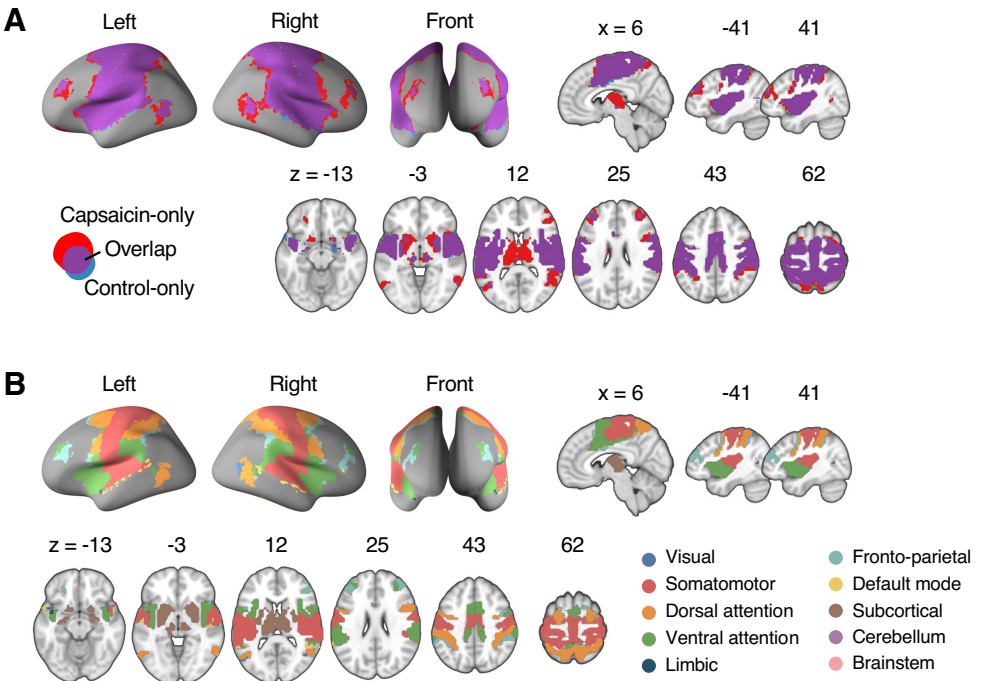

**Appendix 1—figure 5.** Reconfiguration of the consensus Community 2. (**A**) We compared the spatial distributions of the consensus Community 2 (i.e., the somatomotor dominant community) of the capsaicin and control conditions. The purple regions show the overlap between the two conditions, and the red and blue regions show the unique regions for the capsaicin and control conditions, respectively. Blue region was comparatively small, indicating that the somatomotor community mainly expanded primarily during the capsaicin condition compared to the control condition. (**B**) The spatial distribution of the 10 canonical brain networks within the consensus Community 2. The expansion of Community 2 during the capsaicin condition (red regions in [**A**]) was mainly driven by the brain voxels within the frontoparietal network (e.g., dorsolateral prefrontal cortex and inferior parietal cortex) and the subcortical regions (e.g., thalamus and basal ganglia).

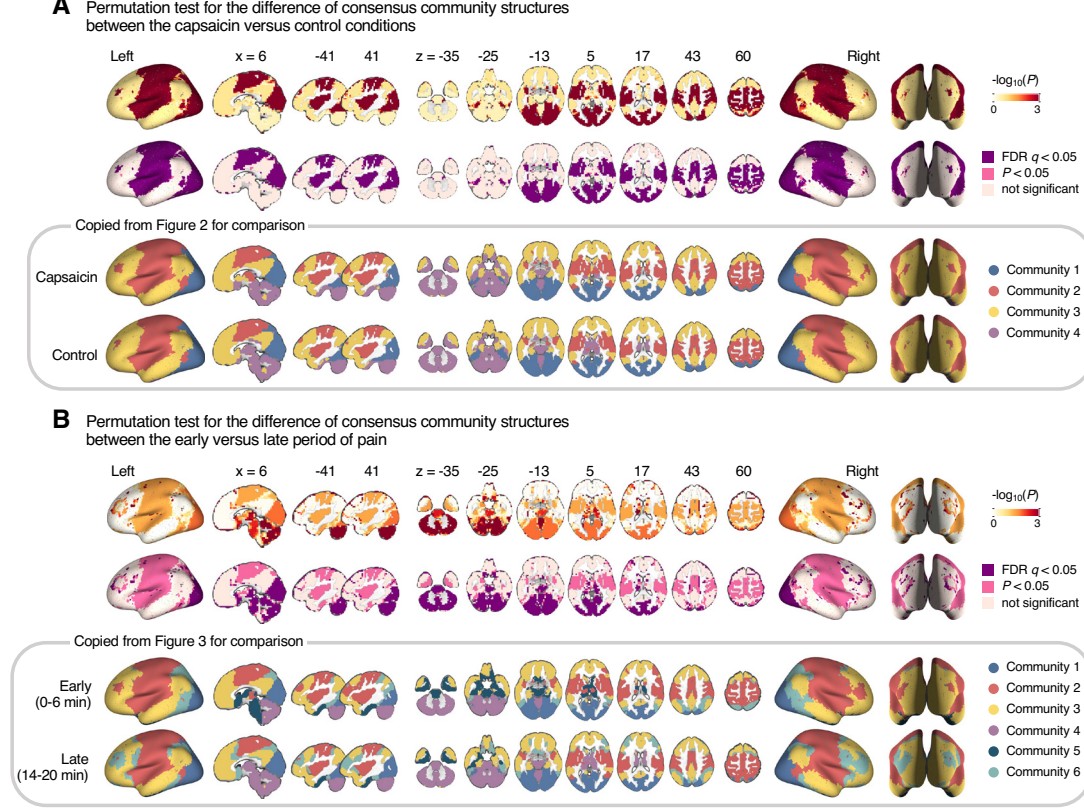

**Appendix 1—figure 6.** Permutation tests for the community structure changes. We examined the regional differences in community structures between (**A**) the capsaicin versus control or (**B**) the early versus late period of pain. To quantitatively compare the group-level consensus community structures between the capsaicin versus control conditions and the early versus late periods, we first obtained a seed-based module allegiance map for each voxel (i.e., using each voxel as a seed). Then, we calculated a correlation coefficient of the module allegiance values between two different conditions for each voxel. This correlation coefficient can serve as an estimate of the voxel-level similarity of the consensus community profile. Because module allegiance is a binary variable, these correlation values are Phi coefficients ($\varphi$). To calculate the statistical significance of the Phi coefficient, we conducted permutation tests, in which we randomly shuffled the condition labels in each participant and obtained the group-level consensus community structure for each shuffled condition. Then, we calculated the voxel-level correlations of the module allegiance values between the two shuffled conditions. We repeated this procedure 1000 times to generate the null distribution of the Phi coefficients, and calculated the proportion of null samples that have a smaller Phi coefficient (i.e., a more dissimilar regional community structure) than the nonshuffled original data, which is the p-value, one-tailed. First row: $-\log_{10}p$ values from the permutation tests. Second row: thresholded maps based on the p-values from the permutation tests. Colored as purple for false discovery rate (FDR)-corrected $q < 0.05$, pink for uncorrected $p<0.05$, and pale pink for nonsignificance. Third and last row: Group-level consensus community structures from **Figures 2 and 3**.

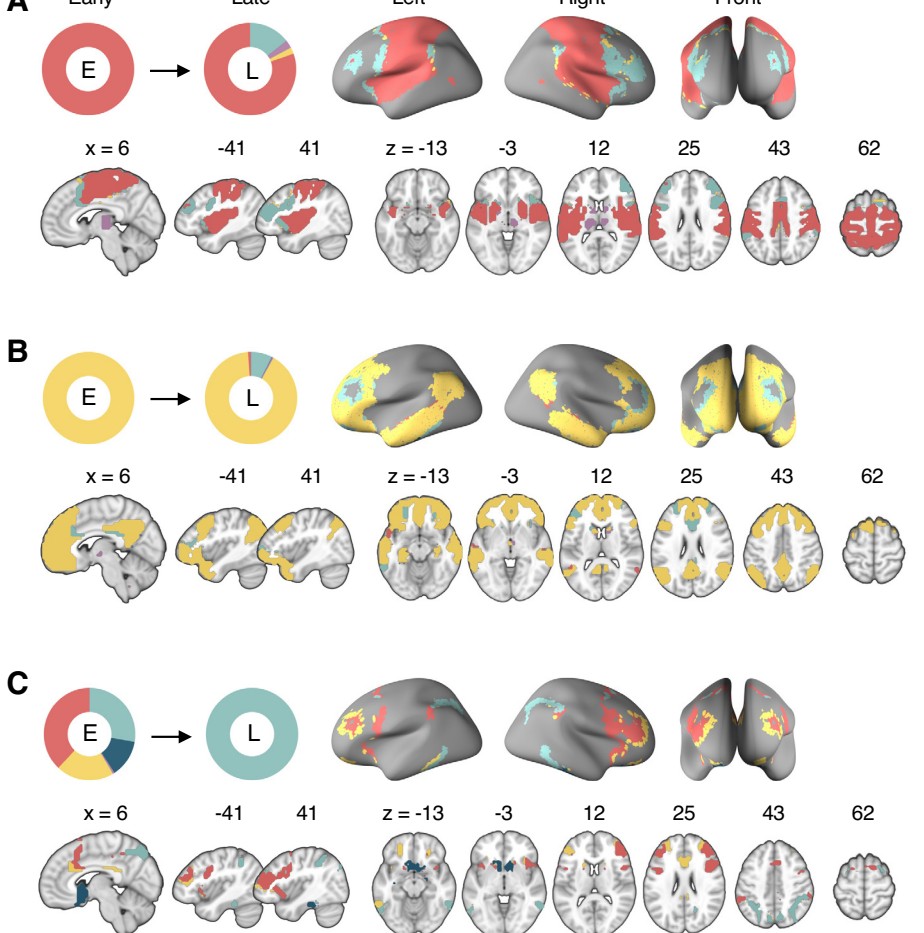

**Appendix 1—figure 7.** Reconfiguration of the consensus communities 2, 3, and 6. The reconfiguration pattern of the community assignments of the brain voxels that were assigned to (**A**) the consensus Community 2 (somatomotor network dominant community) in the early period of sustained pain, (**B**) the consensus Community 3 (default-mode network dominant community) in the early period of sustained pain, and (**C**) the consensus Community 6 (frontoparietal network dominant community) in the late period of sustained pain. E, early; L, late.

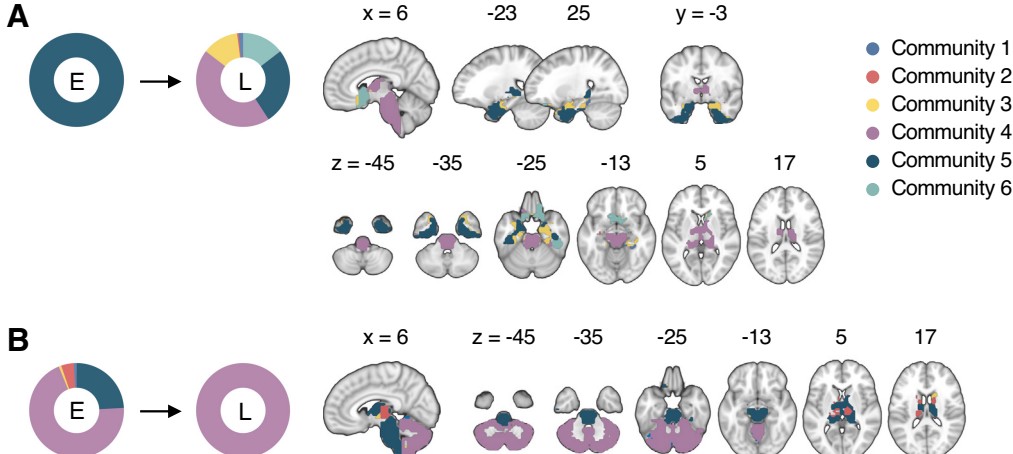

**Appendix 1—figure 8.** Reconfiguration of the consensus communities 4 and 5. The reconfiguration pattern of the community assignments of the brain voxels that were assigned to (**A**) the consensus Community 5 (limbic network dominant community) in the early period of sustained pain, and (**B**) the consensus Community 4 (cerebellum-dominant community) in the late period of sustained pain. E, early; L, late.

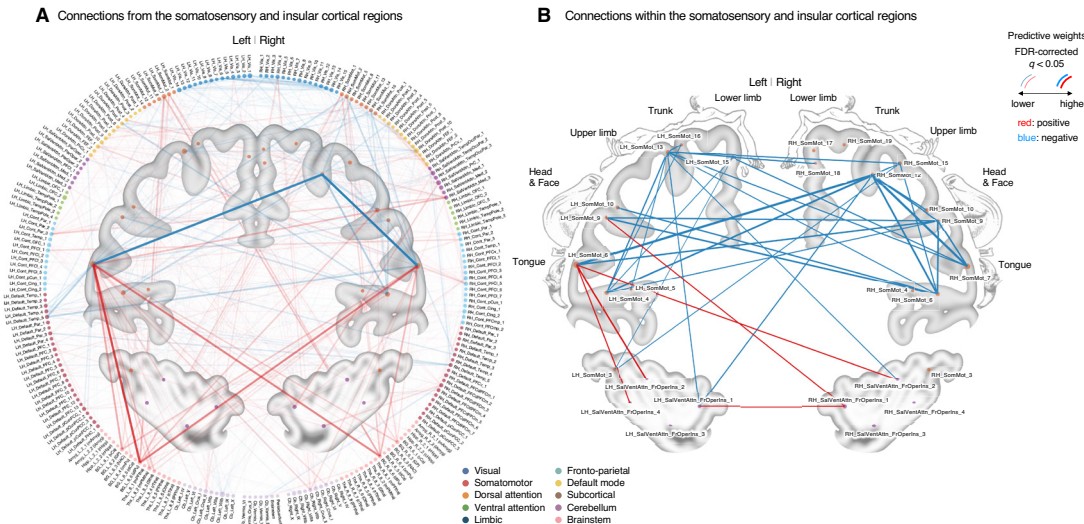

**Appendix 1—figure 9.** Classifier weights of the somatosensory and insular cortical regions. (**A**) Thresholded connections (false discovery rate [FDR] $q < 0.05$, which corresponds to uncorrected p<0.003, two-tailed, bootstrap test with 10,000 iterations) showing the predictive weights between the somatosensory and insular cortical regions and the other remaining whole brain regions. Line thickness and transparency indicate the absolute magnitude of predictive weights. There were strong negative weights within the somatosensory cortical regions and strong positive weights between the tongue primary somatosensory regions and subcortical regions (e.g., basal ganglia). (**B**) Thresholded connections (FDR $q < 0.05$, which corresponds to uncorrected p<0.003, two-tailed, bootstrap test with 10,000 iterations) showing the predictive weights between the primary and secondary somatosensory and insular cortical regions. Line thickness indicates the absolute magnitude of the predictive weights. Note that the location of the nodes on the brain map may not reflect the exact center coordinates of the regions-of-interests, though we marked them on the closest locations on the map. The somatosensory homunculus (modified from Fig 17 in **Penfield and Rasmussen, 1950**, p.44) represents the overall somatotopic gradients.

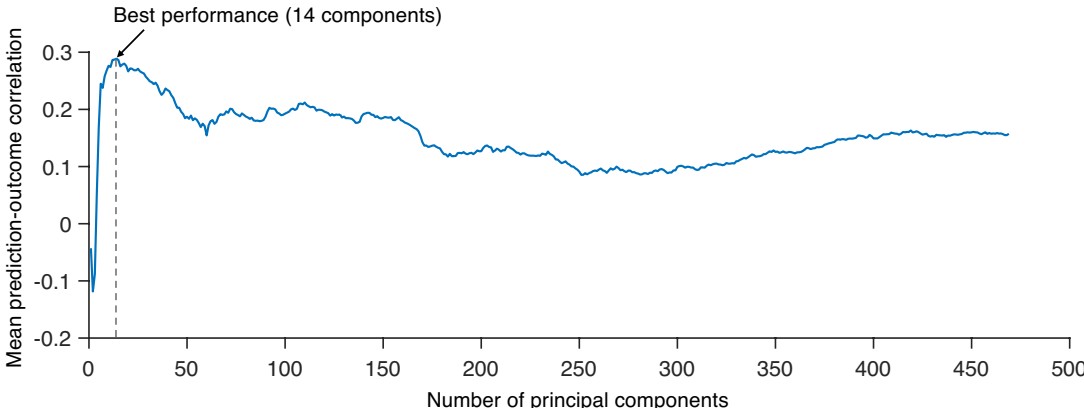

**Appendix 1—figure 10.** Predictive performances across different numbers of principal components (PCs). We tested principal component regression (PCR) with a different number of PCs to find the best model to predict the within-individual variation of sustained pain ratings and calculated the mean correlation between actual and predicted pain ratings (10 ratings). Predictive performances were based on leave-one-subject-out cross-validation. The best model used 14 PCs for prediction.

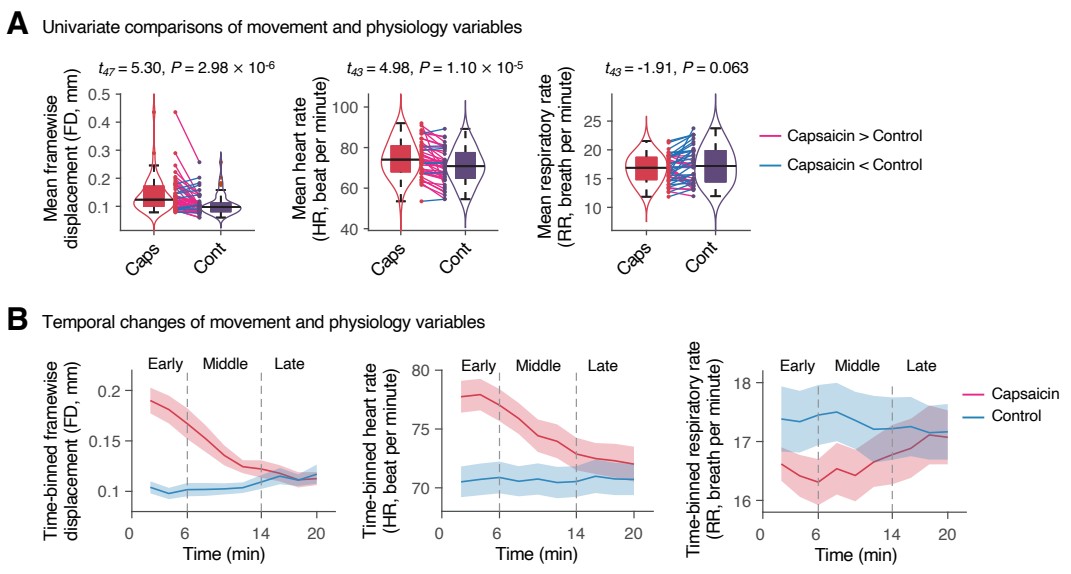

**Appendix 1—figure 11.** Head movement and physiology variables in Study 1. (**A**) Comparisons of head motion (framewise displacement) and physiological measures (heart and respiratory rate) between the capsaicin versus control conditions. For significance testing, we conducted paired *t*-tests. (**B**) Temporal changes of head motion and physiology variables. Data were divided into 10 time-bins (2 min per bin). The vertical dashed lines show how we define the early, middle, and late periods of sustained pain. The solid lines represent group mean ratings (red for capsaicin, and blue for control), and the shading represents standard errors of the mean (SEM). Note that we needed to exclude four participants' data due to technical issues with the physiological data acquisition.

**A** Capsaicin vs. Control using SVM model with regressing out movement and physiological variables

**B** Predicting avoidance ratings using PCR model with regressing out movement and physiological variables

**Appendix 1—figure 12.** Prediction performance after controlling for head motion and physiological variables. To examine whether head motion and physiology responses influenced the predictive model performance, we regressed out the framewise displacement (FD), heart rate (HR), and respiratory rate (RR) from the cross-validated model predictions in Study 1 (n = 44). (**A**) A forced-choice test to compare the model responses of the support vector machine (SVM) model for the capsaicin versus control conditions after regressing out the mean FD, HR, and RR. For regression, all the conditions and participants were concatenated as we trained the original SVM model. Left: receiver-operating characteristics (ROC) curve. Right: model responses for different conditions. Each line connecting dots represents an individual participant's paired data (red: correct classification; blue: incorrect classification). p-Value was based on a binomial test, two-tailed. (**B**) Actual versus predicted values of the principal component regression (PCR) model after regressing out the 10-binned (2 min per bin) FD, HR, and RR. For regression, all the time-bins and participants were concatenated as we trained the original PCR model. Each colored line (and symbol) represents individual participant's ratings during the capsaicin run (red: higher *r*; yellow: lower *r*; blue: *r* < 0). p-Value was based on bootstrap tests, two-tailed.

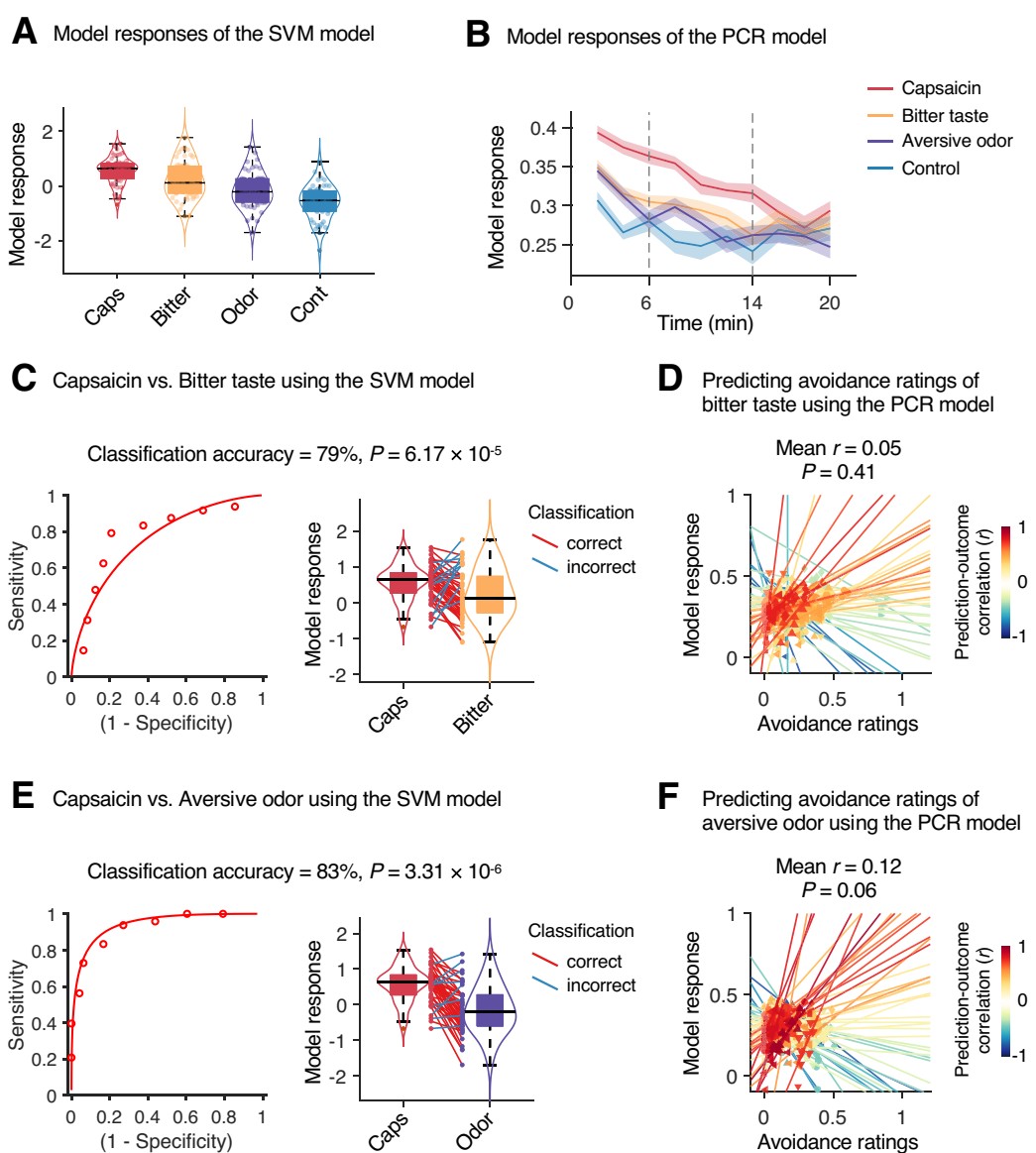

**Appendix 1—figure 13.** Specificity test results. We tested the module allegiance-based support vector machine (SVM) and principal component regression (PCR) models of pain on the bitter taste (quinine) or aversive odor (fermented skate) conditions in Study 1 (n = 48) to examine the specificity of the models, using leave-one-participant-out cross-validation. (**A, B**) Model responses of the SVM model (**A**) and the PCR model (**B**) for capsaicin, bitter taste, aversive odor, and control conditions. The solid lines and shading in (**B**) represent the group mean ratings and standard errors of the mean (SEM), respectively. (**C, E**) Forced-choice test results of comparing the model responses for the capsaicin versus bitter taste (**C**) or aversive odor (**E**) conditions. Left: receiver-operating characteristics (ROC) curve. Right: model responses for different conditions. Each line connecting dots represents an individual participant's paired data (red: correct classification; blue: incorrect classification). p-Values were based on a binomial test, two-tailed. (**D, F**) Actual versus predicted ratings. Each colored line (and symbol) represents individual participant's ratings during the bitter taste run (**D**) or aversive odor run (**F**) (red: higher $r$; yellow: lower $r$; blue: $r < 0$). p-Value was based on bootstrap tests, two-tailed.

**Appendix 1—table 1.** Top 50 stable connections of the classification model.

| Rank | Weights | ROI names | MNI coordinates |
|---|---|---|---|
| *Positive connections* | | | |
| #1 | 0.0222 | LH_SomMot_6 - BG_L_6_6 (dlPu) | (−56,–8,30) - (−28,–6,2) |

*Appendix 1—table 1 Continued on next page*

*Appendix 1—table 1 Continued*

| Rank | Weights | ROI names | MNI coordinates |
|------|---------|-----------|-----------------|
| #2 | 0.0200 | LH_SomMot_6 - BG_R_6_6 (dlPu) | (−56,−8,30) - (30,−4,2) |
| #3 | 0.0200 | RH_SomMot_7 - BG_L_6_6 (dlPu) | (58,−4,30) - (−28,−6,2) |
| #4 | 0.0181 | LH_SomMot_6 - BG_L_6_2 (GP) | (−56,−8,30) - (−22,−2,4) |
| #5 | 0.0170 | RH_SomMot_7 - BG_R_6_6 (dlPu) | (58,−4,30) - (30,−4,2) |
| #6 | 0.0162 | LH_SomMot_7 - BG_R_6_6 (dlPu) | (−48,−8,46) - (30,−4,2) |
| #7 | 0.0161 | RH_SomMot_7 - BG_L_6_2 (GP) | (58,−4,30) - (−22,−2,4) |
| #8 | 0.0160 | LH_SomMot_7 - BG_L_6_6 (dlPu) | (−48,−8,46) - (−28,−6,2) |
| #9 | 0.0156 | RH_DorsAttn_Post_9 - RH_SalVentAttn_TempOccPar_2 | (8,-56,62) - (60,−38,16) |
| #10 | 0.0154 | LH_SomMot_6 - BG_R_6_2 (GP) | (−56,−8,30) - (22,−2,4) |
| #11 | 0.0154 | RH_SalVentAttn_PrC_1 - RH_SalVentAttn_FrOperIns_3 | (50,4,40) - (36,24,4) |
| #12 | 0.0154 | LH_Cont_Par_3 - RH_SalVentAttn_Med_3 | (−46,−42,46) - (8,4,66) |
| #13 | 0.0153 | LH_SomMot_2 - BG_R_6_6 (dlPu) | (−52,−24,10) - (30,−4,2) |
| #14 | 0.0161 | LH_SomMot_6 - Cb_Right_VIIb | (−56,−8,30) - (−24,−58,−52) |
| #15 | 0.0160 | RH_SomMot_1 - BG_L_6_6 (dlPu) | (52,−14,6) - (−28,−6,2) |
| #16 | 0.0156 | LH_DorsAttn_FEF_1 - RH_Cont_PFCl_5 | (−32,−4,54) - (30,48,28) |
| #17 | 0.0154 | LH_SomMot_6 - Tha_L_8_2 (mPMtha) | (−56,−8,30) - (−18,−14,4) |
| #18 | 0.0154 | LH_SomMot_6 - Cb_Right_VI | (−56,−8,30) - (24,−58,−26) |
| #19 | 0.0153 | RH_SomMot_1 - BG_R_6_6 (dlPu) | (52,−14,6) - (30,−4,2) |
| #20 | 0.0139 | RH_SomMot_2 - BG_R_6_6 (dlPu) | (64,−24,8) - (30,−4,2) |
| #21 | 0.0133 | LH_SomMot_2 - BG_R_6_2 (GP) | (−52,−24,10) - (22,−2,4) |
| #22 | 0.0131 | RH_SomMot_2 - BG_R_6_2 (GP) | (64,−24,8) - (22,−2,4) |
| #23 | 0.0130 | RH_SomMot_7 - BG_R_6_2 (GP) | (58,−4,30) - (22,−2,4) |
| #24 | 0.0127 | LH_SomMot_7 - BG_L_6_2 (GP) | (−48,−8,46) - (−22,−2,4) |
| #25 | 0.0127 | LH_SomMot_6 - Cb_Left_VI | (−56,−8,30) - (−22,−58,−24) |
| #26 | 0.0125 | RH_SomMot_10 - BG_R_6_6 (dlPu) | (46,−12,48) - (30,−4,2) |
| #27 | 0.0118 | LH_SomMot_6 - Cb_Left_VIIb | (−56,−8,30) - (−26,−66,−50) |
| #28 | 0.0113 | LH_SomMot_6 - Tha_R_8_8 (lPFtha) | (−56,−8,30) - (12,−16,6) |
| #29 | 0.0112 | LH_SomMot_7 - BG_R_6_2 (GP) | (−48,−8,46) - (22,−2,4) |
| #30 | 0.0111 | LH_Vis_8 - Tha_L_8_2 (mPMtha) | (−48,−70,10) - (−18,−14,4) |
| #31 | 0.0104 | LH_SomMot_13 - LH_Default_PFC_7 | (−26,−38,68) - (−8,58,20) |
| #32 | 0.0104 | LH_Vis_8 - LH_Cont_Cing_1 | (−48,−70,10) - (−4,−28,26) |
| #33 | 0.0103 | RH_SomMot_2 - Tha_L_8_2 (mPMtha) | (64,−24,8) - (−18,−14,4) |
| #34 | 0.0095 | LH_SomMot_13 - LH_Default_PFC_2 | (−26,−38,68) - (−6,36,−10) |
| #35 | 0.0092 | LH_SomMot_7 - Cb_Right_VI | (−48,−8,46) - (24,−58,−26) |
| #36 | 0.0092 | LH_SomMot_13 - RH_Default_PFCdPFCm_1 | (−26,−38,68) - (4,36,−14) |

*Appendix 1—table 1 Continued on next page*

*Appendix 1—table 1 Continued*

| Rank | Weights | ROI names | MNI coordinates |
|---|---|---|---|
| #37 | 0.0077 | LH_Cont_Cing_1 - RH_Vis_5 | (−4,−28,26) - (48,−72,−6) |
| #38 | 0.0067 | LH_SomMot_13 - LH_Limbic_OFC_2 | (−26,−38,68) - (−10,36,−20) |
| *Negative connections* | | | |
| #1 | −0.0235 | RH_SomMot_7 - RH_SomMot_12 | (58,−4,30) - (40,−24,58) |
| #2 | −0.0231 | LH_SomMot_6 - RH_SomMot_12 | (−56,−8,30) - (40,−24,58) |
| #3 | −0.0160 | LH_SomMot_4 - RH_SomMot_12 | (−54,−4,10) - (40,−24,58) |
| #4 | −0.0145 | RH_SomMot_6 - RH_SomMot_14 | (56,−12,14) - (32,−22,64) |
| #5 | −0.0143 | LH_SomMot_6 - LH_Default_Temp_3 | (−56,−8,30) - (−56,−6,−12) |
| #6 | −0.0141 | LH_SomMot_6 - LH_Default_Temp_4 | (−56,−8,30) - (−58,−30,−4) |
| #7 | −0.0135 | LH_SomMot_6 - LH_Default_pCunPCC_1 | (−56,−8,30) - (−12,−56,14) |
| #8 | −0.0124 | LH_SomMot_4 - LH_SomMot_13 | (−54,−4,10) - (−26,−38,68) |
| #9 | −0.0115 | LH_SomMot_4 - RH_SomMot_14 | (−54,−4,10) - (32,−22,64) |
| #10 | −0.0092 | RH_SalVentAttn_Med_1 - Cb_Left_VIIIb | (8,8,42) - (0,−64,−42) |
| #11 | −0.0084 | LH_SalVentAttn_FrOperIns_4 - Cb_Left_VIIIb | (−52,8,10) - (0,−64,−42) |
| #12 | −0.0083 | LH_SalVentAttn_Med_1 - Cb_Left_VIIIb | (−6,10,42) - (0,−64,−42) |

Top 50 stable connections based on bootstrap tests with 10,000 iterations (edge-level $p<1.9 \times 10^{-7}$, FDR $q < 1.4 \times 10^{-4}$).
FDR, false discovery rate; ROI, region of interest.

**Appendix 1—table 2.** Top 50 stable connections of the regression model.

| Rank | Weights | ROI names | MNI coordinates |
|---|---|---|---|
| *Positive connections* | | | |
| #1 | 0.000496 | LH_SomMot_12 - RH_Default_pCunPCC_3 | (−32,−22,64) - (6,−58,44) |
| #2 | 0.000460 | LH_SomMot_10 - RH_Default_pCunPCC_3 | (−40,−26,58) - (6,−58,44) |
| *Negative connections* | | | |
| #1 | −0.000768 | RH_Cont_Cing_1 - Cb_Left_Crus_II | (6,−26,30) - (−26,−74,−42) |
| #2 | −0.000756 | RH_Cont_Cing_1 - Cb_Left_Crus_I | (6,−26,30) - (−36,−68,−32) |
| #3 | −0.000743 | Tha_R_8_1 (mPFtha) - Cb_Vermis_VI | (8,−10,6) - (0,−70,−22) |
| #4 | −0.000711 | Tha_R_8_1 (mPFtha) - Cb_Vermis_VIIIa | (8,−10,6) - (26,−58,−54) |
| #5 | −0.000697 | RH_Cont_Cing_1 - Cb_Right_Crus_I | (6,−26,30) - (38,−68,−32) |
| #6 | −0.000695 | RH_Cont_Cing_1 - Cb_Vermis_IX | (6,−26,30) - (6,−54,−48) |
| #7 | −0.000694 | Tha_R_8_1 (mPFtha) - Cb_Left_VIIb | (8,−10,6) - (−26,−66,−50) |
| #8 | −0.000673 | RH_Cont_Cing_1 - Cb_Right_Crus_II | (6,−26,30) - (26,−76,−42) |
| #9 | −0.000662 | Tha_R_8_1 (mPFtha) - Cb_Left_Crus_I | (8,−10,6) - (−36,−68,−32) |
| #10 | −0.000655 | Tha_R_8_1 (mPFtha) - Cb_Vermis_IX | (8,−10,6) - (6,−54,−48) |
| #11 | −0.000655 | LH_Cont_Cing_1 - Cb_Left_Crus_I | (−4,−28,26) - (−36,−68,−32) |
| #12 | −0.000654 | RH_Cont_PFCmp_1 - Cb_Left_Crus_I | (8,30,28) - (−36,−68,−32) |

*Appendix 1—table 2 Continued on next page*

*Appendix 1—table 2 Continued*

| Rank | Weights | ROI names | MNI coordinates |
|------|---------|-----------|-----------------|
| #13 | –0.000652 | RH_Cont_PFCmp_1 - Cb_Left_Crus_II | (8,30,28) - (−26,−74,−42) |
| #14 | –0.000650 | Tha_R_8_1 (mPFtha) - Cb_Left_Crus_II | (8,-10,6) - (−26,−74,−42) |
| #15 | –0.000648 | LH_Cont_Cing_1 - Cb_Left_Crus_II | (−4,−28,26) - (−26,−74,−42) |
| #16 | –0.000647 | Tha_L_8_1 (mPFtha) - Cb_Vermis_VI | (−6,−12,6) - (0,−70,−22) |
| #17 | –0.000643 | LH_Cont_Cing_1 - Cb_Right_Crus_I | (−4,−28,26) - (38,−68,−32) |
| #18 | –0.000642 | Tha_R_8_1 (mPFtha) - Cb_Right_VIIb | (8,−10,6) - (−24,−58,−52) |
| #19 | –0.000626 | BG_R_6_1 (vCa) - Cb_Vermis_VI | (14,14,−2) - (0,−70,−22) |
| #20 | –0.000624 | LH_Cont_Cing_1 - Cb_Right_Crus_II | (−4,−28,26) - (26,−76,−42) |
| #21 | –0.000598 | Tha_R_8_1 (mPFtha) - Cb_Right_VI | (8,−10,6) - (24,−58,−26) |
| #22 | –0.000592 | Tha_R_8_1 (mPFtha) - Cb_Right_Crus_I | (8,−10,6) - (38,−68,−32) |
| #23 | –0.000578 | RH_SalVentAttn_FrOperIns_3 - Cb_Left_VIIb | (36,24,4) - (−26,−66,−50) |
| #24 | –0.000565 | LH_SalVentAttn_Med_1 - Cb_Left_VIIb | (−6,10,42) - (−26,−66,−50) |
| #25 | –0.000549 | Tha_L_8_7 (cTtha) - Cb_Left_Crus_I | (−10,−22,14) - (−36,−68,−32) |
| #26 | –0.000543 | Tha_L_8_7 (cTtha) - Cb_Left_VIIb | (−10,−22,14) - (−26,−66,−50) |
| #27 | –0.000538 | BG_L_6_5 (dCa) - Cb_Vermis_VIIIa | (−14,2,16) - (26,−58,-54) |
| #28 | –0.000532 | LH_SalVentAttn_Med_1 - Cb_Right_VIIb | (−6,10,42) - (−24,−58,−52) |
| #29 | –0.000530 | BG_L_6_5 (dCa) - Cb_Vermis_VI | (−14,2,16) - (0,−70,−22) |
| #30 | –0.000530 | Tha_L_8_7 (cTtha) - Cb_Left_Crus_II | (−10,−22,14) - (−26,−74,−42) |
| #31 | –0.000523 | BG_R_6_5 (dCa) - Cb_Left_Crus_I | (14,6,14) - (−36,−68,−32) |
| #32 | –0.000520 | BG_R_6_5 (dCa) - Cb_Left_VIIb | (14,6,14) - (−26,−66,−50) |
| #33 | –0.000519 | Tha_L_8_1 (mPFtha) - Cb_Left_Crus_I | (−6,−12,6) - (−36,−68,−32) |
| #34 | –0.000515 | BG_R_6_5 (dCa) - Cb_Left_Crus_II | (14,6,14) - (−26,−74,−42) |
| #35 | –0.000494 | BG_L_6_5 (dCa) - Cb_Left_VIIb | (−14,2,16) - (−26,−66,−50) |
| #36 | –0.000481 | BG_L_6_4 (vmPu) - Cb_Right_VI | (−22,6,−4) - (24,−58,-26) |
| #37 | –0.000477 | Tha_R_8_4 (rTtha) - Cb_Left_Crus_II | (2,−12,6) - (−26,−74,−42) |
| #38 | –0.000473 | Tha_R_8_4 (rTtha) - Cb_Left_Crus_I | (2,−12,6) - (−36,−68,−32) |
| #39 | –0.000470 | BG_L_6_5 (dCa) - Cb_Left_Crus_I | (−14,2,16) - (−36,−68,−32) |
| #40 | –0.000469 | BG_R_6_5 (dCa) - Cb_Right_Crus_I | (14,6,14) - (38,−68,−32) |
| #41 | –0.000454 | BG_R_6_5 (dCa) - Cb_Right_VIIb | (14,6,14) - (−24,−58,−52) |
| #42 | –0.000448 | BG_L_6_5 (dCa) - Cb_Left_Crus_II | (−14,2,16) - (−26,−74,−42) |
| #43 | –0.000444 | RH_Cont_PFCv_1 - Cb_Left_Crus_II | (34,22,−8) - (−26,−74,−42) |
| #44 | –0.000443 | BG_L_6_5 (dCa) - Cb_Right_VIIb | (−14,2,16) - (−24,−58,−52) |
| #45 | –0.000434 | BG_R_6_5 (dCa) - Cb_Right_VI | (14,6,14) - (24,−58,−26) |
| #46 | –0.000421 | BG_L_6_5 (dCa) - Cb_Right_Crus_I | (−14,2,16) - (38,−68,−32) |
| #47 | –0.000405 | BG_L_6_5 (dCa) - Cb_Right_VI | (−14,2,16) - (24,−58,−26) |
| #48 | –0.000393 | BG_L_6_5 (dCa) - Cb_Left_VI | (−14,2,16) - (−22,−58,−24) |

Top 50 stable connections based on bootstrap tests with 10,000 iterations (edge-level $p < 6.1 \times 10^{-5}$, FDR $q < 0.043$).

FDR, false discovery rate; ROI, region of interest.

