## [Editor Report]

This article will be of great interest to researchers interested in the brain mechanisms of pain. It shows how the connectivity of brain networks associated with sustained pain changes over time. These findings are supported by compelling fMRI analyses of a tonic pain paradigm in two cohorts of healthy human participants. These important insights advance the understanding of the brain mechanisms of sustained pain, which is the hallmark of chronic pain as a major healthcare problem.

---

## [Decision Letter]

**Decision letter after peer review:**

Thank you for submitting your article "Dynamic Functional Brain Reconfiguration During Sustained Pain" for consideration by *eLife*. Your article has been reviewed by 3 peer reviewers, including Markus Ploner as the Reviewing Editor and Reviewer #1, and the evaluation has been overseen by a Reviewing Editor and Timothy Behrens as the Senior Editor. The following individual involved in review of your submission has agreed to reveal their identity: Tamas Spisak (Reviewer #3).

Essential revisions:

1) It remains unclear whether the changes of brain networks over time simply reflect the duration of sustained pain or whether they essentially reflect different levels of pain intensity/avoidance. Therefore, analyzing and/or discussing whether brain network changes reflect pain duration or pain intensity would be crucial for the interpretation of the findings.

2) Although the manuscript is very well-written it might benefit from an even clearer and simpler explanation of what the consensus community structure and the underlying module allegiance measure assesses.

3) The added value of the assessment of the dynamics of brain networks remains unclear. Specifically, it is unclear whether the current analysis of brain networks dynamics allows for a clearer distinction between and prediction of pain and no-pain states than other measures of static or dynamic brain activity or static measures of brain connectivity. Therefore, clarifying the added value of the community structure analysis as compared to other more common analyses of brain activity and brain connectivity would significantly strengthen the case.

4) The Authors do not touch upon the concept of temporal summation of pain, historically associated with tonic pain. Please comment on the relationship of the present study to temporal summation particularly since chronic pain patients often exhibit increased temporal summation of pain.

5) Please consider a recent related paper by Cheng et al., Arthritis Rheumatol, 2021 that shares most of the methodological pipeline to highlight similarities and novelties and deepen the comparison with the associated literature.

6) The data analysis is entirely conducted on young healthy subjects. This is not a limitation per se, but the conclusion about offering new insights into understanding mechanisms at the basis of chronic pain is too far from the results. A similar pipeline has been actually applied to chronic pain patients (Cheng et al., Arthritis Rheumatol, 2021, Lee et al., Nat Med. 2021). Discussing the results of the present paper in relationship to those, could offer a more robust way to connect the Authors' results to networks behavior in pathological brains.

7) The behavioral measure used to assess evoked pain perception (avoidance ratings), has been developed for chronic pain patients and never validated on healthy controls. It might not be an appropriate measure considering the total absence of pain variability in the reported responses over forty-eight subjects. Please discuss this important point. Moreover, please address the following questions:

• How does the rating scale look like? Is the meaning of the anchors and of the thresholds for weak, moderate, strong, and very strong displayed on a screed during the scan? What are the instructions given to the participants before the start of the experiment? Do they induce any kind of pain expectancy (e.g., "After 6 minutes the pain should start decreasing" or analogous expressions)? A combination of those elements could explain the lack of pain variability.

• The behavioral outcome is called "pain avoidance" and the Authors hypothesize that it is proportional to the perceived pain. Is there any evidence proving this correlation? The collected ratings are the answer to the question "how much do you want to avoid this experience in the future?". Is it possible that this question is too generic to be called pain avoidance? Did the Authors quantify in any way the effect of laying down in a scanner for long time? It might play a role in the avoidance index.

8) The dynamic measure employed by the Authors is better described from the term "windowed functional connectivity". It is often considered a measure of dynamic functional connectivity and it gives information about fluctuations of the connectivity patterns over time. Nevertheless, the entire focus of the paper, including the title, is on dynamic networks, which inaccurately leads one to think of time-varying measures with higher temporal resolution. This allows one to follow network reorganization over time without averaging 2-min intervals in which several different brain mechanisms might play an important role. In summary, the assumption of constant response throughout 2-min periods of tonic pain and the use of Pearson correlations do not mirror the idea of dynamic analysis expressed by the Authors in title and introduction. Please consider removing "dynamic" from the title, reduce the emphasis on this concept, address possible confounds introduced by the choice of long windows and rephrase the aim of the study in terms of brain network reconfiguration over the main phases of tonic pain experience.

9) Procedure chosen for evoking sustained pain. The measures in figure 1B suggest that the intensity of the painful stimulation is not constant as expected for sustained pain (probably the effect washes out with the saliva). In this case, the first six-minute interval requires particular attention because it encapsulates the real tonic pain phase, and the following ones require more appropriate labels. Ideally the authors should cite previous studies showing that tongue evoked pain elicits a very specific behavioral response (summation, habituation/decrease of pain, absence of pain perception). Moreover, please address the following points:

• Does the procedure include a calibration phase? If yes, please add description in the Methods section. If not, how do the Authors explain the relatively small standard error of the mean reported in Figure 1?

• If possible, add citation proving that there is a very consistent behavioral (pain related) response to the capsaicin: no pain variability against what most of the evoked pain experiments showed.

• Please report in the supplementary material the dots distribution (box plot with visible dots) of the ratings at minute 0, 6, 14, 20.

10) Community detection analysis. Please clarify the following issues:

• The thresholding of the connectivity matrices for the binarization of the networks is certainly a weakness of the study and, more in general, of most of the connectivity analysis (only few estimators have known null distributions). Here, the chosen optimal threshold is the one that maximize the difference between conditions (capsaicin vs controls) in terms of global graph measures, including modularity. I suggest adding a comment on the effect of maximizing a measure depending on the modularity of the networks, on the subsequent community detection algorithm, also based on maximizing modularity (within the network this time). What is the Authors' opinion on this possible confound? Also, what is the rationale/hypothesis behind this procedure for obtaining sparse matrices?

• Did the Authors consider running the analysis with other resolution parameters (γ and omega)?

Group-level consensus community detection: I found this section difficult to follow, especially in terms of reasoning for specific choices and steps of the analysis.

• Step III: the standard definition of allegiance is a binary matrix whose elements are equal to one only when the two correspondent nodes belong to the same community (as the Authors described). After step III, a new index is computed as the mean of allegiance matrices over time and across subjects. Its value indicates the proportion (percentage) of subjects showing the two nodes in the same community at specific time points. Or how many times the two nodes belong to the same community during the early/middle/late stage of the experiment. Using the name allegiance (a binary measure) to indicate those percentages, might be misleading. I suggest using more appropriate names (measures like "agreement", "dwell time" and similar might be useful) and providing more explanatory examples on how to read the value of the computed measures.

• Step IV: please specify number of permutations.

• Step VI: in the same spirit as the two previous comments, I suggest either reconsidering the necessity of this computation, or explaining the reasons for applying a community detection algorithm twice. I believe that additional layers of complexity always require a clear question that they can answer.

11) It remains unclear, how specific the results are to pain. Differences between the control resting state and the capsaicin trials might be – at least partially – driven by other factors, like motion artifacts, saliency, attention, axiety, etc. Differences between stages over the time-course might, additionally, be driven by scanner drifts (to which the applied approach might be less sensitive, but the possibility is still there ) or other gradual processes, e.g. shifts in arousal, attention shifts, alertness, etc. All the above factors might emerge as confounding bias in both of the predictive models. This problem should be thoroughly discussed, and at least the following extra analyses are recommended, in order to attenuate concerns related to the overall specificity and neurobiological validity of the results:

• Reporting of, and testing for motion estimates (mean, max, median framewise displacement or anything similar).

• Examining whether these factors might, at least partially, drive the predictive models.

• e.g. Applying the PCR model on the resting state data and verifying of the predicted timecourse is flat (no inverse U-shape, that is characteristic to all capsaicin trials).

12) Statistical inference. An important issue is the (apparent) lack of statistical inference when analyzing the differences in the group-level consensus community structures (both when comparing capsaicin to control and when analysing changes over the time-course of the capsaicin-challenge). Although the observed changes seem biologically plausible and fit very well to previous results, without proper statistical inference we can't determine, how likely such differences are to emerge just by chance. This makes all results on Figures 2 and 3, and points 1, 4 and 5 in the discussion partially or fully speculative or weakly underpinned, comprising a large proportion of the current version of the manuscript. There are two main ways of handling this issue:

• Enhancing (or clarifying potential misunderstandings regarding) the methodology (see my concrete, and hopefully feasible, suggestions in the "private part" of the review). There are likely many ways to test the significance of these differences. Two permutation testing-based ideas are (i) permuting the labels ctr-capsaicin, or early-mid-late, repeating the analysis, constructing the proper null distribution of e.g. the community size changes and obtain the p-values and (ii) "trace back" communities to the individual level and do (nonparametric) statistical inference there.

• De-weighting the presentation and the discussion of the related results.

*Reviewer #1 (Recommendations for the authors):*

• The authors emphasize the term "pain supersystem". This term is not very well-introduced yet and the necessity for such a term is unclear. I recommend that the authors rely less on this term and omit it at least from the abstract.

• The statement in the abstract "In the early stage, the orofacial areas of the primary somatomotor cortex were separated from the other primary somatomotor cortices and integrated with…" is a bit ambiguous. It might better read "In the early stage, the orofacial areas of the primary somatomotor cortex were separated from other areas of the primary somatomotor cortex and integrated with…"

*Reviewer #2 (Recommendations for the authors):*

• I suggest reducing the amount of text in the figures. All the information needed to understand the illustrations should be included in the captions. Figure 1 and Figure 7 are the ones that require the most attention in this respect.

• It might be a good idea to specify when any previous evidence used to justify the current analysis or to make inferences on the obtained results actually come from the Authors' previous publications. Especially if they are extracted from the same dataset, this information is relevant.

• In their previous paper, the Authors had access to a dataset including the experimental conditions: tonic capsaicin pain, tonic aversive taste, and tonic aversive odor. Did the Authors analyze the communities structure during those controls conditions? Did they consider testing their classifier on them? In my opinion, it would add a lot of robustness to the study findings, and it would make the obtained results reliable and unquestionably pain related (thinking of the more general avoidance ratings).

• In terms of data availability, the Authors declared that data and codes will be shared upon publication. I would appreciate their availability if there will be a second loop of revisions before potential publication.

*Reviewer #3 (Recommendations for the authors):*

– As the authors mention the cross-validated evaluation of the PCR model is biased due to hyperparameter optimization. While the independent evaluation resolves any related concerns, the authors might consider applying a nested cross-validation framework, to have unbiased estimates for the discovery dataset, as well.

– Optimizing the network density threshold in the same dataset, especially on one of the conditions-of-interest (Q1: capsaicin vs. controls) may be circular (as the optimized global network metrics may well be associated to the community structure). On the other hand, this potential circularity does not affect all he results (e.g definitely not the results based on the independent test dataset) and in general, I don't think this would significantly affect the results. Nevertheless, performing (or reproducing) the optimization on independent data would be reassuring. Alternatively, this issue must be discussed as a potential limitation/bias.

– While this is not explicitly stated, prediction performance is evaluated only on the within subject-level. For better comparability to other methods, please report and discuss the "between-subject" estimates, too (i.e. how well can we classify/predict from a single session/window of a single subject).

– Introduction: discussing the possible relation of the present work to chronic pain or other clinical pain conditions is not sufficient.

– More information is needed about the individual variability of the pain-related behavioral time-courses (maybe in the supplementary info). Was remission complete in all participants?

– Some participants might be more tolerant for capsaicin than others, due to eating habits. Please discuss whether this could potentially affect the results.

– At many points, e.g. in paragraph 25 on page x or 5 on page 25, it is mentioned that the models generalized across two datasets. While the terminology is currently heterogenous, I kindly suggest to use the term "generalization" only to the independent test dataset (here the models really had to generalize to scanning parameters, paradigm differences, etc.)

– it's a bit unclear why the pain avoidance ratings fall. One would, somewhat naively, hypothesize that if the participant once though she would never repeat this experiment again, why would she change her mind a couple of minutes later, when the memories of pain are still vivid. Please comment on this.

– Please add a short discussion of the differences of the behavioral ratings and how they might affect the findings. (this might be positive thing: a sign of generalization across behavioral assessment protocols).

– Please clarify why *pain* avoidance (slightly) increased in the control resting state scan.

– Please provide more rationale for the choice of ML algorithms.

– How were the hyperparameters set for the SVM? Why were those not optimized, too?

– Why was only one hub selected for the seed-based analysis in the case of the classifier?

– While the prediction performances are obviously significant, testing for this with bootstrapping may be suboptimal, as bootstrap samples may inherit non-normality from the parent dataset. Permutation test would be more "elegant" in my opinion.

– Discussion: relation to consciousness might be somewhat speculative, should be hedged.

– Will the raw data also be shared?

---

## [Author Response]

Essential revisions:1) It remains unclear whether the changes of brain networks over time simply reflect the duration of sustained pain or whether they essentially reflect different levels of pain intensity/avoidance. Therefore, analyzing and/or discussing whether brain network changes reflect pain duration or pain intensity would be crucial for the interpretation of the findings.

We appreciate the editor and reviewer’s comment on this issue. With the current experimental paradigm, it is difficult to dissociate the pain duration from the level of pain because the delivery of oral capsaicin commonly induces initial bursting and then a gradual decrease of pain over time. That is, the pain duration is correlated with the pain intensity in our task.

However, when we examined the time-course of the ratings at each individual level (as shown in Appendix 1—figure 2 ), the time duration explained 53.7% of the rating variance, R^2^ = 0.537 ± 0.315 (mean ± standard deviation). In addition, if we constrain the β coefficient of the time duration to be negative (i.e., ratings should decrease over time), the explained variance decreases to 48.2%, R^2^ = 0.482 ± 0.457, leaving us enough variance (i.e., greater than 50%) for examining the distinct effects of time duration and ratings on the patterns of functional brain reorganization.

Indeed, the two main analyses included in the manuscript—consensus community detection and predictive modeling—were designed to examine those two aspects of the task, i.e., time duration and pain avoidance ratings, respectively. First, through the consensus community detection analysis, we examined the community structure that changes over time, i.e., across the early, middle, and late periods (as shown in Figure 3). We then developed predictive models of pain avoidance ratings in the second main analysis (as shown in Figure 5).

Though it is still a caveat that we cannot fully dissociate the effects of time duration versus pain ratings, we could interpret the first set of results to be more about time duration, while the second set of results is more about pain ratings.

We now added a description of the implication of predictive modeling for isolating the effects of pain ratings. In addition, a discussion on the caveat of the current experimental design and relevant future direction.

Revisions to the main manuscript:

p. 25:

“Moreover, developing models to directly predict the pain ratings is helpful to complement the group-level analysis, because the changes in consensus community structure over the early, middle, and late periods only indirectly reflect the different levels of pain.”

p. 27:

“This study also had some limitations. First, with the current experimental paradigm, it is difficult to dissociate the pain duration from the level of pain because the delivery of oral capsaicin commonly induces initial bursting and then a gradual decrease of pain over time. Though we aimed to model the effects of pain duration and pain avoidance ratings with our two primary analyses, i.e., consensus community detection and predictive modeling, we cannot fully dissociate the impact of time duration versus pain ratings.”

2) Although the manuscript is very well-written it might benefit from an even clearer and simpler explanation of what the consensus community structure and the underlying module allegiance measure assesses.

We thank you for the suggestion. Now we added additional (but simple) descriptions of module allegiance and consensus community detection methods.

Revisions to the main manuscript:

pp. 8-9:

“Here, the consensus community means the group-level representative structures of the distinct community partitions of individuals. To determine the consensus community across different individuals and times, we first obtained the module allegiance (Bassett et al., 2011) from the community assignment of each individual. Module allegiance assesses how much a pair of nodes is likely to be affiliated with the same community label, and is defined as a matrix T whose element T_ij_ is 1 when nodes i and j are assigned to the same community and 0 when assigned to different communities. This conversion of the categorical community assignments to the continuous module allegiance values allows group-level summarization of different community structures of individuals.”

p. 14:

“Here, high module allegiance indicates the voxels of two regions are likely to be in the same community affiliation, and vice versa.”

3) The added value of the assessment of the dynamics of brain networks remains unclear. Specifically, it is unclear whether the current analysis of brain networks dynamics allows for a clearer distinction between and prediction of pain and no-pain states than other measures of static or dynamic brain activity or static measures of brain connectivity. Therefore, clarifying the added value of the community structure analysis as compared to other more common analyses of brain activity and brain connectivity would significantly strengthen the case.

The main goal (and thus, the added value) of the current study was to provide a “mechanistic” understanding of the brain processes of sustained pain, rather than the “prediction.” Even though we included the results from the predictive modeling, as in Figures 4-6, our focus was more on the interpretation of the model to quantitatively examine the functional changes in the brain, not on the maximization of the prediction performance.

Indeed, maximizing the prediction performance was the main goal of our previous study (Lee et al., 2021), in which we developed a predictive model of sustained pain based on the patterns of dynamic functional connectivity. The model showed better prediction performances compared to the current study, but it was challenging to interpret the model because of the high dimensionality of the model and its features. In addition, functional connectivity itself provides only limited insight into how functional brain networks are structured and reconfigured over time.

In this sense, the multi-layer community detection method has several advantages to achieving our goal. First, the community detection analysis allows us to summarize the complex, high-dimensional whole-brain connectivity patterns into neurobiologically interpretable subsystems. Second, the multi-layer community detection method allows us to study the temporal changes in community structure by connecting the same nodes across different time points.

Now we added a description of the rationale behind the choice of the multi-layer community detection analysis over the conventional functional connectivity methods, and the added value of our study.

Revisions to the main manuscript:

p. 3:

“In this study, we examined the reconfiguration of whole-brain functional networks underlying the natural fluctuation in sustained pain to provide a mechanistic understanding of the brain responses to sustained pain.”

p. 7:

“In this study, we used this approach to examine the temporal changes of brain network structures during sustained pain, which cannot be done with conventional functional connectivity-based analyses (Lee et al., 2021).”

p. 27:

“However, the previous model provides a limited level of mechanistic understanding because of the high dimensionality of the model and its features. In addition, functional connectivity itself provides only limited insight into how functional brain networks are structured and reconfigured over time.”

4) The Authors do not touch upon the concept of temporal summation of pain, historically associated with tonic pain. Please comment on the relationship of the present study to temporal summation particularly since chronic pain patients often exhibit increased temporal summation of pain.

We thank the reviewer and editor for the comment on this important topic. Temporal summation of pain indicates progressively increased sensation of pain during prolonged noxious stimulation (Price, Hu, Dubner, and Gracely, 1977), and has been suggested as a hallmark of chronic pain disorders including fibromyalgia (Cheng et al., 2022; Price et al., 2002). In a recent study by Cheng et al. (2022), the authors induced tonic pain using constantly high cuff pressure and examined whether the participants experienced increased pain in the late period compared to the early period of pain. On the contrary, in our experimental paradigm, the capsaicin liquid initially delivered into the oral cavity is being cleaned out by saliva, and thus overall pain intensity was decreasing over time, not increasing (Figure 1B). Therefore, the temporal summation of pain may occur in a limited period (e.g., the early period of the run), but it is difficult to examine its effect systematically in our study.

However, it is notable that Cheng et al.’s results overlap with our findings. For example, Cheng et al. reported the intra-network segregation within the somatomotor network and the inter-network integration between the somatomotor and other networks during the temporal summation of pressure pain in patients with fibromyalgia, which were similar to the findings we reported in Appendix 1—figure 9 and Figure 4. Although it is unclear whether these results reflect the temporal summation of pain, these network-level features shared across the two studies are likely to be an essential component of the sustained pain processes in the brain.

Now we added a comment on the temporal summation of pain in the main manuscript.

Revisions to the main manuscript (p. 26):

“Interestingly, a recent fMRI study on the temporal summation of pain in fibromyalgia patients reported results similar to ours (Cheng et al., 2022), including the intra-network dissociation within the somatomotor network and the inter-network integration between the somatomotor and other networks during pain. Although we cannot directly examine whether the temporal summation of pain gave rise to these network-level changes due to the limitation of our experimental paradigm, these consistent findings between the two studies may suggest that our findings could be generalized to clinical conditions.”

5) Please consider a recent related paper by Cheng et al., Arthritis Rheumatol, 2021 that shares most of the methodological pipeline to highlight similarities and novelties and deepen the comparison with the associated literature.

We thank the reviewer and editor for the information about this recent publication. Cheng et al. (2022) was not published at the time we wrote the manuscript, and we were surprised that Cheng et al. shares many aspects with our study, e.g., both used multilayer community detection and also reported similar findings, as described above.

However, there were some differences between the two studies as well.

First, the focus of our study was on the brain dynamics during the natural time-course of sustained pain from its initiation to remission in healthy participants, whereas the focus of Cheng et al. was on the temporal summation phenomenon of pain (TSP) and the enhanced TSP in patients with fibromyalgia patients. Because of this difference in the research focuses, our study and Cheng et al. are providing many nonoverlapping results and insights. For example, our study paid particular attention to the coping mechanisms of the brain (e.g., the network-level changes in the subcortical and frontoparietal network regions) and the brain systems that are correlated with the natural decrease of pain (e.g., the cerebellum in Figure 5). In contrast, Cheng et al. (2022) identified the brain connectivity and network features important for the increased TSP in fibromyalgia patients.

Second, our great interest was in identifying and visualizing the fine-grained spatiotemporal patterns of functional brain network changes over the period of sustained pain. To utilize fine-grained brain activity information, we conducted our main analyses at a voxel-level resolution and on the native brain space, such as in Figures 2-3 and Appendix 1—figures 5, 7, and 8. With this fine-grained spatiotemporal mapping, we were able to identify small, but important voxel-level dynamics.

We now cited Cheng et al. (2022) in multiple places and revised the manuscript accordingly.

Revisions to the main manuscript (p. 26):

“Interestingly, a recent fMRI study on the temporal summation of pain in fibromyalgia patients reported results similar to ours (Cheng et al., 2022), including the intra-network dissociation within the somatomotor network and the inter-network integration between the somatomotor and other networks during pain. Although we cannot directly examine whether the temporal summation of pain gave rise to these network-level changes due to the limitation of our experimental paradigm, these consistent findings between the two studies may suggest that our findings could be generalized to clinical conditions.”

6) The data analysis is entirely conducted on young healthy subjects. This is not a limitation per se, but the conclusion about offering new insights into understanding mechanisms at the basis of chronic pain is too far from the results. A similar pipeline has been actually applied to chronic pain patients (Cheng et al., Arthritis Rheumatol, 2021, Lee et al., Nat Med. 2021). Discussing the results of the present paper in relationship to those, could offer a more robust way to connect the Authors' results to networks behavior in pathological brains.

We are grateful for the opportunity to discuss the clinical implication of our study. First of all, we agree with the reviewer and editor that we cannot make a definitive claim about chronic pain with the current study, and thus, we revised the last sentence of the abstract to tone down our claim.

Revisions to the main manuscript (p. 2, in the abstract):

“This study provides new insights into how multiple brain systems dynamically interact to construct and modulate pain experience, advancing our mechanistic understanding of sustained pain.”

However, as we noted above in Essential revisions 4, some of our findings were consistent with the findings from a previous clinical study (Cheng et al., 2022), suggesting the potential to generalize our study to clinical pain conditions. In addition, we previously reported that a predictive model of sustained pain derived from healthy participants performed better at predicting the pain severity of chronic pain patients than the model derived directly from chronic pain patients (Lee et al., 2021), highlighting the advantage of the “component process approach.”

The component process approach aims to develop brain-based biomarkers for basic component processes first, which can then serve as intermediate features for the modeling of multiple clinical conditions (Woo, Chang, Lindquist, and Wager, 2017). This has been one of the core ideas of the Research Domain Criteria (RDoC) (Insel et al., 2010) and the Hierarchical Taxonomy of Psychopathology (HiTOP) (Kotov et al., 2017). If the clinical pain of a patient group is modeled as a whole, it becomes unclear what is being modeled because of the multidimensional and heterogeneous nature of clinical pain (Melzack, 1999) as well as other co-occurring health conditions (e.g., mental health issues, medication use, etc.). The component process approach, in contrast, can specify which components are being modeled and are relatively free from heterogeneity and comorbidity issues by experimentally manipulating the specific component of interest in healthy participants.

The current study was conducted on healthy young adults based on the component process approach. We used oral capsaicin to experimentally induce sustained pain, which unfolds over protracted time periods and has been suggested to reflect some of the essential features of clinical pain (Rainville, Feine, Bushnell, and Duncan, 1992; Stohler and Kowalski, 1999). Therefore, the detailed characterization of the brain processes of sustained pain will be able to serve as an intermediate feature of multiple clinical conditions in future studies.

Now we added the discussion on the clinical generalizability issue in the Discussion section.

Revisions to the main manuscript:

p. 26:

“An interesting future direction would be to examine whether the current results can be generalized to clinical pain. Experimental tonic pain has been known to share similar characteristics with clinical pain (Rainville et al., 1992; Stohler and Kowalski, 1999). In addition, in a recent study, we showed that an fMRI connectivity-based signature for capsaicin-induced orofacial tonic pain can be generalized to chronic back pain (Lee et al., 2021). Therefore, a detailed characterization of the brain responses to sustained pain has the potential to provide useful information about clinical pain.”

p. 26:

“Interestingly, a recent fMRI study on the temporal summation of pain in fibromyalgia patients reported results similar to ours (Cheng et al., 2022), including the intra-network dissociation within the somatomotor network and the inter-network integration between the somatomotor and other networks during pain. Although we cannot directly examine whether the temporal summation of pain gave rise to these network-level changes due to the limitation of our experimental paradigm, these consistent findings between the two studies may suggest that our findings could be generalized to clinical conditions.”

7) The behavioral measure used to assess evoked pain perception (avoidance ratings), has been developed for chronic pain patients and never validated on healthy controls. It might not be an appropriate measure considering the total absence of pain variability in the reported responses over forty-eight subjects. Please discuss this important point.

We acknowledge that pain avoidance measures are not fully validated in the healthy population. Nevertheless, we used this measure in this study for the following two main reasons that outweigh the limitations.

First, a pain avoidance rating provides an integrative measure that can reflect the multi-dimensional aspects of sustained pain. One of the essential functions of pain is to avoid harmful situations and promote survival, and the avoidance motivation induced by pain is composed of not only sensory-discriminative, but also cognitive components including learning, valuation, and contexts (Melzack, 1999). According to the fear-avoidance model (Vlaeyen and Linton, 2012), if the pain-induced avoidance motivation is not resolved for a long time and is maladaptively associated with innocuous environments, chronic pain is likely to develop, suggesting the importance and clinical relevance of pain avoidance measures. In addition, our experimental design is particularly suitable for the use of avoidance rating because the oral capsaicin stimulation is accompanied by the urge to avoid the painful sensation, but it cannot immediately be resolved similar to chronic pain. Moreover, capsaicin is sometimes experienced as intense but less aversive (or even appetitive) in some cases, e.g., spicy food craver (Stevenson and Yeomans, 1993). In this case, avoidance ratings can provide a more reasonable measure of pain compared to the intensity rating.

Second, the avoidance measure provides a common scale on which we can compare different types of aversive experiences, allowing us to conduct specificity tests for a predictive model of pain. For example, a recent study successfully compared the brain representations of two types of pain and two types of aversive, but non-painful experiences (e.g., aversive auditory and visual experiences) using the same avoidance measure (Ceko, Kragel, Woo, Lopez-Sola, and Wager, 2022). These comparisons were possible because the avoidance measure provided one common scale for all the aversive experiences regardless of their types of stimuli.

To provide a better justification for the use of the avoidance measure, we now included the specificity test results of our pain predictive models. More specifically, we tested our module allegiance-based SVM and PCR models of pain on the aversive taste and aversive odor conditions (Appendix 1—figure 13 ).

Despite these advantages, the use of avoidance rating without thorough validation is a limitation of the current study, and thus future studies need to examine the psychometric properties of the avoidance rating, e.g., examining the relationship among pain intensity, unpleasantness, and avoidance measures. However, the current study showed that the predictive models derived with pain avoidance rating (Study 1) could be used to predict the pain intensity rating (Study 2). In addition, the overall time-course of pain avoidance ratings in Study 1 was similar to the time-course of pain intensity ratings in Study 2, providing some supporting evidence for the convergent validity of the pain avoidance measure.

As to the following comment, *“It might not be an appropriate measure considering the total absence of pain variability in the reported responses over forty-eight subjects,”* there are pieces of evidence supporting that the low between-individual variability of ratings is due to the characteristics of our experimental design, not to the fact that we used the avoidance measure. As we discussed in more detail in our response to Essential revisions 1, our experimental procedure based on capsaicin liquid commonly induces the initial burst of painful sensation and the subsequent gradual relief for most of the participants (Figure 1B, left). A similar time-course pattern of ratings was observed in Study 2 (Figure 1B, right), which used the pain “intensity” rating, not the pain avoidance rating. In addition, previous studies with a similar experimental design (i.e., intra-oral capsaicin application) (Berry and Simons, 2020; Lu, Baad-Hansen, List, Zhang, and Svensson, 2013; Ngom, Dubray, Woda, and Dallel, 2001) also showed a similar time-course of pain ratings with low between-individual variability regardless of the rating types (e.g., VAS or irritation intensity), confirming that this observation is not unique to the pain avoidance rating.

Now we added descriptions on the small between-individual variability of pain ratings and the use of avoidance ratings.

Revisions to the main manuscript:

pp. 5-7:

“Note that the overall trend of pain ratings over time was similar across participants because of the characteristics of our experimental design, which has also been observed in the previous studies that used oral capsaicin (Berry and Simons, 2020; Lu et al., 2013; Ngom et al., 2001). However, also note that each individual’s time-course of pain ratings were not entirely the same (Appendix 1—figures 2 and 3).”

p. 26:

“However, there are also differences between the characteristics of capsaicin-induced tonic pain versus clinical pain. For example, clinical pain continuously fluctuates over time in an idiosyncratic pattern (Apkarian, Krauss, Fredrickson, and Szeverenyi, 2001), whereas capsaicin-induced tonic pain showed a similar time-course pattern across the participants—i.e., increasing rapidly and then decreasing gradually (Figure 1B). This typical time-course of pain ratings has been reported in previous studies that used oral capsaicin (Berry and Simons, 2020; Lu et al., 2013; Ngom et al., 2001).”

pp. 26-27:

“Note that Study 1 used a pain avoidance measure that is not yet fully validated in healthy participants. However, we chose to use the pain avoidance measure, which can provide integrative information on the multi-dimensional aspects of pain (Melzack, 1999; Waddell, Newton, Henderson, Somerville, and Main, 1993). It also has a clinical implication considering that the maladaptive associations of pain avoidance to innocuous environments have been suggested as a putative mechanism of transition to chronic pain (Vlaeyen and Linton, 2012). Lastly, the avoidance measure can provide a common scale across different modalities of aversive experience, allowing us to compare their distinct brain representations (Ceko et al., 2022) or test the specificity of their predictive models (Lee et al., 2021) (Appendix 1—figure 13). Although the psychometric properties of the pain avoidance measure should be a topic of future investigation, we expect that the pain avoidance measure would have a high level of convergent validity with pain intensity given the observed similarity between pain avoidance (Study 1) and pain intensity (Study 2) in their temporal profiles. The generalizability of our PCR model across Studies 1 and 2 also supports this speculation. However, there would also be situations in which pain avoidance is dissociated from pain intensity. For example, capsaicin can be experienced to be intense but less aversive or even appetitive in some contexts, such as cravings for spicy food (Stevenson and Yeomans, 1993). In addition, the gradual rise of avoidance ratings during the late period of the control condition in Study 1 would not be observed if the intensity measure was used. Future studies need to examine the relationship between pain avoidance and the other pain assessments and the advantage of using the pain avoidance measure.”

8) Moreover, please address the following questions:• How does the rating scale look like? Is the meaning of the anchors and of the thresholds for weak, moderate, strong, and very strong displayed on a screed during the scan? What are the instructions given to the participants before the start of the experiment? Do they induce any kind of pain expectancy (e.g., "After 6 minutes the pain should start decreasing" or analogous expressions)? A combination of those elements could explain the lack of pain variability.

We appreciate this question. First, we now added the scale and rating instruction that we used in Study 1 to present the experimental procedure more clearly (Appendix 1—figure 1).

Next, below are the scripts for explaining the instruction of pain avoidance rating translated into English.

“We will explain the rating scale that you will use with a trackball mouse during the fMRI scan. You will be asked to report ratings during the experiment, so please carefully listen to the following instructions.

Throughout all sessions except for the structural scan, you will see the following scale on the screen. Use the trackball to move the yellow bar like this to report your ratings.

You will see the question on the screen, “how much do you want to avoid this experience in the future?”. You will report your ratings using the following anchors, ranging from “Not at all” to “Most.” These anchors will not be shown during the scan, so please remember the location of each indicator. To help understand the rating scale, you can think about “how much money would I be paid to do this experience again?”. “Not at all” is an experience you can have again without any monetary reward, whereas “Most” is an experience you do not want to do again no matter how much money you will be paid. “Moderately” is an experience you want to avoid moderately, and “Very strong” is an experience that you would want to avoid immediately, like grabbing and immediately dropping a cup of coffee that you just bought.

Please explain how you understood the scale back to me. (Participants explain) In case you intermittently experience very severe pain, you can rate your pain over “Very strongly.” It is totally okay to rate your pain higher than “Very strongly,” but if your ratings are continuously higher than “Very strongly,” we will consider it too much pain for you and immediately stop the experiment.”

As described above, we did not show anchors and descriptors during scans, and there was no instruction that may induce any kind of pain expectancy. Therefore, these factors are not likely to cause the low variability of pain ratings.

We also put substantial effort into the design and instruction of the rating scale to reduce between-individual variance potentially coming from the different understanding and different uses across participants. For example, we used the general labeled magnitude scale (gLMS) (Bartoshuk et al., 2004), which features intermediate anchors with quasi-logarithmic spacing to permit valid comparisons of sensory experiences between individuals. We explained the meaning of these anchors to participants in great detail and thoroughly checked whether they understood them correctly before the scan. In addition, we let the participants practice the use of the rating scale in the scanner before we started the experiment. We believe that these efforts helped reduce the unwanted between-subject variance in ratings.

9) The behavioral outcome is called "pain avoidance" and the Authors hypothesize that it is proportional to the perceived pain. Is there any evidence proving this correlation? The collected ratings are the answer to the question "how much do you want to avoid this experience in the future?". Is it possible that this question is too generic to be called pain avoidance? Did the Authors quantify in any way the effect of laying down in a scanner for long time? It might play a role in the avoidance index.

As discussed in Essential revisions 7 above, the pain avoidance measure is not fully validated in a healthy population, which is a limitation of our study. There is currently no previous literature that proved the correlation between continuous ratings of pain avoidance and those of pain intensity. However, we think that the pain avoidance and intensity are correlated at least within our experimental paradigm, considering the similar patterns over time between pain avoidance ratings in Study 1 and pain intensity ratings in Study 2 (Figure 1B). The two studies used the same experimental procedure for capsaicin delivery, except that Study 2 used half amount of capsaicin compared to Study 1 because of the shorter scan time.

However, pain avoidance and pain intensity are not identical of course, and there can be some conditions in which the two measures are dissociated (Rainville, Duncan, Price, Carrier, and Bushnell, 1997). Our research group will examine the relationship between pain intensity and avoidance in future studies.

The instruction for the avoidance rating was “Please continuously report how much you want to avoid this experience in the future” (Appendix 1-figure 1). Indeed, we did not specify “pain” in the instruction and intentionally made it generic to use it across different experimental conditions including the bitter taste and aversive odor conditions. The main reason for this was to compare these multiple conditions on the same scale (Appendix 1-figure 13). For this reason, we also observed a slow increase in the avoidance rating score during the control run (Figure 1B), and this is understandable because lying down in a scanner for a long time could become aversive.

Although we understand the concern that our rating was too generic to be called ‘pain avoidance,’ we called it ‘pain avoidance’ because the effect of pain on the changes in avoidance ratings was evident in our analyses (e.g., capsaicin vs. control, early vs. middle vs. late, etc.), and thus the use of ‘pain avoidance’ is not a misnomer or overstatement. However, we will be happy to reconsider it and change the term into an alternative name, such as the ‘avoidance rating,’ if the reviewer thinks it is better.

Now we added a detailed discussion on the use of pain avoidance ratings.

Revisions to the main manuscript (pp. 26-27):

“Note that Study 1 used a pain avoidance measure that is not yet fully validated in healthy participants. However, we chose to use the pain avoidance measure, which can provide integrative information on the multi-dimensional aspects of pain (Melzack, 1999; Waddell et al., 1993). It also has a clinical implication considering that the maladaptive associations of pain avoidance to innocuous environments have been suggested as a putative mechanism of transition to chronic pain (Vlaeyen and Linton, 2012). Lastly, the avoidance measure can provide a common scale across different modalities of aversive experience, allowing us to compare their distinct brain representations (Ceko et al., 2022) or test the specificity of their predictive models (Lee et al., 2021) (Appendix 1—figure 13). Although the psychometric properties of the pain avoidance measure should be a topic of future investigation, we expect that the pain avoidance measure would have a high level of convergent validity with pain intensity given the observed similarity between pain avoidance (Study 1) and pain intensity (Study 2) in their temporal profiles. The generalizability of our PCR model across Studies 1 and 2 also supports this speculation. However, there would also be situations in which pain avoidance is dissociated from pain intensity. For example, capsaicin can be experienced to be intense but less aversive or even appetitive in some contexts, such as cravings for spicy food (Stevenson and Yeomans, 1993). In addition, the gradual rise of avoidance ratings during the late period of the control condition in Study 1 would not be observed if the intensity measure was used. Future studies need to examine the relationship between pain avoidance and the other pain assessments and the advantage of using the pain avoidance measure.”

10) The dynamic measure employed by the Authors is better described from the term "windowed functional connectivity". It is often considered a measure of dynamic functional connectivity and it gives information about fluctuations of the connectivity patterns over time. Nevertheless, the entire focus of the paper, including the title, is on dynamic networks, which inaccurately leads one to think of time-varying measures with higher temporal resolution. This allows one to follow network reorganization over time without averaging 2-min intervals in which several different brain mechanisms might play an important role. In summary, the assumption of constant response throughout 2-min periods of tonic pain and the use of Pearson correlations do not mirror the idea of dynamic analysis expressed by the Authors in title and introduction. Please consider removing "dynamic" from the title, reduce the emphasis on this concept, address possible confounds introduced by the choice of long windows and rephrase the aim of the study in terms of brain network reconfiguration over the main phases of tonic pain experience.

Now we removed the word ‘dynamic’ from many places in the manuscript, including the title. In addition, we added a brief discussion on the reason we chose to use the long and non-overlapping windows for connectivity calculation.

Revisions to the main manuscript (p. 8):

“Although the long duration of the time window without overlaps may obscure the fine-grained temporal dynamics in functional connectivity patterns, we chose to use this long time window based on previous literature (Bassett et al., 2011; Robinson, Atlas, and Wager, 2015), which also used long time windows to obtain more reliable estimates of network structures and their transitions.”

11) Procedure chosen for evoking sustained pain. The measures in figure 1B suggest that the intensity of the painful stimulation is not constant as expected for sustained pain (probably the effect washes out with the saliva). In this case, the first six-minute interval requires particular attention because it encapsulates the real tonic pain phase, and the following ones require more appropriate labels. Ideally the authors should cite previous studies showing that tongue evoked pain elicits a very specific behavioral response (summation, habituation/decrease of pain, absence of pain perception).

We thank the reviewer for the important comment and suggestions. We indeed conducted an extensive search to find appropriate labels for the three phases (i.e., early, middle, and late) for orofacial capsaicin pain, but we could not find one that fits our experimental design well. In addition, we have been hesitant about using the suggested terms, such as *habituation* or *summation*. For example, Segerdahl et al. (2015) used the term “habituation” for the late period of the tonic leg pain induced by topical capsaicin cream. This term seems reasonable for the study in which there was a decrease in pain without the removal of topical capsaicin cream. However, in our case, the decrease in pain can be driven by 1) the gradual removal of capsaicin by saliva over time and 2) active top-down pain regulation. These cannot be described with the term, *habituation*. Segerdahl et al. (2015) also used “relief” to describe the remission of pain after the analgesic procedure (e.g., cooling), but our experiment included no analgesic procedure and thus labeling the late period of pain in our study as “relief” may mislead the readers.

We also had a detailed discussion about the relationship between our study and “temporal summation” in Essential revisions 4. Briefly here, the temporal summation of pain may occur in our experiment, but only in a limited period of time (e.g., the very early period of the run). In addition, it is difficult to systematically investigate the temporal summation effect in our study due to our experimental design. Therefore, it is not clear to us whether the term “summation” can convey the precise meaning of the early period of pain in our study.

According to our literature search, most of the previous studies on oral capsaicin used simple descriptive terms such as increasing or decreasing, which are similar to our description of the three phases of pain—(1) the initiation and maintenance of pain, (2) gradual decrease of pain, and (3) full remission of pain. Some of the previous studies labeled the late period of capsaicin-induced pain as the “waning” period (Chang, Arendt-Nielsen, Graven-Nielsen, Svensson, and Chen, 2001a, 2001b; Coghill, Sang, Berman, Bennett, and Iadarola, 1998; Iadarola et al., 1998). Although we could not come up with better labels than our current labels at this point, we are open to suggestions and will be happy to change the terms in the next round of revision if possible.

12) Moreover, please address the following points:• Does the procedure include a calibration phase? If yes, please add description in the Methods section. If not, how do the Authors explain the relatively small standard error of the mean reported in Figure 1?

Now we added a description that there was no calibration phase before the capsaicin stimulation in the Methods section.

Revisions to the main manuscript (p. 29):

“We did not include pre-calibration to match the subjective level of pain.”

We think the small standard error of the mean in Study 1 is presumably attributed to the characteristics of our experimental design. For a more detailed explanation, please see Essential revisions 1, Essential revisions 7, and Essential revisions 8 above.

13) If possible, add citation proving that there is a very consistent behavioral (pain related) response to the capsaicin: no pain variability against what most of the evoked pain experiments showed.

We now added the citations of the studies showing the small between-individual variability of pain ratings with orofacial capsaicin stimulation.

Revisions to the main manuscript (pp. 5-7):

“Note that the overall trend of pain ratings over time was similar across participants because of the characteristics of our experimental design, which has also been observed in the previous studies that used oral capsaicin (Berry and Simons, 2020; Lu et al., 2013; Ngom et al., 2001). However, also note that each individual’s time-course of pain ratings were not entirely the same (Appendix 1—figures 2 and 3).”

14) Please report in the supplementary material the dots distribution (box plot with visible dots) of the ratings at minute 0, 6, 14, 20.

We now provided dots distribution of the ratings at 0, 6, 14, and 20 min.

15) Community detection analysis. Please clarify the following issues:• The thresholding of the connectivity matrices for the binarization of the networks is certainly a weakness of the study and, more in general, of most of the connectivity analysis (only few estimators have known null distributions). Here, the chosen optimal threshold is the one that maximize the difference between conditions (capsaicin vs controls) in terms of global graph measures, including modularity. I suggest adding a comment on the effect of maximizing a measure depending on the modularity of the networks, on the subsequent community detection algorithm, also based on maximizing modularity (within the network this time). What is the Authors' opinion on this possible confound? Also, what is the rationale/hypothesis behind this procedure for obtaining sparse matrices?

We appreciate the comment and understand the concerns regarding the choice of thresholding parameters. In the following paragraphs, we tried our best to explain the rationale behind our connectivity thresholding and how we mitigated the possible confound from this procedure.

It is often considered a prerequisite to apply thresholding to functional connectivity matrices for brain network analyses (van den Heuvel et al., 2017) because many of the graph analytics were developed for sparse networks (Newman, 2010) and functional connectivity data were likely to contain many spurious correlations. What remains controversial is how to determine the level of thresholding. In an ideal condition, results for the multiple levels of thresholding can be separately reported and discussed. However, we could not repeat all the main analyses multiple times to test different threshold levels (e.g., 0.01, 0.05, 0.1, 0.2, 0.3, 0.4 as Appendix 1—figure 4), each of which takes approximately a few months because the current study used high-dimensional voxel-level connectome data and also the multiple iterations of community detection processes to obtain robust estimates of community structures. Because of the huge computational burden, we could not try all different options and had to determine the one threshold level among the possible candidate values, which should be based on a priori criteria to avoid an arbitrary decision like picking up a specific level of network density (e.g., 0.05 or 0.1). In the current study, we chose the threshold level that maximizes the contrast between the global-level network measures of the capsaicin and control conditions. The chosen threshold level was 0.05, which is a common value for the thresholding of functional connectivity in network neuroscience studies.

In addition, there could be a concern that we examined the network-level contrast between the capsaicin vs. control conditions (e.g., Figure 2) using the threshold that maximized the difference between the conditions. However, its confounding effect should be minimal for the following reasons. First, our main analysis (i.e., multi-layer community detection) was based on weighted networks from all ten time-bins, whereas the global-level network attributes that we used to determine the threshold level were obtained by averaging all time-bin network data and binarizing them to make it unweighted networks. Thus, two analyses were based on different data and the confounding effect should be minimal. Second, most of the global-level network measures used for determining the threshold level were not directly relevant to our community detection analyses except for the modularity measure. Even without the modularity measure, the selection of the threshold level is unchanged. Third, the time-varying changes in brain networks within the capsaicin condition (e.g., Figures 3 and 5) are not relevant to the choice of thresholding, which was based on the contrast between the capsaicin vs. control conditions. Lastly, our SVM (Figure 4) and PCR models (Figure 5) were generalized to the independent test dataset (Study 2, *n* = 74) that was not used to determine the threshold level.

Now we added a discussion on the choice of network density threshold.

Revisions to the main manuscript (p. 28):

“Lastly, the optimization of the network density threshold to maximize the differences in a priori network attributes between the capsaicin versus control conditions may influence the results comparing the community structures between the two conditions. However, our main results were based on temporal dynamics across all time bins, whereas the global-level network attributes that we used to determine the level of threshold were based on the averaged data across all time bins. In addition, our SVM and PCR models were generalized to the independent test dataset, which was not used to determine the threshold level. Therefore, the confounding effect of thresholding on our main results should be minimal.”

16) Did the Authors consider running the analysis with other resolution parameters (γ and omega)?

We thank the reviewer for the important suggestion. However, it is currently not feasible to try different options of parameter sets because of the extremely heavy computational load as described in Essential revisions 15. We followed the conventional choice (γ = 1 and ω = 1) (Bassett, Yang, Wymbs, and Grafton, 2015) similar to many previous studies (Bassett et al., 2011; Bassett, Wymbs, et al., 2013; Bassett et al., 2015; Betzel, Satterthwaite, Gold, and Bassett, 2017; Braun et al., 2016; Braun et al., 2015; Cocuzza, Ito, Schultz, Bassett, and Cole, 2020; Finc et al., 2020; Gifford et al., 2020; Han et al., 2020; Khambhati, Mattar, Wymbs, Grafton, and Bassett, 2018; Lydon-Staley, Ciric, Satterthwaite, and Bassett, 2019; Lydon-Staley, Kuehner, et al., 2019; Pedersen, Zalesky, Omidvarnia, and Jackson, 2018; Shine, Koyejo, and Poldrack, 2016; Telesford et al., 2016). However, we agree that it would be important to test multiple sets of resolution parameters (Betzel and Bassett, 2017) to examine the effects of the parameters. This is a part of the reason that we conducted the predictive modeling, which allows us to avoid excessive reverse-inference from the parameter-sensitive results alone (as described in the Discussion, pp. 27-28).

17) Group-level consensus community detection: I found this section difficult to follow, especially in terms of reasoning for specific choices and steps of the analysis.• Step III: the standard definition of allegiance is a binary matrix whose elements are equal to one only when the two correspondent nodes belong to the same community (as the Authors described). After step III, a new index is computed as the mean of allegiance matrices over time and across subjects. Its value indicates the proportion (percentage) of subjects showing the two nodes in the same community at specific time points. Or how many times the two nodes belong to the same community during the early/middle/late stage of the experiment. Using the name allegiance (a binary measure) to indicate those percentages, might be misleading. I suggest using more appropriate names (measures like "agreement", "dwell time" and similar might be useful) and providing more explanatory examples on how to read the value of the computed measures.

To our best knowledge, the term “allegiance,” “mean allegiance,” or “averaged allegiance” have been used to indicate the time-averaged, subject-averaged, or both time-and subject-averaged module allegiance matrix in previous studies (Bassett et al., 2015; Farahani et al., 2022; Finc et al., 2020; Gifford et al., 2020). Therefore, the use of “allegiance” in our study does not seem to be problematic. However, we also know the term “agreement” has often been used as an alternative to the “allegiance” (Barroso et al., 2021; Baum et al., 2017; Cheng et al., 2022; Mano et al., 2018). Thus, we are willing to revise the term accordingly if the reviewer thinks we should change the term.

18) Step IV: please specify number of permutations.

Permutation was performed once per each of the original community assignments as in the previous study by Bassett (Bassett et al., 2015). We now specified the number of permutations.

Revisions to the main manuscript (p. 34):

“Permutation was performed once per each of the original community assignments.”

19) Step VI: in the same spirit as the two previous comments, I suggest either reconsidering the necessity of this computation, or explaining the reasons for applying a community detection algorithm twice. I believe that additional layers of complexity always require a clear question that they can answer.

Since the multilayer community detection algorithm is non-deterministic and has near-degeneracies, it has been recommended to repeat the community detection multiple times and obtain the consensus partition among the iterations (Bassett, Porter, et al., 2013). In the current study, we followed this recommendation as in many previous studies (Bassett, Porter, et al., 2013; Baum et al., 2017; Braun et al., 2015; Marquez-Legorreta et al., 2022; Mattar, Thompson-Schill, and Bassett, 2018; Petrican and Levine, 2018). Although we understand the considerate comment regarding the complexity of the consensus clustering method, we believe this method provides a reliable and representative community partition that can be replicated over repeated iterations, which is particularly beneficial for the comparison of group-level community structures between different conditions (Figure 2) or time (Figure 3).

20) It remains unclear, how specific the results are to pain. Differences between the control resting state and the capsaicin trials might be – at least partially – driven by other factors, like motion artifacts, saliency, attention, anxiety, etc. Differences between stages over the time-course might, additionally, be driven by scanner drifts (to which the applied approach might be less sensitive, but the possibility is still there) or other gradual processes, e.g. shifts in arousal, attention shifts, alertness, etc. All the above factors might emerge as confounding bias in both of the predictive models. This problem should be thoroughly discussed, and at least the following extra analyses are recommended, in order to attenuate concerns related to the overall specificity and neurobiological validity of the results:• Reporting of, and testing for motion estimates (mean, max, median framewise displacement or anything similar).• Examining whether these factors might, at least partially, drive the predictive models.• e.g. Applying the PCR model on the resting state data and verifying of the predicted timecourse is flat (no inverse U-shape, that is characteristic to all capsaicin trials).

We thank the reviewer for this comment on the important issue regarding the specificity of our results and the potential influences of noise. The effects of head motion and physiological confounds are particularly relevant to pain studies because pain involves substantial physiological changes and often causes head motion. To address the related concerns of specificity, we conducted additional analyses assessing the independence of our predictive models (i.e., SVM and PCR models) from head movement and physiology variables and the specificity of our models to pain versus non-painful aversive conditions (i.e., bitter taste and aversive odor) in Study 1.

First, we examined the overall changes of framewise displacement (FD) (Power, Barnes, Snyder, Schlaggar, and Petersen, 2012), heart rate (HR), and respiratory rate (RR) in the capsaicin condition (Appendix 1—figure 11 ). For the univariate comparison between the capsaicin vs. control conditions (Appendix 1—figure 11A), the results showed that, as expected, the capsaicin condition caused significant changes in head motion and autonomic responses. The mean FD and HR were significantly higher, and the RR was lower in the capsaicin condition compared to the control condition (FD: t_47_ = 5.30, P = 2.98 × 10^-6^; HR: t_43_ = 4.98, P = 1.10 × 10^-5^; RR: t_43_ = -1.91, P = 0.063, paired t-test). In addition, the increased motion and autonomic responses were more prominent in the early period of pain (Appendix 1—figure 11B). The 10-binned (2 mins per time-bin) FD and HR showed a decreasing trend while the RR showed an increasing trend over time in the capsaicin condition. The comparisons between the early (1-3 bins, 0-6 min) vs. late (8-10 bins, 14-20 min) periods of the capsaicin condition showed significant differences both for FD and HR (FD: t_47_ = 6.45, P = 8.12 × 10^-8^; HR: t_43_ = 6.52, P = 6.41 × 10^-8^; RR: t_43_ = -1.61, P = 0.11, paired t-test). These results suggest that while participants were experiencing capsaicin tonic pain, particularly during the early period, head motion and heart rate were increased, while breathing was slowed down. Note that we needed to exclude 4 participants’ data in this analysis due to technical issues with the physiological data acquisition.

Next, we examined whether the changes in head motion and physiological responses influenced our predictive model performance (Appendix 1—figure 12). We first regressed out the mean FD, HR, and RR (concatenated across conditions and participants as we trained the SVM model) from the predicted values of the SVM model with leave-one-subject-out cross-validation (2 conditions × 44 participants = 88) and then calculated the classification accuracy again (Appendix 1—figure 12A). The results showed that the SVM model showed a reduced, but still significant classification accuracy for the capsaicin versus control conditions in a forced-choice test (n = 44, accuracy = 89%, P = 1.41 × 10^-7^, binomial test, two-tailed). We also did the same analysis for the PCR model (10 time-bins × 44 participants = 440) and the PCR model also showed a significant prediction performance (n = 44, mean prediction-outcome correlation r = 0.20, P = 0.003, bootstrap test, two-tailed, mean squared error = 0.159 ± 0.022 [mean ± s.e.m.]) (Appendix 1—figure 12B). These results suggest that our SVM and PCR models capture unique variance in tonic pain above and beyond the head movement and physiological changes.

Lastly, we examined the specificity of our predictive models to pain, by testing the models on the non-painful but aversive conditions including the bitter taste (induced by quinine) and aversive odor (induced by fermented skate) conditions (Appendix 1-figure 13, please see Essential revisions 7). All the model responses were obtained using leave-one-participant-out cross-validation. The results showed that the overall model responses of the SVM model for the bitter taste and aversive odor conditions were higher than those for the control condition but lower than the capsaicin condition (Appendix 1-figure 13A). Classification accuracies for comparing capsaicin vs. bitter taste and capsaicin vs. aversive odor were all significant (for capsaicin vs. bitter taste, accuracy = 79%, P = 6.17 × 10^-5^, binomial test, two-tailed, Appendix 1-figure 13C; for capsaicin vs. aversive odor, accuracy = 83%, P = 3.31 × 10^-6^, binomial test, two-tailed, Appendix 1-figure 13E), supporting the specificity of our SVM model of pain. Similarly, the model responses of the PCR model for the bitter taste and aversive odor conditions were lower than the capsaicin condition, and their temporal trajectories were less steep and fluctuating compared to the capsaicin condition (Appendix 1-figure 13B). The time-course of the model responses for the control condition was flatter than all other conditions and did not show the inverted U-shape. Furthermore, the model responses of the bitter taste and aversive odor conditions did not show the significant correlations with the actual avoidance ratings (bitter taste: mean prediction-outcome correlation r = 0.05, P = 0.41, bootstrap test, two-tailed, mean squared error = 0.036 ± 0.006 [mean ± s.e.m.], Appendix 1-figure 13D; aversive odor: mean prediction-outcome correlation r = 0.12, P = 0.06, bootstrap test, two-tailed, mean squared error = 0.044 ± 0.004 [mean ± s.e.m.], Appendix 1-figure 13F), suggesting the specificity of PCR model to pain.

Overall, we have provided evidence that our models can predict pain ratings above and beyond the head motion and physiological changes and that the models are more responsive to pain compared to non-painful aversive conditions.

Now we added descriptions on the specificity tests to the main manuscript and also to the Appendix 1.

Revisions to the main manuscript (p. 20):

“Specificity of the module allegiance-based predictive models

To examine whether the predictive models were specific to pain and the prediction performances were not influenced by confounding variables such as head motion and physiological changes, we conducted additional analyses as shown in Appendix 1—figures 11-13. The SVM and PCR models showed significant prediction performances even after controlling for head motion (i.e., framewise displacement) and physiological responses (i.e., heart rate and respiratory rate) (Appendix 1—figures 11 and 12) and did not respond to the non-painful but aversive conditions including the bitter taste and aversive odor conditions (Appendix 1—figure 13), supporting the specificity of our predictive to pain. For details, please see Appendix 1.”

Revisions to the Appendix 1 (pp. 2-4):

“Specificity analysis (Appendix 1—figures 11-13)

To examine whether the predictive models (i.e., SVM and PCR models) were specific to pain and not influenced by confounding noises, we conducted additional specificity analysis assessing the independence of the models from head movement and physiology variables and specificity of our models to pain versus non-painful aversive conditions (i.e., bitter taste and aversive odor) in Study 1.

[…]

Overall, we have provided evidence that the module allegiance-based models can predict pain ratings above and beyond the movement and physiological changes, and are more responsive to pain compared to non-painful aversive conditions, which suggest the specificity of our results to pain.”

21) Statistical inference. An important issue is the (apparent) lack of statistical inference when analyzing the differences in the group-level consensus community structures (both when comparing capsaicin to control and when analysing changes over the time-course of the capsaicin-challenge). Although the observed changes seem biologically plausible and fit very well to previous results, without proper statistical inference we can't determine, how likely such differences are to emerge just by chance. This makes all results on Figures 2 and 3, and points 1, 4 and 5 in the discussion partially or fully speculative or weakly underpinned, comprising a large proportion of the current version of the manuscript. There are two main ways of handling this issue:• Enhancing (or clarifying potential misunderstandings regarding) the methodology (see my concrete, and hopefully feasible, suggestions in the "private part" of the review). There are likely many ways to test the significance of these differences. Two permutation testing-based ideas are (i) permuting the labels ctr-capsaicin, or early-mid-late, repeating the analysis, constructing the proper null distribution of e.g. the community size changes and obtain the p-values and (ii) "trace back" communities to the individual level and do (nonparametric) statistical inference there.• De-weighting the presentation and the discussion of the related results.

We appreciate this important comment. We did not conduct statistical inference when comparing the group-level consensus community affiliations of the different conditions (Figure 2) or different phases (Figure 3) because of the difficulty in matching the community affiliation values of the networks to be compared.

For example, let us assume that the 800 out of 1,000 voxels of community #1 and 1,000 out of 4,000 voxels of community #2 in the control condition are commonly affiliated with the same community #3 in the capsaicin condition. To compare the community affiliation between two conditions, we should first match the community label of the capsaicin condition (i.e., #3) to that of the control condition (i.e., #1 or #2), and here a dilemma occurs; if we prioritize the proportion of the overlapping voxels for the matching, the common community should be labeled as #1, whereas if we prioritize the number of the overlapping voxels for the matching, the label of the common community should be #2. Although both choices look reasonable, none of them can be a perfect solution.

As the example above, it is impossible to exactly match the community affiliation of the different networks. We must choose an imperfect criterion for the matching procedure, which essentially affects the comparison of network structure. This was the main reason that we limited our results of Figures 2-3 to a qualitative description based on visual inspection. Moreover, the group-level consensus community structures in Figures 2-3 are not a simple group statistic like sample mean; they were obtained from multiple steps of analyses including permutation-based thresholding and unsupervised clustering, which could further complicate the interpretation of statistical tests.

Alternatively, there is a slightly different but more rigorous approach to the comparisons of the community structures, which is the Phi-test (Alexander-Bloch et al., 2012; Lerman-Sink_off_ and Barch, 2016). Instead of direct use of the community labels, this method converts the community label of each voxel into a list of module allegiance values between the seed voxel and all the voxels of the brain (i.e., 1 if the seed and target voxels have the same community label and 0 otherwise). This allows quantitative comparisons of voxel-level community profiles between different conditions without an arbitrarily matching of the community labels. We adopted this Phi-test for our analyses to examine whether the regional community affiliation pattern is significantly different between (i) the capsaicin vs. control conditions and (ii) the early vs. late periods of pain (Appendix 1-figure 6), which correspond to the main findings of the Figures 2 and 3 in our manuscript, respectively.

More specifically, to compare the group-level consensus community structures between the capsaicin vs. control conditions and the early vs. late periods, we first obtained a seed-based module allegiance map for each voxel (i.e., using each voxel as a seed). Then, we calculated a correlation coefficient of the module allegiance values between two different conditions for each voxel. This correlation coefficient can serve as an estimate of the voxel-level similarity of the consensus community profile.

Because module allegiance is a binary variable, these correlation values are Phi coefficients. A small Phi coefficient means that the spatial pattern of brain regions that have the same community affiliation with the given voxel are different between the two conditions. For example, if a voxel is connected to the somatomotor-dominant community during the capsaicin condition and the default-mode-dominant community during the control condition, the brain regions that have the same community label with the voxel will be very different, and thus the Phi coefficient will become small. Moreover, the Phi coefficient can be small even if a voxel is affiliated as the same (matched) community label for both conditions, when the spatial patterns of the same community is different between conditions.

To calculate the statistical significance of the Phi coefficient, we conducted permutation tests, in which we randomly shuffled the condition labels in each participant and obtained the group-level consensus community structure for each shuffled condition. Then, we calculated the voxel-level correlations of the module allegiance values between the two shuffled conditions. We repeated this procedure 1,000 times to generate the null distribution of the Phi coefficients, and calculated the proportion of null samples that have a smaller Phi coefficient (i.e., a more dis-similar regional community structure) than the non-shuffled original data.

Results showed that there are multiple voxels with statistical significance (permutation tests with 1,000 iterations, one-tailed) in the area where the community affiliations of the two contrasting conditions were different (Appendix 1—figure 6). For example, the frontoparietal and subcortical regions for the capsaicin vs. control (c.f., Figure 2), and the frontoparietal, subcortical, brainstem, and cerebellar regions for the early vs. late period of pain (c.f., Figure 3) contain voxels that survived after thresholding with FDR-corrected *q* < 0.05, suggesting the robustness of our main results.

Particularly, the somatomotor and insular cortices showed statistical significance in the permutation test, and this may reflect the large changes in other areas that are connecting to the somatomotor and insular cortices across different conditions. The statistical significance was also observed in the visual cortex, which was unexpected. We interpret that the spatial distribution of the visual network community is too stable across conditions, and thus the null distribution from permutation formed a very narrow distribution of Phi coefficients. Therefore, a small change in the community structure could achieve statistical significance.

Now we added descriptions on the permutation tests.

Revisions to the main manuscript:

p. 9:

“Permutation tests confirmed that the community assignment in the frontoparietal and subcortical regions showed significant changes between the capsaicin versus control conditions (Appendix 1—figure 6A).”

pp. 13-14:

“Permutation tests further confirmed that the community assignment in the frontoparietal, subcortical, brainstem, and cerebellar regions showed significant changes between the early versus late period of pain (Appendix 1—figure 6B).”

pp. 36-37:

“Permutation tests for regional differences in community structures. To test the statistical significance of the voxel-level difference of consensus community structures (Figures 2 and 3), we performed the following Phi-test (Alexander-Bloch et al., 2012; Lerman-Sink_off_ and Barch, 2016). First, for each given voxel, we compared the community label of the voxel to the community label of all the voxels, generating a list of voxel-seed module allegiance values that allow quantitative comparison of voxel-level community profile (e.g., [1, 0, 1, 1, 0, 0, …], whose element is equal to 1 if the seed and target voxels were assigned to the same community and 0 otherwise). Next, a correlation coefficient was calculated between the module allegiance values of the two different brain community structures (i.e., capsaicin versus control, and early versus late). This correlation coefficient is an estimate of the regional similarity of community profiles (here, the correlation coefficient is Phi coefficient because module allegiance is a binary variable). To estimate the statistical significance of the Phi coefficient, we performed permutation tests, in which we randomly shuffled the labels and then obtained the group-level consensus community structures from the shuffled data. Then, the Phi coefficient between the module allegiance values of the two shuffled consensus community structures was calculated. We repeated this procedure 1,000 times to generate the null distribution of the Phi coefficient for each voxel. Lastly, we examined the probability to observe a smaller Phi coefficient (i.e., a more dissimilar community profile) than the one from the non-shuffled original data, which corresponds to the P-value of the permutation test. All the P-values were one-tailed as the hypothesis of this permutation test is unidirectional.”

Reviewer #1 (Recommendations for the authors):• The authors emphasize the term "pain supersystem". This term is not very well-introduced yet and the necessity for such a term is unclear. I recommend that the authors rely less on this term and omit it at least from the abstract.

We agree with the reviewer’s suggestion. The term “pain supersystem” is now mostly replaced with a more descriptive one (i.e., an extended somatomotor-dominant community), including the abstract.

• The statement in the abstract "In the early stage, the orofacial areas of the primary somatomotor cortex were separated from the other primary somatomotor cortices and integrated with…" is a bit ambiguous. It might better read "In the early stage, the orofacial areas of the primary somatomotor cortex were separated from other areas of the primary somatomotor cortex and integrated with…"

We thank the reviewer for this helpful comment, and now revised the statement in the abstract as suggested.

Reviewer #2 (Recommendations for the authors):• I suggest reducing the amount of text in the figures. All the information needed to understand the illustrations should be included in the captions. Figure 1 and Figure 7 are the ones that require the most attention in this respect.

We agree with this suggestion, and now substantially reduced the amount of text in Figures 1 and 7.

• It might be a good idea to specify when any previous evidence used to justify the current analysis or to make inferences on the obtained results actually come from the Authors' previous publications. Especially if they are extracted from the same dataset, this information is relevant.

We now specified when we used our previous publication to interpret or justify our current findings.

Revisions to the main manuscript:

p. 3:

“Tonic pain has long been used as an experimental model of clinical pain (Dubuisson and Dennis, 1977), and our previous study demonstrated that capsaicin-induced tonic orofacial pain shows a network-level brain representations similar to clinical pain, suggesting its clinical relevance (Lee et al., 2021).”

p. 7:

“In this study, we used this approach to examine the temporal changes of brain network structures during sustained pain, which cannot be done with conventional functional connectivity-based analyses (Lee et al., 2021).”

p. 26:

“In addition, in a recent study, we showed that an fMRI connectivity-based signature for capsaicin-induced orofacial tonic pain can be generalized to chronic back pain (Lee et al., 2021).”

• In their previous paper, the Authors had access to a dataset including the experimental conditions: tonic capsaicin pain, tonic aversive taste, and tonic aversive odor. Did the Authors analyze the communities structure during those controls conditions? Did they consider testing their classifier on them? In my opinion, it would add a lot of robustness to the study findings, and it would make the obtained results reliable and unquestionably pain related (thinking of the more general avoidance ratings).

We thank the reviewer for this helpful comment. We conducted additional analyses to demonstrate the specificity of our predictive models to pain by testing the models on non-painful but aversive conditions including the bitter taste and aversive odor conditions in Essential revisions 20 (Appendix 1-figure 13). The results showed that the model responses of the SVM model were significantly higher in capsaicin condition than the bitter taste and aversive odor conditions, and that the PCR model was not predictive of the avoidance ratings of the bitter taste and aversive odor conditions, supporting the specificity of the predictive models.

• In terms of data availability, the Authors declared that data and codes will be shared upon publication. I would appreciate their availability if there will be a second loop of revisions before potential publication.

We are now ready to share the codes and processed data (e.g., brain community assignment) for generating the main figures (https://github.com/cocoanlab/brain_reconfig_pain). The raw data of Study 1 will be shared upon request, and the data of Study 2 will be shared later as a part of the large-scale dataset (including heat and capsaicin pain) that is still ongoing and will be publicly open.

Reviewer #3 (Recommendations for the authors):– As the authors mention the cross-validated evaluation of the PCR model is biased due to hyperparameter optimization. While the independent evaluation resolves any related concerns, the authors might consider applying a nested cross-validation framework, to have unbiased estimates for the discovery dataset, as well.

We agree that the nested cross-validation could provide less biased estimates of prediction performance in the discovery dataset. Following the suggestion, we conducted a nested leave-one-participant-out cross-validation of the PCR model. The results also showed a significant prediction performance (mean prediction-outcome correlation *r* = 0.28, *P* = 1.00 × 10^-5^, bootstrap test, two-tailed, mean squared error = 0.044 ± 0.006 [mean ± s.e.m.], number of components = 13.94 ± 0.14 [mean ± s.e.m.]), which was comparable to the previous results (*r* = 0.29).

We now added the following description to the manuscript.

Revisions to the main manuscript:

p. 19:

“To obtain a less biased estimate of performance in the training data, we used nested leave-one-participant-out cross-validation that separates the hyper-parameter tuning and testing (see Materials and methods for details). The results showed that prediction performance was significant (mean prediction-outcome correlation r = 0.28, P = 1.00 × 10^-5^, bootstrap test, two-tailed, mean squared error = 0.044 ± 0.006 [mean ± s.e.m.], number of principal components = 13.94 ± 0.14 [mean ± s.e.m.]), suggesting that the individuals’ brain community structures are predictive of the temporal change of sustained pain.”

p. 38:

“To overcome the potential bias in the performance estimation due to the optimal selection of PC number, we additionally conducted nested cross-validation which has double loops of leave-one-participant-out cross-validation; the inner loop where the hyper-parameter (i.e., the PC number) was selected, and the outer loop where the actual prediction was done using the hyperparameters chosen from the inner loop. Since the hyper-parameter tuning and testing were separated into the inner and outer loops, this procedure provides a less biased estimate of prediction performance even in the training dataset (Study 1).”

– Optimizing the network density threshold in the same dataset, especially on one of the conditions-of-interest (Q1: capsaicin vs. controls) may be circular (as the optimized global network metrics may well be associated to the community structure). On the other hand, this potential circularity does not affect all he results (e.g definitely not the results based on the independent test dataset) and in general, I don't think this would significantly affect the results. Nevertheless, performing (or reproducing) the optimization on independent data would be reassuring. Alternatively, this issue must be discussed as a potential limitation/bias.

We addressed this comment in Essential revisions 15.

– While this is not explicitly stated, prediction performance is evaluated only on the within subject-level. For better comparability to other methods, please report and discuss the "between-subject" estimates, too (i.e. how well can we classify/predict from a single session/window of a single subject).

The reason we mainly focused on the within-individual prediction was that the main purpose of our study was to examine the brain network changes during sustained pain within individuals, not to examine the between-individual differences. Nonetheless, we agree that providing the between-individual prediction performance could be helpful to compare our models with other existing models. We now provide additional results of between-individual prediction.

For the SVM model, we evaluated the classification accuracy for the capsaicin versus control conditions across all the participants, instead of the forced-choice test that compared the two conditions within individuals. The results for Study 1 were as follows: accuracy with an optimal threshold = 88% (*P* = 1.82 × 10^-14^, binomial test, two-tailed), 85% sensitivity, 90% specificity, area under the curve (AUC) = 0.94. The results for Study 2 were as follows: accuracy with an optimal threshold = 76% (*P* = 2.84 × 10^-10^, binomial test, two-tailed), 72% sensitivity, 80% specificity, AUC = 0.80.

For the PCR model, we calculated the correlation between mean pain ratings and mean model responses (i.e., between-individual prediction-outcome correlation) for capsaicin condition. The results for Study 1 were as follows: *r* = 0.41, *P* = 0.004, one-sample *t*-test, two-tailed, mean squared error = 0.024. The results for Study 2 were as follows: *r* = 0.27, *P* = 0.018, one-sample *t*-test, two-tailed, mean squared error = 0.022.

Now we added descriptions on the between-individual prediction to the main manuscripts and also to the Appendix 1.

Revisions to the main manuscript:

p. 14:

“For the classification accuracy across all the participants instead of the forced-choice test, please see Appendix 1.”

p. 19:

“For the between-individual prediction-outcome correlation of mean pain ratings, please see Appendix 1.”

Revisions to the Appendix 1 (p. 4):

“Between-individual predictive performances

For SVM model, we evaluated the classification accuracy for the capsaicin versus control conditions across all the participants, instead of the forced-choice test that compared the two conditions within individuals. The results for Study 1 were as follows: accuracy with an optimal threshold = 88% (P = 1.82 × 10^-14^, binomial test, two-tailed), 85% sensitivity, 90% specificity, area under the curve (AUC) = 0.94. The results for Study 2 were as follows: accuracy with an optimal threshold = 76% (P = 2.84 × 10^-10^, binomial test, two-tailed), 72% sensitivity, 80% specificity, AUC = 0.80.

For PCR model, we calculated the correlation between mean pain ratings and mean signature responses (i.e., between-individual prediction-outcome correlation) for capsaicin condition. The results for Study 1 were as follows: r = 0.41, P = 0.004, one-sample t-test, two-tailed, mean squared error = 0.024. The results for Study 2 were as follows: r = 0.27, P = 0.018, one-sample t-test, two-tailed, mean squared error = 0.022.”

– Introduction: discussing the possible relation of the present work to chronic pain or other clinical pain conditions is not sufficient.

It seems that this comment is similar to the Essential revisions 6; we now added the discussion on the relationship with clinical pain conditions in the main manuscript. We are also happy to revise the current manuscript if more discussion is needed in the other sections including the Introduction.

Revisions to the main manuscript (p. 26):

“An interesting future direction would be to examine whether the current results can be generalized to clinical pain. Experimental tonic pain has been known to share similar characteristics with clinical pain (Rainville et al., 1992; Stohler and Kowalski, 1999). In addition, in a recent study, we showed that an fMRI connectivity-based signature for capsaicin-induced orofacial tonic pain can be generalized to chronic back pain (Lee et al., 2021). Therefore, a detailed characterization of the brain responses to sustained pain has the potential to provide useful information about clinical pain. However, there are also differences between the characteristics of capsaicin-induced tonic pain versus clinical pain. For example, clinical pain continuously fluctuates over time in an idiosyncratic pattern (Apkarian et al., 2001), whereas capsaicin-induced tonic pain showed a similar time-course pattern across the participants—i.e., increasing rapidly and then decreasing gradually (Figure 1B). This typical time-course of pain ratings has been reported in previous studies that used oral capsaicin (Berry and Simons, 2020; Lu et al., 2013; Ngom et al., 2001). Although we would expect our results reflect the general pattern of brain network changes during the rise and fall of sustained pain, it remains an empirical question how much they will be generalizable across different clinical pain conditions. Interestingly, a recent fMRI study on the temporal summation of pain in fibromyalgia patients reported results similar to ours (Cheng et al., 2022), including the intra-network dissociation within the somatomotor network and the inter-network integration between the somatomotor and other networks during pain. Although we cannot directly examine whether the temporal summation of pain gave rise to these network-level changes due to the limitation of our experimental paradigm, these consistent findings between the two studies may suggest that our findings could be generalized to clinical conditions.”

– More information is needed about the individual variability of the pain-related behavioral time-courses (maybe in the supplementary info). Was remission complete in all participants?

We think this comment is partially addressed in Essential revisions 1, Essential revisions 7 and Essential revisions 8. Although between-individual variability of avoidance ratings was small because of the characteristics of our experimental design that commonly induces initial burst of painful sensation and the subsequent gradual relief, each individual’s timecourse of pain ratings actually shows distinct patterns and there were some participants who report severe pain in the late period (Appendix 1-figure 2, please see Essential revisions 1). The term “remission” was used because the difference between avoidance ratings of capsaicin vs. control conditions became insignificant in the late period of pain (from 17.3 min to the end, two-tailed *P*s > 0.05, paired *t*-test, BF_01_ = 1.01-4.71), which does not mean that all participants experienced complete remission of pain.

We also added descriptions on the between-individual variability of pain ratings.

Revisions to the main manuscript:

pp. 5-7:

“Note that the overall trend of pain ratings over time was similar across participants because of the characteristics of our experimental design, which has also been observed in the previous studies that used oral capsaicin (Berry and Simons, 2020; Lu et al., 2013; Ngom et al., 2001). However, also note that each individual’s time-course of pain ratings were not entirely the same (Appendix 1—figures 2 and 3).”

p. 26:

“However, there are also differences between the characteristics of capsaicin-induced tonic pain versus clinical pain. For example, clinical pain continuously fluctuates over time in an idiosyncratic pattern (Apkarian et al., 2001), whereas capsaicin-induced tonic pain showed a similar time-course pattern across the participants—i.e., increasing rapidly and then decreasing gradually (Figure 1B). This typical time-course of pain ratings has been reported in previous studies that used oral capsaicin (Berry and Simons, 2020; Lu et al., 2013; Ngom et al., 2001).”

– Some participants might be more tolerant for capsaicin than others, due to eating habits. Please discuss whether this could potentially affect the results.

Since we measured the pain avoidance that reflects multidimensional aspects of pain, multiple factors such as cravings for spicy food (Stevenson and Yeomans, 1993) can affect their subjective reports. However, we did not systematically screen out participants based on those factors. This is because we wanted to capture as much variance as possible, which potentially helps to get more generalizable results for different settings. Although the group-level consensus community may obscure the between-individual differences, predictive models can efficiently reflect those differences and provide novel findings compared to the group-level consensus community analysis.

We now added brief discussion on the individual differences in pain sensitivity.

Revisions to the main manuscript (p. 27):

“However, there would also be situations in which pain avoidance is dissociated from pain intensity. For example, capsaicin can be experienced to be intense but less aversive or even appetitive in some contexts, such as cravings for spicy food (Stevenson and Yeomans, 1993).”

– At many points, e.g. in paragraph 25 on page x or 5 on page 25, it is mentioned that the models generalized across two datasets. While the terminology is currently heterogenous, I kindly suggest to use the term "generalization" only to the independent test dataset (here the models really had to generalize to scanning parameters, paradigm differences, etc.)

We agree with the Reviewer’s opinion, and now revised the paragraph accordingly.

Revisions to the main manuscript:

p. 4:

“These models were further generalized to the independent tonic pain dataset (Study 2).”

p. 25:

“Lastly, the predictive modeling approach can provide information about the robustness and usefulness of the multi-layer community detection method by allowing us to test the prediction performance for the discovery dataset and generalizability for the independent test dataset.”

– it's a bit unclear why the pain avoidance ratings fall. One would, somewhat naively, hypothesize that if the participant once though she would never repeat this experiment again, why would she change her mind a couple of minutes later, when the memories of pain are still vivid. Please comment on this.

The gradual decrease of pain avoidance ratings was because we asked participants to continuously report the avoidance of the experience at the moment. For your information, the instruction was “*How much do you want to avoid this experience in the future?*”. If we asked this question only once per condition, the term ‘this experience’ would be understood as the experience of the whole experiment as the Reviewer’s comment. However, we gave the introduction that they should continuously answer the question during the scans. We provided detailed descriptions of the rating scale and introduction procedure in Essential revisions 8.

– Please add a short discussion of the differences of the behavioral ratings and how they might affect the findings. (this might be positive thing: a sign of generalization across behavioral assessment protocols).

We think this comment is partially addressed in Essential revisions 7 and Essential revisions 9. We think that the pain avoidance measure would have a high level of convergent validity with pain intensity, considering that the overall time-course of pain avoidance ratings in Study 1 was similar to the time-course of the pain intensity ratings in Study 2, and the predictive models derived with pain avoidance rating (Study 1) could be used to predict the pain intensity rating (Study 2). However, pain avoidance and pain intensity are not identical of course, and there can be some conditions in which the two measures are dissociated. Our research group will examine the relationship between pain intensity and avoidance in future studies.

We now added descriptions on the difference between pain avoidance and intensity ratings.

Revisions to the main manuscript (p. 27):

“Although the psychometric properties of the pain avoidance measure should be a topic of future investigation, we expect that the pain avoidance measure would have a high level of convergent validity with pain intensity given the observed similarity between pain avoidance (Study 1) and pain intensity (Study 2) in their temporal profiles. The generalizability of our PCR model across Studies 1 and 2 also supports this speculation. However, there would also be situations in which pain avoidance is dissociated from pain intensity. For example, capsaicin can be experienced to be intense but less aversive or even appetitive in some contexts, such as cravings for spicy food (Stevenson and Yeomans, 1993). In addition, the gradual rise of avoidance ratings during the late period of the control condition in Study 1 would not be observed if the intensity measure was used. Future studies need to examine the relationship between pain avoidance and the other pain assessments and the advantage of using the pain avoidance measure.”

– Please clarify why *pain* avoidance (slightly) increased in the control resting state scan.

We think this comment is partially addressed in Essential revisions 9. We could measure the avoidance rating in the control condition because the question does not contain the term “pain” (please see Essential revisions 8 for details of our rationale behind the rating scale) and thus can provide a generalizable rating scale shared across different conditions. The increase of avoidance rating in the control condition is expected because lying down in a scanner for a long time can cause participants to feel bored and tired, or even induce painful sensations (e.g., back pain). However, we think this effect did not significantly affect the current results including the predictive models, considering that the model responses of the PCR model for the control condition (Appendix 1-figure 13, please see Essential revisions 7) did not show a similar time-course as the avoidance rating of the control condition. Although we understand the concern that our rating was too generic to be called ‘pain avoidance,’ we called it ‘pain avoidance’ because the effect of pain on the changes in avoidance ratings was evident in our analyses (e.g., capsaicin vs. control, early vs. middle vs. late, etc.), and thus the use of ‘pain avoidance’ is not a misnomer or overstatement. However, we will be happy to reconsider it and change the term into an alternative name, such as the ‘avoidance rating,’ if the reviewer thinks it is better.

– Please provide more rationale for the choice of ML algorithms.

The SVM and PCR are widely accepted algorithms for finding the low-dimensional latent components of highly correlated data such as brain networks. Although the other types of algorithms are available for the current study, we think it may hamper the simplicity and interpretability of the predictive modeling if we try many different options of algorithms. Thus we chose the most representative algorithms for classification (SVM) and regression (PCR) of brain data, respectively.

We now added the descriptions for the choice of machine learning algorithms.

Revisions to the main manuscript (p. 14):

“We chose to use the SVM and PCR because they are representative linear algorithms for finding the low-dimensional latent components of highly correlated data such as brain networks.”

– How were the hyperparameters set for the SVM? Why were those not optimized, too?

We did not conduct hyperparameter tuning for the SVM model and used the conventional choice of hyperparameter C=1, which is a widely used, default value for SVM modeling. However, to our knowledge, there is no conventional choice of hyperparameter for the PCR algorithm. Sometimes the principal components that explained 90%, 95%, or 99% of the variance were used, but the level of threshold also varies by researchers. Therefore, we had no choice but to conduct an exhaustive search for the number of principal components despite the possibility of overfitting and then tested the chosen PCR model onto the independent dataset to prove its generalizability.

We now added the descriptions for the choice of hyperparameter of the SVM model.

Revisions to the main manuscript (p. 37):

“A regularization hyperparameter C was set to 1, which is a conventional choice for SVM.”

– Why was only one hub selected for the seed-based analysis in the case of the classifier?

It was coincidental that the left ventral primary somatomotor region (tongue area) was selected as the common hub region for both positive and negative weights. We speculate that this result shows the high importance of the hub region.

– While the prediction performances are obviously significant, testing for this with bootstrapping may be suboptimal, as bootstrap samples may inherit non-normality from the parent dataset. Permutation test would be more "elegant" in my opinion.

We thank the Reviewer’s suggestion and agree that bootstrapping does not always generate a normal distribution depending on the characteristics of the original data. Therefore, we examine the normality of the 10,000 bootstrapped mean prediction-outcome correlation coefficients of the PCR model for Study 1 (Author response image 1). The results from Lilliefors test did not reject the alternative hypothesis (*P* = 0.27, two-tailed), suggesting that we cannot assume the bootstrapped distribution as a non-normal one.

**Author response image 1. sa2fig1:** The distribution of the bootstrapped prediction-outcome correlations. We conducted bootstrap tests to examine whether the distribution of within-individual prediction-outcome correlation coefficients of the PCR model were significantly different from zero for Study 1. The distribution of correlation coefficients met normality assumption, *P* = 0.27, Lilliefors test, two-tailed.

For clarification, we additionally tried the permutation test to examine whether the mean of within-individual prediction-outcome correlation coefficients is different from zero. To generate the null distribution, we randomly flip the sign of each correlation value, e.g., 0.24 to -0.24, -0.51 to 0.51, etc., and then calculate the mean of those randomly sign-flipped correlation values. This procedure was repeated with 10,000 iterations to obtain the null distribution of the mean correlation. The two-tailed *P*-value from this permutation test was 0.0002 for Study 1 and 0.0002 for Study 2, all of which were far below 0.05.

– Discussion: relation to consciousness might be somewhat speculative, should be hedged.

We now removed the word ‘conscious’ or ‘consciousness’ from the whole manuscript.

– Will the raw data also be shared?

The raw data of Study 1 will be shared upon request. For the Study 2 data, we are planning to collect data from more than 100 participants with the goal of publicly sharing them as a part of a large-scale pain dataset (including heat and capsaicin pain). Thus, we will eventually publicly share the Study 2 data. Of course, the raw data of Study 2 can be shared upon request before we fully open the whole dataset.

References

Alexander-Bloch, A., Lambiotte, R., Roberts, B., Giedd, J., Gogtay, N., and Bullmore, E. (2012). The discovery of population differences in network community structure: new methods and applications to brain functional networks in schizophrenia. *Neuroimage, 59*(4), 3889-3900. doi:10.1016/j.neuroimage.2011.11.035

Apkarian, A. V., Krauss, B. R., Fredrickson, B. E., and Szeverenyi, N. M. (2001). Imaging the pain of low back pain: functional magnetic resonance imaging in combination with monitoring subjective pain perception allows the study of clinical pain states. *Neurosci Lett, 299*(1-2), 57-60. doi:10.1016/s0304-3940(01)01504-x

Barroso, J., Wakaizumi, K., Reis, A. M., Baliki, M., Schnitzer, T. J., Galhardo, V., and Apkarian, A. V. (2021). Reorganization of functional brain network architecture in chronic osteoarthritis pain. *Hum Brain Mapp, 42*(4), 1206-1222. doi:10.1002/hbm.25287

Bartoshuk, L. M., Duffy, V. B., Green, B. G., Hoffman, H. J., Ko, C. W., Lucchina, L. A.,... Weiffenbach, J. M. (2004). Valid across-group comparisons with labeled scales: the gLMS versus magnitude matching. *Physiol Behav, 82*(1), 109-114. doi:10.1016/j.physbeh.2004.02.033

Bassett, D. S., Porter, M. A., Wymbs, N. F., Grafton, S. T., Carlson, J. M., and Mucha, P. J. (2013). Robust detection of dynamic community structure in networks. *Chaos, 23*(1), 013142. doi:10.1063/1.4790830

Bassett, D. S., Wymbs, N. F., Porter, M. A., Mucha, P. J., Carlson, J. M., and Grafton, S. T. (2011). Dynamic reconfiguration of human brain networks during learning. *Proc Natl Acad Sci U S A, 108*(18), 7641-7646. doi:10.1073/pnas.1018985108

Bassett, D. S., Wymbs, N. F., Rombach, M. P., Porter, M. A., Mucha, P. J., and Grafton, S. T. (2013). Task-based core-periphery organization of human brain dynamics. *PLoS Comput Biol, 9*(9), e1003171. doi:10.1371/journal.pcbi.1003171

Bassett, D. S., Yang, M., Wymbs, N. F., and Grafton, S. T. (2015). Learning-induced autonomy of sensorimotor systems. *Nat Neurosci, 18*(5), 744-751. doi:10.1038/nn.3993

Baum, G. L., Ciric, R., Roalf, D. R., Betzel, R. F., Moore, T. M., Shinohara, R. T.,... Satterthwaite, T. D. (2017). Modular Segregation of Structural Brain Networks Supports the Development of Executive Function in Youth. *Curr Biol, 27*(11), 1561-1572 e1568. doi:10.1016/j.cub.2017.04.051

Berry, D. N., and Simons, C. T. (2020). Assessing regional sensitivity and desensitization to capsaicin among oral cavity mucosae. *Chem Senses*. doi:10.1093/chemse/bjaa033

Betzel, R. F., and Bassett, D. S. (2017). Multi-scale brain networks. *Neuroimage, 160*, 73-83. doi:10.1016/j.neuroimage.2016.11.006

Betzel, R. F., Satterthwaite, T. D., Gold, J. I., and Bassett, D. S. (2017). Positive affect, surprise, and fatigue are correlates of network flexibility. *Sci Rep, 7*(1), 520. doi:10.1038/s41598-017-00425-z

Boudreau, S. A., Wang, K., Svensson, P., Sessle, B. J., and Arendt-Nielsen, L. (2009). Vascular and psychophysical effects of topical capsaicin application to orofacial tissues. *J Orofac Pain, 23*(3), 253-264. Retrieved from https://www.ncbi.nlm.nih.gov/pubmed/19639105

Braun, U., Schafer, A., Bassett, D. S., Rausch, F., Schweiger, J. I., Bilek, E.,... Tost, H. (2016). Dynamic brain network reconfiguration as a potential schizophrenia genetic risk mechanism modulated by NMDA receptor function. *Proc Natl Acad Sci U S A, 113*(44), 12568-12573. doi:10.1073/pnas.1608819113

Braun, U., Schafer, A., Walter, H., Erk, S., Romanczuk-Seiferth, N., Haddad, L.,... Bassett, D. S. (2015). Dynamic reconfiguration of frontal brain networks during executive cognition in humans. *Proc Natl Acad Sci U S A, 112*(37), 11678-11683. doi:10.1073/pnas.1422487112

Ceko, M., Kragel, P. A., Woo, C. W., Lopez-Sola, M., and Wager, T. D. (2022). Common and stimulus-type-specific brain representations of negative affect. *Nat Neurosci, 25*(6), 760-770. doi:10.1038/s41593-022-01082-w

Chang, P. F., Arendt-Nielsen, L., Graven-Nielsen, T., Svensson, P., and Chen, A. C. (2001a). Different EEG topographic effects of painful and non-painful intramuscular stimulation in man. *Exp Brain Res, 141*(2), 195-203. doi:10.1007/s002210100864

Chang, P. F., Arendt-Nielsen, L., Graven-Nielsen, T., Svensson, P., and Chen, A. C. (2001b). Topographic effects of tonic cutaneous nociceptive stimulation on human electroencephalograph. *Neurosci Lett, 305*(1), 49-52. doi:10.1016/s0304-3940(01)01802-x

Cheng, J. C., Anzolin, A., Berry, M., Honari, H., Paschali, M., Lazaridou, A.,... Napadow, V. (2022). Dynamic Functional Brain Connectivity Underlying Temporal Summation of Pain in Fibromyalgia. *Arthritis Rheumatol, 74*(4), 700-710. doi:10.1002/art.42013

Cocuzza, C. V., Ito, T., Schultz, D., Bassett, D. S., and Cole, M. W. (2020). Flexible Coordinator and Switcher Hubs for Adaptive Task Control. *J Neurosci, 40*(36), 6949-6968. doi:10.1523/JNEUROSCI.2559-19.2020

Coghill, R. C., Sang, C. N., Berman, K. F., Bennett, G. J., and Iadarola, M. J. (1998). Global cerebral blood flow decreases during pain. *J Cereb Blood Flow Metab, 18*(2), 141-147. doi:10.1097/00004647-199802000-00003

Dubuisson, D., and Dennis, S. G. (1977). The formalin test: a quantitative study of the analgesic effects of morphine, meperidine, and brain stem stimulation in rats and cats. *Pain, 4*(2), 161-174. doi:10.1016/0304-3959(77)90130-0

Farahani, F. V., Karwowski, W., D'Esposito, M., Betzel, R. F., Douglas, P. K., Sobczak, A. M.,... Fafrowicz, M. (2022). Diurnal variations of resting-state fMRI data: A graph-based analysis. *Neuroimage, 256*, 119246. doi:10.1016/j.neuroimage.2022.119246

Finc, K., Bonna, K., He, X., Lydon-Staley, D. M., Kuhn, S., Duch, W., and Bassett, D. S. (2020). Dynamic reconfiguration of functional brain networks during working memory training. *Nat Commun, 11*(1), 2435. doi:10.1038/s41467-020-15631-z

Gifford, G., Crossley, N., Kempton, M. J., Morgan, S., Dazzan, P., Young, J., and McGuire, P. (2020). Resting state fMRI based multilayer network configuration in patients with schizophrenia. *Neuroimage Clin, 25*, 102169. doi:10.1016/j.nicl.2020.102169

Green, B. G. (1991). Temporal characteristics of capsaicin sensitization and desensitization on the tongue. *Physiol Behav, 49*(3), 501-505. doi:10.1016/0031-9384(91)90271-o

Han, S., Cui, Q., Wang, X., Li, L., Li, D., He, Z.,... Chen, H. (2020). Resting state functional network switching rate is differently altered in bipolar disorder and major depressive disorder. *Hum Brain Mapp, 41*(12), 3295-3304. doi:10.1002/hbm.25017

Iadarola, M. J., Berman, K. F., Zeffiro, T. A., Byas-Smith, M. G., Gracely, R. H., Max, M. B., and Bennett, G. J. (1998). Neural activation during acute capsaicin-evoked pain and allodynia assessed with PET. *Brain, 121 ( Pt 5)*, 931-947. doi:10.1093/brain/121.5.931

Insel, T., Cuthbert, B., Garvey, M., Heinssen, R., Pine, D. S., Quinn, K.,... Wang, P. (2010). Research domain criteria (RDoC): toward a new classification framework for research on mental disorders. *Am J Psychiatry, 167*(7), 748-751. doi:10.1176/appi.ajp.2010.09091379

Khambhati, A. N., Mattar, M. G., Wymbs, N. F., Grafton, S. T., and Bassett, D. S. (2018). Beyond modularity: Fine-scale mechanisms and rules for brain network reconfiguration. *Neuroimage, 166*, 385-399. doi:10.1016/j.neuroimage.2017.11.015

Kotov, R., Krueger, R. F., Watson, D., Achenbach, T. M., Althoff, R. R., Bagby, R. M.,... Zimmerman, M. (2017). The Hierarchical Taxonomy of Psychopathology (HiTOP): A dimensional alternative to traditional nosologies. *J Abnorm Psychol, 126*(4), 454-477. doi:10.1037/abn0000258

Lee, J.-J., Kim, H. J., Čeko, M., Park, B.-y., Lee, S. A., Park, H.,... Woo, C.-W. (2021). A neuroimaging biomarker for sustained experimental and clinical pain. *Nature Medicine, 27*(1), 174-182. doi:10.1038/s41591-020-1142-7

Lerman-Sinkoff, D. B., and Barch, D. M. (2016). Network community structure alterations in adult schizophrenia: identification and localization of alterations. *Neuroimage Clin, 10*, 96-106. doi:10.1016/j.nicl.2015.11.011

Lu, S., Baad-Hansen, L., List, T., Zhang, Z., and Svensson, P. (2013). Somatosensory profiling of intra-oral capsaicin and menthol in healthy subjects. *Eur J Oral Sci, 121*(1), 29-35. doi:10.1111/eos.12014

Lydon-Staley, D. M., Ciric, R., Satterthwaite, T. D., and Bassett, D. S. (2019). Evaluation of confound regression strategies for the mitigation of micromovement artifact in studies of dynamic resting-state functional connectivity and multilayer network modularity. *Netw Neurosci, 3*(2), 427-454. doi:10.1162/netn_a_00071

Lydon-Staley, D. M., Kuehner, C., Zamoscik, V., Huffziger, S., Kirsch, P., and Bassett, D. S. (2019). Repetitive negative thinking in daily life and functional connectivity among default mode, fronto-parietal, and salience networks. *Transl Psychiatry, 9*(1), 234. doi:10.1038/s41398-019-0560-0

Mano, H., Kotecha, G., Leibnitz, K., Matsubara, T., Nakae, A., Shenker, N.,... Seymour, B. (2018). Classification and characterisation of brain network changes in chronic back pain: A multicenter study [version 1; referees: 3 approved]. *Wellcome Open Research, 3*(19). doi:10.12688/wellcomeopenres.14069.1

Marquez-Legorreta, E., Constantin, L., Piber, M., Favre-Bulle, I. A., Taylor, M. A., Blevins, A. S.,... Scott, E. K. (2022). Brain-wide visual habituation networks in wild type and fmr1 zebrafish. *Nat Commun, 13*(1), 895. doi:10.1038/s41467-022-28299-4

Mattar, M. G., Thompson-Schill, S. L., and Bassett, D. S. (2018). The network architecture of value learning. *Netw Neurosci, 2*(2), 128-149. doi:10.1162/netn_a_00021

Melzack, R. (1999). From the gate to the neuromatrix. *Pain, Suppl 6*, S121-S126. doi:10.1016/S0304-3959(99)00145-1

Newman, M. E. J. (2010). *Networks : an introduction*. Oxford ; New York: Oxford University Press.

Ngom, P. I., Dubray, C., Woda, A., and Dallel, R. (2001). A human oral capsaicin pain model to assess topical anesthetic-analgesic drugs. *Neurosci Lett, 316*(3), 149-152. doi:10.1016/s0304-3940(01)02401-6

Pedersen, M., Zalesky, A., Omidvarnia, A., and Jackson, G. D. (2018). Multilayer network switching rate predicts brain performance. *Proc Natl Acad Sci U S A, 115*(52), 13376-13381. doi:10.1073/pnas.1814785115

Petrican, R., and Levine, B. T. (2018). Similarity in functional brain architecture between rest and specific task modes: A model of genetic and environmental contributions to episodic memory. *Neuroimage, 179*, 489-504. doi:10.1016/j.neuroimage.2018.06.057

Power, J. D., Barnes, K. A., Snyder, A. Z., Schlaggar, B. L., and Petersen, S. E. (2012). Spurious but systematic correlations in functional connectivity MRI networks arise from subject motion. *Neuroimage, 59*(3), 2142-2154. doi:10.1016/j.neuroimage.2011.10.018

Price, D. D., Hu, J. W., Dubner, R., and Gracely, R. H. (1977). Peripheral suppression of first pain and central summation of second pain evoked by noxious heat pulses. *Pain, 3*(1), 57-68. doi:10.1016/0304-3959(77)90035-5

Price, D. D., Staud, R., Robinson, M. E., Mauderli, A. P., Cannon, R., and Vierck, C. J. (2002). Enhanced temporal summation of second pain and its central modulation in fibromyalgia patients. *Pain, 99*(1-2), 49-59. doi:10.1016/s0304-3959(02)00053-2

Rainville, P., Duncan, G. H., Price, D. D., Carrier, B., and Bushnell, M. C. (1997). Pain affect encoded in human anterior cingulate but not somatosensory cortex. *Science, 277*(5328), 968-971. doi:10.1126/science.277.5328.968

Rainville, P., Feine, J. S., Bushnell, M. C., and Duncan, G. H. (1992). A psychophysical comparison of sensory and affective responses to four modalities of experimental pain. *Somatosens Mot Res, 9*(4), 265-277. doi:10.3109/08990229209144776

Robinson, L. F., Atlas, L. Y., and Wager, T. D. (2015). Dynamic functional connectivity using state-based dynamic community structure: method and application to opioid analgesia. *Neuroimage, 108*, 274-291. doi:10.1016/j.neuroimage.2014.12.034

Segerdahl, A. R., Mezue, M., Okell, T. W., Farrar, J. T., and Tracey, I. (2015). The dorsal posterior insula subserves a fundamental role in human pain. *Nat Neurosci, 18*(4), 499-500. doi:10.1038/nn.3969

Shine, J. M., Koyejo, O., and Poldrack, R. A. (2016). Temporal metastates are associated with differential patterns of time-resolved connectivity, network topology, and attention. *Proc Natl Acad Sci U S A, 113*(35), 9888-9891. doi:10.1073/pnas.1604898113

Stevenson, R. J., and Yeomans, M. R. (1993). Differences in ratings of intensity and pleasantness for the capsaicin burn between chili likers and non-likers; implications for liking development. *Chemical Senses, 18*(5), 471-482. doi:10.1093/chemse/18.5.471

Stohler, C. S., and Kowalski, C. J. (1999). Spatial and temporal summation of sensory and affective dimensions of deep somatic pain. *Pain, 79*(2-3), 165-173. Retrieved from https://www.ncbi.nlm.nih.gov/pubmed/10068162

Telesford, Q. K., Lynall, M. E., Vettel, J., Miller, M. B., Grafton, S. T., and Bassett, D. S. (2016). Detection of functional brain network reconfiguration during task-driven cognitive states. *Neuroimage, 142*, 198-210. doi:10.1016/j.neuroimage.2016.05.078

van den Heuvel, M. P., de Lange, S. C., Zalesky, A., Seguin, C., Yeo, B. T. T., and Schmidt, R. (2017). Proportional thresholding in resting-state fMRI functional connectivity networks and consequences for patient-control connectome studies: Issues and recommendations. *Neuroimage, 152*, 437-449. doi:10.1016/j.neuroimage.2017.02.005

Vlaeyen, J. W. S., and Linton, S. J. (2012). Fear-avoidance model of chronic musculoskeletal pain: 12 years on. *Pain, 153*(6), 1144-1147. doi:10.1016/j.pain.2011.12.009

Waddell, G., Newton, M., Henderson, I., Somerville, D., and Main, C. J. (1993). A Fear-Avoidance Beliefs Questionnaire (FABQ) and the role of fear-avoidance beliefs in chronic low back pain and disability. *Pain, 52*(2), 157-168. doi:10.1016/0304-3959(93)90127-B

Woo, C. W., Chang, L. J., Lindquist, M. A., and Wager, T. D. (2017). Building better biomarkers: brain models in translational neuroimaging. *Nat Neurosci, 20*(3), 365-377. doi:10.1038/nn.4478